# WHY MULTI-GRADE DEEP LEARNING OUTPERFORMS SINGLE-GRADE: THEORY AND PRACTICE

## ABSTRACT

Multi-grade deep learning (MGDL) has recently emerged as an alternative to standard end-to-end training, referred to here as single-grade deep learning (SGDL), showing strong empirical promise. This work provides both theoretical and experimental evidence of MGDL's computational advantages. We establish convergence guarantees for gradient descent (GD) applied to MGDL, demonstrating greater robustness to learning-rate choices compared to SGDL. In the case of ReLU activations with single-layer grades, we further show that MGDL reduces to a sequence of convex optimization subproblems. For more general settings, we analyze the eigenvalue distributions of Jacobian matrices from GD iterations, revealing structural properties underlying MGDL's enhanced stability. Practically, we benchmark MGDL against SGDL on image regression, denoising, and deblurring tasks, as well as on CIFAR-10 and CIFAR-100, covering fully connected networks, CNNs, and transformers. These results establish MGDL as a scalable framework that unites rigorous theoretical guarantees with broad empirical improvements.

## 1 INTRODUCTION

Deep learning has transformed fields from computer vision He et al. (2016); Krizhevsky et al. (2012) to medicine Chen et al. (2018); Jumper et al. (2021) and scientific computing Raissi et al. (2019). Despite these successes, training deep neural networks (DNNs) remains challenging due to non-convex optimization, vanishing/exploding gradients, and spectral bias that favors low-frequency features Rahaman et al. (2019); Xu et al. (2019). Gradient descent can also exhibit short-term oscillations near the Edge of Stability Arora et al. (2022); Cohen et al. (2021), making conventional training inefficient, hard to interpret, and limited in generalization. These challenges motivate multi-grade deep learning (MGDL) Xu (2025), which incrementally builds networks to improve stability, accuracy, and interpretability.

MGDL decomposes end-to-end optimization into a sequence of smaller problems, each training a shallow network on the residuals of previous grades. Previously learned networks remain fixed and act as adaptive "basis" functions or features. This iterative refinement reduces optimization complexity and progressively enhances learning.

MGDL has demonstrated superior performance over standard end-to-end training, which we refer to here as single-grade deep learning (SGDL), in regression Fang & Xu (2024); Xu (2023), oscillatory Fredholm integral equations Jiang & Xu (2024), and PDEs Xu & Zeng (2023), effectively mitigating spectral bias.

We provide a mathematical explanation for why MGDL outperforms SGDL. Focusing on gradient descent, we establish convergence theorems showing MGDL's greater robustness to learning-rate choices. When each grade uses a single ReLU layer, MGDL reduces a highly nonconvex problem to a sequence of convex subproblems, enhancing trainability. Further analysis of a linear surrogate iterative scheme based on the Jacobian of the original map shows that MGDL's eigenvalues lie within $(-1, 1)$, ensuring stable convergence, whereas SGDL's can exceed this range, causing oscillatory loss. Additional experiments benchmark MGDL against SGDL on image regression, denoising, and deblurring tasks, as well as CIFAR-10 and CIFAR-100 classification, using fully connected networks, CNNs, and transformers. These results demonstrate that MGDL unifies rigorous theoretical guarantees with broad empirical improvements as a scalable framework.

Key contributions of this paper:

1. We provide a rigorous convergence analysis of gradient descent for SGDL and MGDL, offering deeper insight into MGDL's computational advantages.

2. We prove that if each grade of MGDL employs a single hidden ReLU layer, the originally nonconvex optimization problem decomposes into a sequence of convex subproblems.

3. Extensive experiments on image regression, denoising, deblurring, CIFAR-10, and CIFAR-100 classification, including fully connected networks, CNNs, and transformers, demonstrate that MGDL consistently outperforms SGDL with greater stability.

4. We analyze the impact of learning rate, showing that MGDL is more robust than SGDL.

5. We study a linear approximation of GD dynamics and the eigenvalue distribution of the associated Jacobian to explain MGDL's convergence and stability advantages.

## 2 STANDARD DEEP LEARNING MODEL

In this section, we review the standard deep learning model and analyze the convergence of the gradient descent (GD) applied to its optimization problem.

A deep neural network (DNN) is a composition of affine maps and nonlinear activations with input layer, $D-1$ hidden layers, and an output layer. Let $d_0 = d$ (input dimension), $d_D = t$ (output dimension), and $d_j$ the width of layer $j$. For $j = 1, \ldots, D$, the weights and biases are $\mathbf{W}_j \in \mathbb{R}^{d_{j-1} \times d_j}$ and $\mathbf{b}_j \in \mathbb{R}^{d_j}$, with ReLU activation $\sigma(x) = \max\{0, x\}$ applied componentwise.

Given $\mathbf{x} \in \mathbb{R}^d$, the hidden layers are defined recursively:

$$\mathcal{H}_1(\mathbf{x}) := \sigma\left(\mathbf{W}_1^\top \mathbf{x} + \mathbf{b}_1\right), \mathcal{H}_{j+1}(\mathbf{x}) := \sigma\left(\mathbf{W}_{j+1}^\top \mathcal{H}_j(\mathbf{x}) + \mathbf{b}_{j+1}\right), j = 1, \ldots, D-2.$$

The output is $\mathcal{N}_D\left(\{\mathbf{W}_j, \mathbf{b}_j\}_{j=1}^D; \mathbf{x}\right) = \mathcal{N}_D(\mathbf{x}) := \mathbf{W}_D^\top \mathcal{H}_{D-1}(\mathbf{x}) + \mathbf{b}_D$. For data $\mathbb{D} = \{(\mathbf{x}_n, \mathbf{y}_n)\}_{n=1}^N$, the loss is

$$\mathcal{L}(\{\mathbf{W}_j, \mathbf{b}_j\}_{j=1}^D; \mathbb{D}) = \frac{1}{2N} \sum_{n=1}^N \|\mathbf{y}_n - \mathcal{N}_D(\mathbf{x}_n)\|^2. \tag{1}$$

The SGDL model minimizes this loss over parameters $\Theta = \{\mathbf{W}_j, \mathbf{b}_j\}_{j=1}^D$, yielding optimal $\Theta^*$ and trained network $\mathcal{N}_D(\Theta^*; \cdot)$.

Among the most common optimization methods for deep learning are stochastic gradient descent (SGD) Kiefer & Wolfowitz (1952); Robbins & Monro (1951) and Adam Kingma & Ba (2015), both rooted in gradient descent (GD). We therefore study GD for minimizing the loss in equation 1.

To facilitate convergence analysis, we stack all parameters $\{\mathbf{W}_j, \mathbf{b}_j\}_{j=1}^D$ into a single vector. For any matrix or vector $\mathbf{A}$, let $A$ denote its vectorization: stacking columns if $\mathbf{A}$ is a matrix, taking $A = \mathbf{A}$ if it is a column vector, and $A = \mathbf{A}^\top$ if a row vector. The parameter vector is $W := \left(W_1^\top, b_1^\top, \ldots, W_D^\top, b_D^\top\right)^\top$, with total dimension $M = \sum_{j=1}^D (d_{j-1} + 1)d_j$.

We consider GD for a general objective $\mathcal{F} : \mathbb{R}^M \to \mathbb{R}$, assumed nonnegative, twice continuously differentiable, and generally nonconvex. The iteration is

$$W^{k+1} = W^k - \eta \frac{\partial \mathcal{F}}{\partial W}(W^k), \tag{2}$$

where $k$ is the iteration index and $\eta > 0$ the learning rate. In our setting, $\mathcal{F}$ is the loss $\mathcal{L}$ in equation 1.

We analyze the convergence of GD for minimizing equation 1. Assume there exists a compact convex set $\mathcal{W} \subset \mathbb{R}^M$ such that for some $\eta_0 > 0$, all GD iterates $W^k{}_{k=0}^\infty$ from equation 2 with $\mathcal{F} = \mathcal{L}$ remain in $\mathcal{W}$ whenever $\eta \in (0, \eta_0)$. Convergence depends on the Hessian of $\mathcal{L}$ over $\mathcal{W}$, where we set $\alpha := \sup_{W \in \mathcal{W}} \|\mathbf{H}_\mathcal{L}(W)\|$, with $\|\cdot\|$ the spectral norm. Since $\mathbf{H}_\mathcal{L}(W) \in \mathbb{R}^{M \times M}$, $\alpha$ captures the effect of network depth and size.

The following theorem, proved in Appendix A, establishes convergence of GD with $\mathcal{F} := \mathcal{L}$, extending Theorem 6 in Xu (2025), which assumes zero biases.

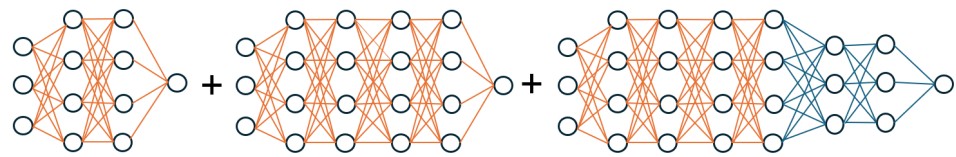

Figure 1: MGDL at grade 3: blue lines denote trainable parameters, while orange lines represent parameters fixed from grades 1 (first term) and 2 (second term). At this grade, only a shallow network is trained, building on features learned in the previous grades.

**Theorem 1.** *Let $\{W^k\}_{k=0}^{\infty}$ be generated by equation 2 with $\mathcal{F} = \mathcal{L}$ and initial guess $W^0$. Suppose $\sigma$ is twice continuously differentiable and the iterates remain in a convex compact set $\mathcal{W} \subset \mathbb{R}^M$. If the learning rate $\eta \in (0, 2/\alpha)$, then:*

    *(i) $\lim_{k\to\infty} \mathcal{L}(W^k) = L^*$ for some $L^* \geq 0$;*

    *(ii) $\lim_{k\to\infty} \frac{\partial \mathcal{L}}{\partial W}(W^k) = 0$;*

    *(iii) Every cluster point $\hat{W}$ of $\{W^k\}$ satisfies $\frac{\partial \mathcal{L}}{\partial W}(\hat{W}) = 0$.*

Deep neural networks are defined by weight matrices and bias vectors, with parameter counts scaling rapidly with depth—for example, LeNet-5 has 60K parameters LeCun et al. (1998), ResNet-152 60.2M He et al. (2016), and GPT-3 175B Brown et al. (2020). End-to-end training at such scales is hampered by optimization and stability issues: (i) deeper networks induce highly nonconvex loss landscapes, often trapping solutions in poor local minima Bengio et al. (2006); and (ii) training suffers from vanishing or exploding gradients, which impede convergence Glorot & Bengio (2010); Goodfellow et al. (2016); Pascanu et al. (2013). To address these challenges, multi-grade deep learning (MGDL) Xu (2025) trains networks in stages, where each shallow grade builds on the residuals of the previous one and propagates its output forward, incrementally approximating the target function.

## 3 MULTI-GRADE DEEP LEARNING

This section reviews MGDL, and analyzes GD convergence at each grade.

Given data $\mathbb{D} = \{(\mathbf{x}_n, \mathbf{y}_n)\}_{n=1}^N$, MGDL decomposes learning a depth-$D$ DNN into $L < D$ sequential grades. Each grade trains a shallow network $\mathcal{N}_{D_l}$ on residuals from the previous grade, with depths $1 < D_l < D$ and $\sum_{l=1}^L D_l = D + L - 1$. Let $\Theta_l = \{\mathbf{W}_{lj}, \mathbf{b}_{lj}\}_{j=1}^{D_l}$ denote grade-$l$ parameters. The model is defined recursively by

$$g_1(\Theta_1; \mathbf{x}) := \mathcal{N}_{D_1}(\Theta_1; \mathbf{x}), \; g_{l+1}(\Theta_{l+1}; \mathbf{x}) := \mathcal{N}_{D_{l+1}}(\Theta_{l+1}; \cdot) \circ \mathcal{H}_{D_l-1}(\Theta_l^*; \cdot) \circ \cdots \circ \mathcal{H}_{D_1-1}(\Theta_1^*; \cdot)(\mathbf{x}). \quad (3)$$

The grade-$l$ loss is

$$\mathcal{L}_l(\Theta_l; \mathbb{D}) = \frac{1}{2N} \sum_{n=1}^N \|\mathbf{e}_{ln} - g_l(\Theta_l; \mathbf{x}_n)\|^2, \quad (4)$$

with residuals $\mathbf{e}_{1n} = \mathbf{y}_n$ and $\mathbf{e}_{(l+1)n} = \mathbf{e}_{ln} - g_l(\Theta_l^*; \mathbf{x}_n)$. Each $\Theta_l^*$ minimizes $\mathcal{L}_l$ given earlier grades. After $L$ grades, the MGDL output is $\bar{g}_L(\{\Theta_l^*\}_{l=1}^L; \mathbf{x}) = \sum_{l=1}^L g_l(\Theta_l^*; \mathbf{x})$. Figure 1 illustrates the multi-grade architecture at grade three.

For optimization, set $\mathbf{x}_{1n} := \mathbf{x}_n$ and recursively define $\mathbf{x}_{ln} := \mathcal{H}_{D_{l-1}-1}(\Theta_{l-1}^*; \cdot) \circ \cdots \circ \mathcal{H}_{D_1-1}(\Theta_1^*; \cdot)(\mathbf{x}_n)$, and dataset $\mathbb{D}_l = \{(\mathbf{x}_{ln}, \mathbf{e}_{ln})\}_{n=1}^N$. The grade-$l$ loss is

$$\mathcal{L}_l(\Theta_l; \mathbb{D}_l) = \frac{1}{2N} \sum_{n=1}^N \|\mathbf{e}_{ln} - \mathcal{N}_{D_l}(\Theta_l; \mathbf{x}_{ln})\|^2.$$

MGDL's training time scales linearly with the number of grades (assuming comparable layer and neuron counts), while its memory cost is much lower than that of a single deep network, since each grade trains only a shallow model. Let $W_l := (W_{l1}^\top, b_{l1}^\top, \ldots, W_{lD_l}^\top, b_{lD_l}^\top)^\top \in \mathbb{R}^{M_l}$, with $M_l := \sum_{j=1}^{D_l} (d_{l(j-1)} + 1)d_{lj}$. The GD iteration is

$$W_l^{k+1} = W_l^k - \eta_l \frac{\partial \mathcal{L}_l}{\partial W_l}(W_l^k).$$

Assuming $W_l^k \subset \mathcal{W}_l$ for some convex compact $\mathcal{W}_l \subset \mathbb{R}^{M_l}$, define $\alpha_l := \sup_{W_l \in \mathcal{W}_l} \|\mathbf{H}_{\mathcal{L}_l}(W_l)\|$.

**Theorem 2.** *Let $\{W_l^k\}$ be generated by the above GD iteration from $W_l^0$. Assume $\{(\mathbf{x}_{ln}, \mathbf{e}_{ln})\} \subset \mathbb{R}^{d_{l0}} \times \mathbb{R}^d$ is bounded, $\sigma$ is twice continuously differentiable, and $\{W_l^k\} \subset \mathcal{W}_l$. If $\eta_l \in (0, 2/\alpha_l)$, then*

*(i)* $\lim_{k\to\infty} \mathcal{L}_l(W_l^k) = L_l^*$ *for some* $L_l^* \geq 0$;

*(ii)* $\lim_{k\to\infty} \frac{\partial \mathcal{L}_l}{\partial W_l}(W_l^k) = 0$;

*(iii)* *Every cluster point $\hat{W}_l$ of $\{W_l^k\}$ satisfies $\frac{\partial \mathcal{L}_l}{\partial W_l}(\hat{W}_l) = 0$.*

Theorem 2, proved in Appendix A, parallels Theorem 1, with the key distinction that MGDL optimizes shallow subproblems at each grade. This mitigates vanishing/exploding gradients and allows a broader admissible learning-rate range ($\eta_l \in (0, 2/\alpha_l)$ with $\alpha_l \ll \alpha$), thereby improving stability and robustness compared to SGDL.

# 4 CONVEX OPTIMIZATION IN MGDL WITH SINGLE-LAYER ReLU GRADES

In this section, we show that when each grade in MGDL is realized as a single hidden-layer ReLU network, the overall nonconvex optimization problem decomposes into a sequence of convex subproblems. For clarity, we consider bias-free networks with scalar output; the extension to biased networks is analogous.

A two-layer ReLU network with $m$ neurons is $\tilde{\mathcal{N}}(\mathbf{x}) := \sum_{j=1}^m \alpha_j \sigma(\tilde{\mathbf{w}}_j^\top \mathbf{x})$, with hidden parameters $\tilde{\mathbf{w}}_j$ and outputs $\alpha_j$. Since $\sigma(a\mathbf{x}) = a\sigma(\mathbf{x})$ for $a \geq 0$, each term can be written as $\alpha_j \sigma(\tilde{\mathbf{w}}_j^\top \mathbf{x}) = \sigma(\mathbf{w}_j^\top \mathbf{x}) - \sigma(\mathbf{v}_j^\top \mathbf{x})$ for suitable $\mathbf{w}_j, \mathbf{v}_j$, making $\tilde{\mathcal{N}}$ equivalent to

$$\mathcal{N}(\mathbf{x}) := \sum_{j=1}^m \left(\sigma(\mathbf{w}_j^\top \mathbf{x}) - \sigma(\mathbf{v}_j^\top \mathbf{x})\right), \tag{5}$$

which we adopt as the building block of MGDL.

Suppose grade $l$ of MGDL is a single hidden-layer ReLU network with $2m_l$ neurons. By equation 5, its output is

$$(\mathcal{N}_l \circ h_{l-1}^*)(\mathbf{x}) := \sum_{j=1}^{m_l} \left(\sigma(\mathbf{w}_{lj}^\top h_{l-1}^*(\mathbf{x})) - \sigma(\mathbf{v}_{lj}^\top h_{l-1}^*(\mathbf{x}))\right). \tag{6}$$

The input features $h_{l-1}^*$ are defined recursively by $h_0^*(\mathbf{x}) = \mathbf{x}$, and $h_{l-1}^*(\mathbf{x}) := (\mathcal{H}_{l-1}^* \circ \cdots \circ \mathcal{H}_1^*)(\mathbf{x})$, with feature map

$$\mathcal{H}_k^*(\mathbf{z}) = \left(\sigma((\mathbf{w}_{k1}^*)^\top \mathbf{z}), \ldots, \sigma((\mathbf{w}_{km_k}^*)^\top \mathbf{z}), \sigma((\mathbf{v}_{k1}^*)^\top \mathbf{z}), \ldots, \sigma((\mathbf{v}_{km_k}^*)^\top \mathbf{z})\right)^\top, \quad k \in \mathbb{N}_{l-1}.$$

Let the data matrix at grade $l$ be $\mathbf{X}_l := [\mathbf{x}_{l1}, \ldots, \mathbf{x}_{lN}]^\top \in \mathbb{R}^{N \times d_l}$ with $\mathbf{x}_{ln} := h_{l-1}^*(\mathbf{x}_n)$. At grade $l$, we solve the nonconvex problem

$$p_l^* := \min_{\{\mathbf{w}_{lj}, \mathbf{v}_{lj}\}_{j=1}^{m_l}} \frac{1}{2} \left\| \sum_{j=1}^{m_l} \left(\sigma(\mathbf{X}_l \mathbf{w}_{lj}) - \sigma(\mathbf{X}_l \mathbf{v}_{lj})\right) - \mathbf{e}_l \right\|^2. \tag{7}$$

Following Pilanci & Ergen (2020), we show that equation 7 is equivalent to a convex program. For any $\mathbf{w}_l \in \mathbb{R}^{m_{l-1}}$, define $\text{diag}(1[\mathbf{X}_l \mathbf{w}_l \geq 0])$, where $1[\mathbf{X}_l \mathbf{w}_l \geq 0] \in \{0, 1\}^N$ with entries $1[\mathbf{x}_{ln}^\top \mathbf{w}_l \geq 0]$. Since $\mathbf{X}_l$ is fixed, only finitely many such matrices exist Cover (2006); Stanley et al. (2007); denote them $\mathbf{D}_{l1}, \ldots, \mathbf{D}_{lP_l}$. This induces a partition $\{C_{li}\}_{i=1}^{P_l}$ of $\mathbb{R}^{m_{l-1}}$, where $C_{li} := \{\mathbf{w}_l : (2\mathbf{D}_{li} - \mathbf{I}_N)\mathbf{X}_l \mathbf{w}_l \geq 0\}$. Each $C_{li}$ is convex, closed under addition, and satisfies $\mathbb{R}^{m_{l-1}} = \bigcup_{i=1}^{P_l} C_{li}$. Within $C_{li}$, ReLU is linear, that is, $\sigma(\mathbf{X}_l \mathbf{w}_l) = \mathbf{D}_{li} \mathbf{X}_l \mathbf{w}_l$, for $\mathbf{w}_l \in C_{li}$.

Using this, we introduce the convex program

$$q_l^* := \min_{\{\mathbf{w}_{li}, \mathbf{v}_{li} \in C_{li}\}_{i=1}^{P_l}} \frac{1}{2} \left\| \sum_{i=1}^{P_l} \mathbf{D}_{li} \mathbf{X}_l (\mathbf{w}_{li} - \mathbf{v}_{li}) - \mathbf{e}_l \right\|^2. \tag{8}$$

**Theorem 3.** *Let $\sigma$ be ReLU. If $m_l \geq P_l$, then problems equation 7 and equation 8 attain the same optimal value. Moreover, any optimal solution of equation 8 is also optimal for equation 7 when $m_l = P_l$.*

*Proof.* Linearity within each region implies that feasible points of equation 8 are feasible for equation 7, hence $p_l^* \leq q_l^*$. Conversely, given an optimal solution $\{\mathbf{w}_{lj}^*, \mathbf{v}_{lj}^*\}$ of equation 7, regrouping parameters by the partition $\{C_{li}\}$ and using closure under addition yields aggregated vectors $\tilde{\mathbf{w}}_{li}^*, \tilde{\mathbf{v}}_{li}^*$ that form a feasible point of equation 8 with the same objective value, so $q_l^* \leq p_l^*$. Thus $p_l^* = q_l^*$. When $m_l = P_l$, the correspondence is exact, and optimal solutions coincide. $\square$

Unlike Pilanci & Ergen (2020), which convexifies single hidden-layer ReLU networks via explicit regularization, our multi-grade decomposition reformulates deep ReLU networks as a sequence of convex programs, extending convexification from shallow to deep architectures.

## 5 PERFORMANCE COMPARISON OF MGDL AND SGDL

In this section, we compare MGDL and SGDL on image reconstruction tasks—regression, denoising, and deblurring—as well as on the CIFAR-100 classification dataset Krizhevsky (2009). The results demonstrate that MGDL consistently outperforms SGDL, which suffers from training instability and lower accuracy.

For image reconstruction, we employ full connected networks for both SGDL and MGDL, and evaluate performance using PSNR equation 30. For classification, we use convolutional neural networks (CNNs). In both cases, ReLU activations are applied, and training is performed using the Adam optimizer Kingma & Ba (2015). Overall, MGDL achieves superior stability and accuracy across both reconstruction and classification tasks.

**Image regression.** We model grayscale images as functions $f : \mathbb{R}^2 \to \mathbb{R}$, mapping pixel coordinates to intensity values. The training set consists of a regularly spaced grid covering one quarter of the pixels, while the test set includes all pixels. We evaluate SGDL and MGDL on six images of varying sizes (Figure 9). For images (b)–(f), we use the fully connected architecture in 26 with $(n_{in}, n_{out}, n_{hidden}, n_h) = (2, 1, 128, 8)$ for SGDL and the architecture in 27 with $(n_{in}, n_{out}, n_{hidden}, n_h, L) = (2, 1, 128, 2, 4)$ for MGDL. For image (g), we employ a deeper network, setting $G = 12$ for SGDL and $g = 3$ for MGDL.

Numerical results are summarized in Table 1 and Figure 11. Table 1 reports PSNR values, showing that MGDL consistently outperforms SGDL with gains of 0.42–3.94 dB across all testing images. Figure 11 plots the training losses: SGDL exhibits persistent oscillations for all images, while MGDL shows image-dependent behavior. For Barbara, Butterfly, and Walnut, MGDL oscillates initially but stabilizes in later stages, whereas for Pirate and Chest, oscillations appear earlier before converging. Overall, MGDL tends to stabilize or decrease steadily over time, in contrast to the sustained oscillations of SGDL.

The *Cameraman* image further illustrates these differences. Figures 10(a)–(b) show the training losses: SGDL suffers from strong oscillations, leading to unstable predictions, as seen in Figures 10(c)–(f) at iterations 9800, 9850, 9900, and 9950, with corresponding PSNR fluctuations. In contrast, MGDL exhibits a steadily decreasing loss (b), and its predictions (g)–(j) improve consistently across iterations. These results highlight the robustness and reliability of MGDL compared with SGDL in image regression tasks.

**Image denoising.** We address the problem of recovering a clean image $\mathbf{f} \in \mathbb{R}^{n \times n}$ from a noisy observation $\hat{\mathbf{f}} := \mathbf{f} + \boldsymbol{\epsilon}$, where the noise entries are i.i.d. Gaussian with zero mean and standard deviation $s$, i.e., $[\boldsymbol{\epsilon}]_{i,j} \sim \mathcal{N}(0, s^2)$. The optimization problem is formulated in Appendix B, with the transform operator $\mathbf{A}$ set to the identity.

SGDL adopts structure 26 $(2, 1, 128, 12)$, while MGDL uses 27 $(2, 1, 128, 3, 4)$. We test six noise levels, $s = 10, 20, 30, 40, 50, 60$, as illustrated in Figure 12. Results are summarized in Table 2 and Figures 13-15. MGDL consistently outperforms SGDL with PSNR gains of 0.16–4.23 dB. During training, SGDL shows persistent oscillations, while MGDL improves steadily, especially from grades 2–4.

Table 1: PSNR comparison for image regression.

| Image | Method | TrPSNR | TePSNR |
|---|---|---|---|
| Cameraman | SGDL | 27.05 | 24.79 |
| | MGDL | **31.80** | **25.21** |
| Barbara | SGDL | 23.14 | 22.75 |
| | MGDL | **24.36** | **23.84** |
| Butterfly | SGDL | 26.22 | 24.87 |
| | MGDL | **28.23** | **27.06** |
| Pirate | SGDL | 24.20 | 24.34 |
| | MGDL | **27.40** | **26.45** |
| Chest | SGDL | 34.77 | 34.56 |
| | MGDL | **39.44** | **38.50** |
| Walnut | SGDL | 19.94 | 20.05 |
| | MGDL | **21.83** | **21.31** |

Table 2: PSNR comparison for image denoising.

| Noise | Method | Butterfly | Pirate | Chest |
|---|---|---|---|---|
| 10 | SGDL | 27.53 | 25.13 | 36.20 |
| | MGDL | **31.67** | **29.36** | **38.58** |
| 20 | SGDL | 26.73 | 25.02 | 35.34 |
| | MGDL | **28.39** | **27.74** | **36.89** |
| 30 | SGDL | 26.05 | 24.63 | 34.30 |
| | MGDL | **27.09** | **27.20** | **35.48** |
| 40 | SGDL | 25.54 | 24.47 | 33.55 |
| | MGDL | **26.37** | **26.25** | **34.61** |
| 50 | SGDL | 24.65 | 24.01 | 33.51 |
| | MGDL | **25.84** | **25.77** | **33.94** |
| 60 | SGDL | 24.30 | 23.82 | 32.90 |
| | MGDL | **25.21** | **25.32** | **33.06** |

Table 3: PSNR comparison for image deblurring.

| image | method | 3 | 5 | 7 |
|---|---|---|---|---|
| Butterfly | SGDL | 25.43 | 24.20 | 22.70 |
| | MGDL | **27.06** | **25.19** | **23.65** |
| Pirate | SGDL | 24.72 | 23.79 | 23.13 |
| | MGDL | **26.47** | **24.95** | **23.98** |
| Chest | SGDL | 35.40 | 34.61 | 33.69 |
| | MGDL | **38.24** | **36.51** | **35.14** |

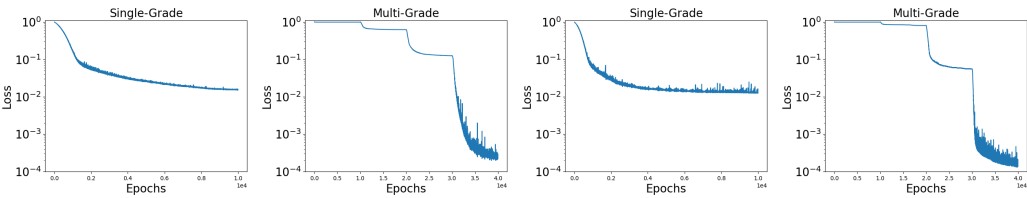

Figure 2: Impact of learning rate.

**Image deblurring.** We address the problem of recovering $\mathbf{f}$ from a blurred observation $\hat{\mathbf{f}} := \mathbf{Kf} + \boldsymbol{\epsilon}$, where $\mathbf{K}$ is a Gaussian blurring operator and $[\boldsymbol{\epsilon}]_{i,j} \sim \mathcal{N}(0, s^2)$ with $s = 3$. The optimization problem and operator $\mathbf{A} = \mathbf{K}$ are detailed in Appendix B.

The SGDL and MGDL structures are the same as those used in *Image Denoising*, respectively. We test three blurring levels ($\hat{s} = 3$, $\hat{s} = 5$, and $\hat{s} = 7$; Figure 16). Results are summarized in Table 3 and Figures 17-19. MGDL achieves PSNR improvements of 0.85–2.84 dB over SGDL. While SGDL exhibits strong PSNR oscillations during training, MGDL shows stable and consistent gains, particularly from grades 2 to 4.

**Classification on CIFAR-100.** We address the problem of image classification on the CIFAR-100 dataset, evaluating SGDL and MGDL in terms of both accuracy and training dynamics. We use mean squared error (MSE) as the loss function, with architectures specified in equation 28 and 29.

We test two learning rates, $5 \times 10^{-4}$ and $1 \times 10^{-4}$. Results are shown in Figure 3. For both settings, SGDL converges to a loss around $10^{-2}$, whereas MGDL reaches approximately $10^{-4}$, nearly two orders of magnitude lower. In terms of stability, SGDL begins oscillating once the loss falls below $10^{-1}$, while MGDL remains stable until reaching $10^{-3}$. These results demonstrate that MGDL delivers superior accuracy and significantly greater training stability compared to SGDL.

Results on image reconstruction and CIFAR-100 classification show that MGDL consistently outperforms SGDL. Whereas SGDL exhibits pronounced oscillations in loss or PSNR during training, MGDL achieves a steady decrease in loss or a consistent increase in PSNR. The underlying reasons are analyzed in Section 7.

Figure 3: Training on CIFAR-100 using SGDL and MGDL(1-2:$\eta = 5 \times 10^{-5}$, 3-4: $\eta = 1 \times 10^{-4}$).

## 6   Impact of Learning Rate on SGDL and MGDL

We examine the effect of learning rate on SGDL and MGDL, both trained using gradient descent.

**Synthetic data regression.** We approximate $g : [0, 1] \to \mathbb{R}$ defined by $g(\mathbf{x}) := \sum_{j=1}^{M} \sin(2\pi\kappa_j \mathbf{x} + \varphi_j)$, $\mathbf{x} \in [0, 1]$, where $\varphi_j \sim \mathcal{U}(0, 2\pi)$. Two settings are considered: (1) $M = 3, \kappa = [1, 5.5, 10]$; (2) $M = 5, \kappa = [1, 8.25, 15.5, 22.75, 30]$. The training set contains 1,024 equally spaced points, and the validation set 1,000 uniformly sampled points.

SGDL adopts structure 26 $(1, 1, 32, 4)$, while MGDL uses structure 27 $(1, 1, 32, 1, 4)$. Learning rates are selected from $[0.001, 0.5]$, with $10^6$ training epochs. Figure 2 illustrates the impact of learning rate (left: Setting 1, right: Setting 2; 'NaN' indicates divergence). In Setting 1 (low-frequency function), both methods perform well, while MGDL is robust across a wider range: SGDL achieves loss $< 0.001$ only for $\eta \in [0.03, 0.08]$, whereas MGDL sustains this performance for $\eta \in [0.01, 0.3]$. In Setting 2 (high-frequency function), SGDL converges only at $\eta \approx 0.005$ and diverges for larger rates, while MGDL remains stable with loss $< 0.01$ for $\eta \in [0.08, 0.3]$.

**Image regression.** We consider image regression as in Section 5. SGDL use 26 $(2, 1, 128, 8)$, while MGDL uses 27 $(2, 1, 128, 2, 4)$. Learning rates are selected from $[0.001, 1]$, with $10^5$ training epochs. Figure 20 illustrates results on 'Resolution Chart', 'Cameraman', 'Barbara', and 'Pirate'. MGDL consistently achieves higher accuracy, while SGDL fails on 'Cameraman' and 'Pirate' for $\eta$ near 1. MGDL remains stable across this wide range of learning rates.

**Summary.** Across both synthetic and image regression, MGDL demonstrates markedly greater robustness to the choice of learning rate, maintaining effective training and high accuracy over a wider interval, whereas SGDL is sensitive and often fails with large learning rates.

## 7   Eigenvalue Analysis for SGDL and MGDL

We analyze gradient descent (GD) equation 2 for SGDL and MGDL, expressing it as a Picard iteration $W^{k+1} = (\mathbf{I} - \eta\frac{\partial \mathcal{F}}{\partial W})W^k$ and linearizing the gradient via Taylor expansion: $\frac{\partial \mathcal{F}}{\partial W}(W^k) = \mathbf{H}_{\mathcal{F}}(W^{k-1})W^k + u^{k-1} + r^{k-1}$, with remainder $r^{k-1}$ of order $(W^k - W^{k-1})^2$. Neglecting $r^{k-1}$ gives the linearized update

$$\tilde{W}^{k+1} = \mathbf{A}^{k-1}\tilde{W}^k - \eta u^{k-1}, \quad \mathbf{A}^{k-1} = \mathbf{I} - \eta\mathbf{H}_{\mathcal{F}}(W^{k-1}).$$

**Theorem 4.** *Let $\mathcal{F} : \mathbb{R}^M \to \mathbb{R}$ be nonnegative and twice continuously differentiable, with $\{W^k\} \subset \Omega$, a convex compact set. If $\tau := \sup_{W \in \Omega} \|\mathbf{I} - \eta\mathbf{H}_{\mathcal{F}}(W)\| < 1$, then $\{\tilde{W}^k\}$ converges. Moreover, if $\mathcal{F}$ is thrice continuously differentiable, the sequences $\{W^k\}$ and $\{\tilde{W}^k\}$ (with matching initializations) converge to the same limit if $\tau < 1$.*

Hence, convergence is governed by the spectrum of $\mathbf{I} - \eta\mathbf{H}_{\mathcal{F}}(W)$. Eigenvalues in $(-1, 1)$ ensure stable loss decay. Explicit Hessians for SGDL ($\mathcal{F} = \mathcal{L}$) and MGDL ($\mathcal{F} = \mathcal{L}_l$) under ReLU are given in the Supplementary Material.

We next monitor the eigenvalues of $\mathbf{I} - \eta\mathbf{H}_{\mathcal{F}}(W^k)$ during training. In deep networks such as SGDL, these eigenvalues often exit $(-1, 1)$, producing oscillatory loss. In contrast, the shallower structure of MGDL keeps them inside $(-1, 1)$, leading to smooth loss decay.

**Synthetic data regression.** Setup follows *Synthetic data regression* in Section 6. Both models are trained via gradient descent with learning rate $\eta \in [0.001, 0.5]$, selected by lowest validation loss. Results are shown in Figures 4 (Setting 1) and 21 (Setting 2).

For SGDL under Setting 1, Figure 4 (first subfigure) shows the ten smallest (solid) and ten largest (dashed) eigenvalues during training ($10^6$ epochs). The smallest eigenvalue drops well below $-1$, while indices 1–5 stay near $-1$. The largest eigenvalues slightly exceed 1. The loss decreases overall but oscillates, correlating with the number of eigenvalues below or near $-1$.

For MGDL, the ten smallest eigenvalues remain within $(-1, 1)$ across grades 1–4, while the largest stay slightly above 1, producing smooth loss decay (Figure 4, second and fourth subfigures).

In Setting 2 (higher-frequency target), SGDL's eigenvalues initially stay in $(-1, 1)$ but later drop to $-1$, causing strong loss oscillations up to $10^6$ epochs. MGDL maintains eigenvalues in $(-1, 1)$,

ensuring stable training and better accuracy (Figure 22, third and fourth subfigures). Across both settings, the smallest eigenvalue predominantly determines loss behavior.

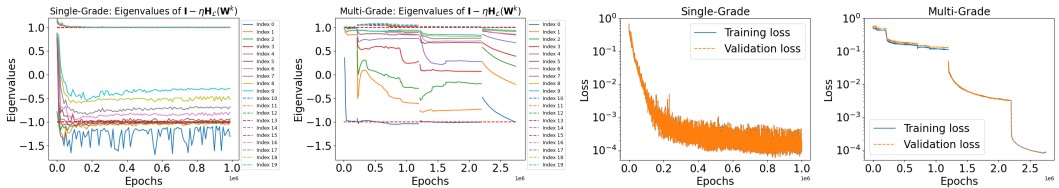

Figure 4: Training process of SGDL ($\eta = 0.08$) and MGDL ($\eta = 0.06$) for Setting 1.

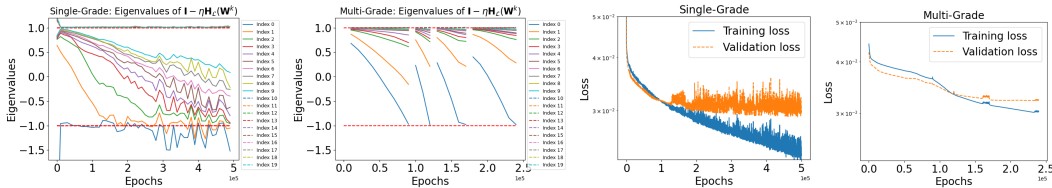

Figure 5: Training process of SGDL ($\eta = 0.02$) and MGDL ($\eta = 0.2$) for 'Resolution chart'.

**Image regression.** Following Section 5, shallow networks are used to enable Hessian computation: SGDL with architecture 26 $(2, 1, 48, 4)$ and MGDL with architecture 27 $(2, 1, 48, 1, 4)$. For SGDL, the smallest eigenvalue approaching $-1$ causes oscillatory loss, while MGDL's eigenvalues remain in $(-1, 1)$, yielding stable reduction (Figures 5-25).

**Image denoising.** SGDL's smallest eigenvalue approaches $-1$, causing oscillatory loss; MGDL keeps all eigenvalues in $(-1, 1)$, ensuring steady reduction (Figures 26-29).

**CIFAR-10 classification.** Using 10,000 sampled images, fully connected ReLU networks (26 $(3072, 10, 128, 8)$ for SGDL and 27 $(3072, 10, 128, 2, 4)$ for MGDL) are trained with squared loss and full-batch gradient descent (Figure 6). With learning rate 0.004 0.004, SGDL reaches loss $7.16 \times 10^{-3}$ in 26,878 s; MGDL achieves $2.56 \times 10^{-3}$ in 22,177 s. SGDL shows strong oscillations with eigenvalues often below $-1$, whereas MGDL exhibits mild oscillations in grade 1 and smooth loss reduction in subsequent grades, with eigenvalues strictly within $(-1, 1)$.

Across tasks— synthetic regression, image regression/denoising, and CIFAR-10—SGDL's eigenvalues often fall below $-1$, causing loss oscillations, while MGDL's stay within $(-1, 1)$, explaining its superior stability.

# 8 MULTI-GRADE TRANSFORMERS (MGT)

The Transformer Vaswani et al. (2017) is a widely used architecture based on self-attention, enabling global information exchange. We introduce a MGT and apply it to time series regression.

A single-grade Transformer (SGT) embeds inputs into $d_{\text{model}}$-dimensional vectors with positional encoding, processes them through $n_h$ Transformer blocks (self-attention + feedforward with residu-

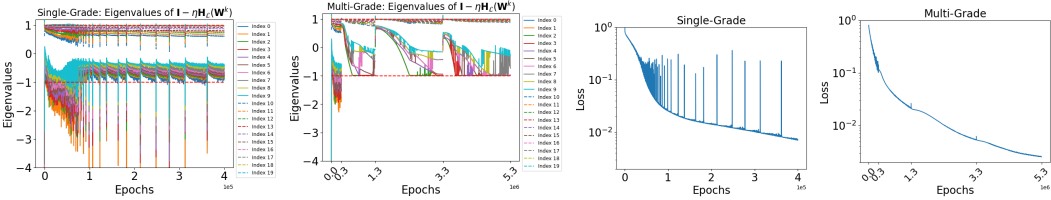

Figure 6: Training on CIFAR-10 using SGDL and MGDL with learning rate $4 \times 10^{-3}$.

als), and outputs predictions:

$$\text{Input} \rightarrow \text{Embedding} \rightarrow (\text{Attention}(d_{\text{model}}, n_{\text{head}}) + \text{MLP}) \times n_h \rightarrow \text{Output}. \tag{9}$$

MGT trains multiple grades, each a Transformer of form equation 9 with a single block. Grade 1 uses positional encoding, while later grades inherit positional information and refine residuals. Unlike SGT, which trains a deep stack at once, MGT decomposes training into smaller stages, yielding greater stability, fewer oscillations, and improved convergence and generalization.

**Time series regression on synthetic data.** We consider predicting the next $s = 1$ value from the past $d = 64$ observations, with problem settings, data generation, and network architectures detailed in Appendix C. The first $80\%$ of the sequence is used for training and the last $20\%$ for testing.

Table 4 reports the training and testing mean squared errors (TrMSE, TeMSE), while Figure 7 shows predictions on data. Although both methods fit the training data effectively, MGT achieves significantly better generalization, attaining a test error of $1.6 \times 10^{-1}$ compared to 2.6 for SGT, while requiring only 28% of the training time. As shown in Figure 7, SGT's predictions deteriorate sharply when test sequences deviate from the training distribution, while MGT maintains accurate predictions.

Table 4: Synthetic time series

| | TrMSE | TeMSE | Time (s) |
|---|---|---|---|
| MGT | $1.2 \times 10^{-2}$ | $1.6 \times 10^{-1}$ | 741 |
| SGT | $7.1 \times 10^{-2}$ | $2.6 \times 10^{0}$ | 2,693 |

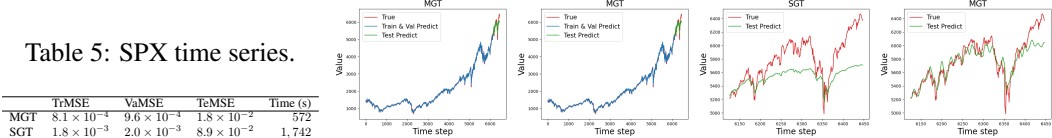

Figure 7: Synthetic time series: train/test (1–2) and zoomed test (3–4).

**Time series regression on financial data.** We analyze the SPX (S&P 500 Index) using daily data from Yahoo Finance or Bloomberg , spanning January 1, 2000, to August 22, 2025. The task is to predict the next $s = 1$ value from the past $d = 20$ observations. Details on data preparation and architectures are given in Appendix C. The last $5\%$ of the data is reserved for testing, with $5\%$ of the remainder for validation and the rest for training.

Table 5 summarizes mean squared errors (TrMSE, VaMSE, TeMSE), and Figure 8 shows predictions. Although oth models fit the training data affectively, MGT achieves substantially better generalization, attaining a test error of $1.8 \times 10^{-2}$ compared to $8.9 \times 10^{-2}$ for SGT, and requires only 33% of the training time. Crucially, as shown in Figure 8, SGT collapses under distribution shift, with predictions diverging sharply from reality, whereas MGT remains accurate and stable throughout.

Table 5: SPX time series.

| | TrMSE | VaMSE | TeMSE | Time (s) |
|---|---|---|---|---|
| MGT | $8.1 \times 10^{-4}$ | $9.6 \times 10^{-4}$ | $1.8 \times 10^{-2}$ | 572 |
| SGT | $1.8 \times 10^{-3}$ | $2.0 \times 10^{-3}$ | $8.9 \times 10^{-2}$ | 1,742 |

Figure 8: SPX time series: train/val/test (1–2), zoomed test (3–4)

## 9 CONCLUSION

We analyzed MGDL from both theoretical and numerical perspectives. Spectral analysis revealed that MGDL keeps eigenvalues of the iteration matrix within $(-1, 1)$, ensuring stable convergence, while SGDL often produces eigenvalues outside this range, leading to oscillatory training. A convergence theorem further confirmed that eigenvalue behavior governs loss dynamics. Experiments on synthetic regression, image reconstruction, and classification consistently showed MGDL's advantages: greater stability, robustness to learning rates, and better accuracy in challenging settings. These results establish MGDL as a principled and effective alternative to SGDL, combining convex reformulations with practical performance gains.

**Use of Large Language Models.** Large Language Models were used to refine the text and ensure grammatical accuracy.

**Reproducibility Statement.** Anonymous code and instructions for all experiments are provided in the supplementary material: `Why MGDL outperforms SGDL`.

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

## A  CONVERGENCE PROOF

**Proofs of Theorem 1 and Theorem 2**

We begin by establishing the convergence of the general gradient descent iteration 2, which serves as the foundation for the proofs of Theorems 1 and 2. For a compact convex set $\Omega \subset \mathbb{R}^M$, we let

$$\alpha := \sup_{W \in \Omega} \|\mathbf{H}_{\mathcal{F}}(W)\| \tag{10}$$

where $\|\cdot\|$ is the spectral norm of a matrix.

**Theorem 5.** *Suppose $\mathcal{F} : \mathbb{R}^M \to \mathbb{R}$ is a nonnegative, twice continuously differentiable function and $\Omega \subset \mathbb{R}^M$ is a convex, compact set. Let $\left\{W^k\right\}_{k=1}^{\infty}$ be a sequence generate from equation 2 for a given initial guess $W^0$ and assume that $\left\{W^k\right\}_{k=1}^{\infty} \subset \Omega$. If the learning rate $\eta \in (0, 2/\alpha)$, then the following statements hold:*

*(i) $\lim_{k\to\infty} \mathcal{F}(W^k) = F^*$ for some $F^* \geq 0$;*

*(ii) $\lim_{k\to\infty} \frac{\partial \mathcal{F}}{\partial W}(W^k) = 0$ and $\lim_{k\to\infty} \|W^{k+1} - W^k\| = 0$;*

*(iii) Every cluster point $\hat{W}$ of $\left\{W^k\right\}_{k=0}^{\infty}$ satisfies $\frac{\partial \mathcal{F}}{\partial W}(\hat{W}) = 0$ .*

*Proof.* Since $\mathcal{F}$ is twice continuously differentiable, we can expand $\mathcal{F}(W^{k+1})$ at $W^k$ yields

$$\mathcal{F}(W^{k+1}) = \mathcal{F}(W^k) + \left(\frac{\partial \mathcal{F}}{\partial W}\right)^\top (W^k)\Delta W^k + r_k$$

with an error term

$$r_k = \frac{1}{2}(\Delta W^k)^\top \mathbf{H}_{\mathcal{F}}(\bar{W})\Delta W^k$$

where $\Delta W^k = W^{k+1} - W^k$ and $\bar{W}$ is a point between $W^k$ and $W^{k+1}$. By using equation equation 2, we have that

$$\frac{\partial \mathcal{F}}{\partial W}\left(W^k\right) = -\frac{1}{\eta}\Delta W^k. \tag{11}$$

Therefore,

$$\mathcal{F}(W^{k+1}) = \mathcal{F}(W^k) - \frac{1}{\eta}\|\Delta W^k\|^2 + r_k \tag{12}$$

We next estimate $r_k$. Since $\mathcal{F}$ is twice continuously differentiable, $\mathbf{H}_{\mathcal{F}}$ is continuous. As $\Omega$ is compact, $\mathbf{H}_{\mathcal{F}}$ is also bounded on $\Omega$. Moreover, since both $W^{k+1}$ and $W^k$ are in the convex set $\Omega$, we have that $\bar{W} \in \Omega$. It follows from equation 10 and compactness of $\Omega$ that

$$r_k \leq \frac{\alpha}{2}\|\Delta W^k\|^2.$$

Substituting the above inequality into the right-hand side of equation equation 12, we have that

$$\mathcal{F}(W^{k+1}) \leq \mathcal{F}(W^k) - (\frac{1}{\eta} - \frac{\alpha}{2})\|\Delta W^k\|^2. \tag{13}$$

Since $\eta \in (0, \frac{2}{\alpha})$, we have that $\frac{1}{\eta} - \frac{\alpha}{2} > 0$. The nonnegative of $\frac{1}{\eta} - \frac{\alpha}{2}$ yields

$$0 \leq \mathcal{F}(W^{k+1}) \leq \mathcal{F}(W^k), \text{ for } k = 0, 2, \ldots.$$

This guarantees that $\left\{\mathcal{F}(W^k)\right\}_{k=0}^{\infty}$ is a convergent sequence, thereby establishing Item (i).

We next prove Item (ii). For any positive integer $K$, summing inequality equation 13 over $k = 0, 1, \ldots, K$ and then we get

$$\sum_{k=0}^{K}(\frac{1}{\eta} - \frac{\alpha}{2})\|\Delta W^k\|^2 \leq \mathcal{F}(W^0) - \mathcal{F}(W^{k+1}) \leq \mathcal{F}(W^0).$$

Since $\frac{1}{\eta} - \frac{\alpha}{2}$ is positive, the above inequality implies that

$$\sum_{k=0}^{\infty}\|\Delta W^k\|^2 < \infty.$$

Therefore,

$$\lim_{k\to\infty}\|\Delta W^k\| = 0. \tag{14}$$

Equation equation 11 yields that

$$\lim_{k\to\infty}\|\frac{\partial \mathcal{F}}{\partial W}(W^k)\| = 0 \tag{15}$$

and

$$\lim_{k\to\infty}\|W^{k+1} - W^k\| = 0$$

which estimates Item (ii).

We next show Item (iii). Let $\hat{W}$ be a cluster point of $\left\{W^k\right\}_{k=0}^{\infty}$. Then there exists a subsequence $\left\{W^{k_i}\right\}_{i=0}^{\infty}$ of $\left\{W^k\right\}_{k=0}^{\infty}$ such that $\lim_{i\to\infty} W^{k_i} = \hat{W}$. The continuous of the gradient with Item (ii) implies that

$$\frac{\partial \mathcal{F}}{\partial W}(\hat{W}) = \lim_{i\to\infty}\frac{\partial \mathcal{F}}{\partial \widetilde{W}}(W^{k_i}) = 0,$$

which proves Item (iii). $\qquad\square$

**Lemma 6.** *Suppose that the activation function $\sigma$ is twice continuously differentiable and the loss function $\mathcal{L}$ is defined by equation 1, then the gradient $\frac{\partial \mathcal{L}}{\partial W}$ and hessian $\mathbf{H}_{\mathcal{L}}$ are continuous.*

*Proof.* The key point of the proof is that a polynomial of a continuous function is continuous and so is a composition of continuous function.

It follows from Lemma 3 of Xu (2025) that the componetwise of $\frac{\partial \mathcal{L}}{\partial W}$ and $\mathbf{H}_{\mathcal{L}}$ are polynomials of $\sigma, \sigma', \{\mathbf{x}_n, \mathbf{y}_n\}_{n=1}^{N}$ and the composition of $\sigma, \sigma', \sigma''$. These ensure that $\frac{\partial \mathcal{L}}{\partial W}$ and $\mathbf{H}_{\mathcal{L}}$ are continuous. $\qquad\square$

*proof of Theorem 1.* We apply Theorem 5 with $\mathcal{F} := \mathcal{L}$. Under the hypothesis that $\sigma$ is twice continuously differentiable, we have shown in Lemma 6 that the gradient $\frac{\partial \mathcal{L}}{\partial W}$ and $\mathbf{H}_{\mathcal{L}}$ are continuous. Therefore, $\mathcal{L}$ is twice continuously differentiable. It follows from the continuity of $\mathbf{H}_{\mathcal{L}}$ and the compactness of the domain $\mathcal{W}$ that $\alpha$ is finite. Thus, the hypothesis of Theorem 5 is satisfied with $\mathcal{F} := \mathcal{L}$. Theorem 1 is a direct consequence of Theorem 5. $\qquad\square$

**Lemma 7.** *Suppose that the activation function $\sigma$ is twice continuously differentiable and the loss function $\mathcal{L}_l$ is defined by equation 4 with $\{\mathbf{x}_{ln}, \mathbf{e}_{ln}\}_{n=1}^N$ being bounded, then the gradient $\frac{\partial \mathcal{L}_l}{\partial W_l}$ and hessian $\mathbf{H}_{\mathcal{L}_l}$ are continuous.*

*Proof.* Since grade $l$ in MGDL is essentially a traditional shallow neural network with the only change being the training data, which is replaced by $\{\mathbf{x}_{ln}, \mathbf{e}_{ln}\}_{n=1}^N$. We further assume that $\{\mathbf{x}_{ln}, \mathbf{e}_{ln}\}_{n=1}^N$ is bounded. This change does not affect the continuity of the gradient and Hessian. Consequently, this lemma follows directly from Lemma 6. □

*Proof of Theorem 2.* We apply Theorem 5 with $\mathcal{F} := \mathcal{L}_l$. Under the hypothesis that $\sigma$ is twice continuously differentiable, we have shown in Lemma 7 that the gradient $\frac{\partial \mathcal{L}_l}{\partial W_l}$ and $\mathbf{H}_{\mathcal{L}_l}$ are continuous. Therefore, $\mathcal{L}_l$ is twice continuously differentiable. It follows from the continuity of $\mathbf{H}_{\mathcal{L}_l}$ and the compactness of the domain $\mathcal{W}_l$ that $\alpha$ is finite. Thus, the hypothesis of Theorem 5 is satisfied with $\mathcal{F} := \mathcal{L}_l$. Theorem 2 is a direct consequence of Theorem 5. □

**Proof of Theorem 4**

We now proceed to the proof of Theorem 4, beginning with the following lemma.

**Lemma 8.** *Suppose $\mathcal{F} : \mathbb{R}^M \to \mathbb{R}$ is nonnegative and twice continuously differentiable, and $\Omega \subset \mathbb{R}^M$ is convex and compact. Let $\eta > 0$ be the learning rate and $\tau$ as defined in Theorem 4. If $\tau < 1$ for all $W \in \Omega$, then $\eta \in (0, 2/\alpha)$, where $\alpha$ is given in equation 10.*

*Proof.* For $W \in \Omega$, let $\lambda_1(W), \ldots, \lambda_M(W)$ be the eigenvalues of $\mathbf{H}_{\mathcal{F}}(W)$. By definition of $\tau$,

$$|1 - \eta\lambda_j(W)| \leq \tau, \quad \text{for all } W \in \Omega, \ j = 1, \ldots, M,$$

which implies

$$1 - \tau \leq \eta\lambda_j(W) \leq 1 + \tau.$$

Since $\tau < 1$ and $\eta > 0$, the left inequality gives

$$\lambda_j(W) \geq \frac{1 - \tau}{\eta} > 0,$$

which together with the definition of $\alpha$ yields

$$\alpha = \sup\{\lambda_j(W) : j = 1, 2, \ldots, M, W \in \Omega\} > 0.$$

The right inequality implies $\eta\alpha \leq 1 + \tau < 2$, hence $\eta \in (0, 2/\alpha)$. □

*Proof of Theorem 4.* We first prove that the linearized GD sequence $\{\tilde{W}^k\}_{k=1}^\infty$ converges. The iteration is

$$\tilde{W}^{k+1} = \left(\prod_{j=0}^{k-1} \mathbf{A}^j\right)\tilde{W}^1 - \eta\sum_{m=0}^{k-1}\left(\prod_{j=m+1}^{k-1} \mathbf{A}^j\right)u^m, \quad (16)$$

where $u^m := \frac{\partial \mathcal{F}}{\partial W}(W^m) - \mathbf{H}_{\mathcal{F}}(W^m)W^m$.

Since $\|A^k\| \leq \tau < 1$,

$$\left\|\left(\prod_{j=0}^{k-1} \mathbf{A}^j\right)\tilde{W}^1\right\| \leq \left(\prod_{j=0}^{k-1} \|\mathbf{A}^j\|\right)\|\tilde{W}^1\| \leq \tau^{k-1}\|\tilde{W}^1\| \to 0,$$

so the first term vanishes. For the second, note that $u^k$ is bounded: continuity of $\frac{\partial \mathcal{F}}{\partial W}$ and $\mathbf{H}_{\mathcal{F}}$ on compact $\Omega$ implies $\|u^k\| \leq C$. Hence,

$$\left\|\left(\prod_{j=m+1}^{k-1} \mathbf{A}^j\right)u^m\right\| \leq \tau^{k-1-m}C$$

and

$$\sum_{m=0}^{k-1} \|(\prod_{j=m+1}^{k-1} \mathbf{A}^j)u^m\| \le C \sum_{m=0}^{k-1} \tau^{k-1-m} = C\frac{1-\tau^k}{1-\tau} \le C\frac{1}{1-\tau}.$$

Thus the second term converges, and $\tilde{W}^k$ converges.

Now consider the full GD iteration:

$$W^{k+1} = \Big(\prod_{j=0}^{k-1} \mathbf{A}^j\Big)W^1 - \eta \sum_{m=0}^{k-1} \Big(\prod_{j=m+1}^{k-1} \mathbf{A}^j\Big)u^m + \sum_{m=0}^{k-1} \Big(\prod_{j=m+1}^{k-1} \mathbf{A}^j\Big)r^m, \qquad (17)$$

where $r^m = -\frac{\eta}{2}(W^{m+1} - W^m)^\top \mathbf{T}_{\mathcal{F}}(\bar{W})(W^{m+1} - W^m)$.

The first two terms behave as in the linearized case; it remains to show the last term vanishes. From Lemma 8, $\tau < 1$ when $\eta \in (0, 2/\alpha)$, so Theorem 5 implies $\|W^{k+1} - W^k\| \to 0$. Since $\mathbf{T}_{\mathcal{F}}$ is bounded on compact $\Omega$, say by $C$,

$$\|r^k\| \le \frac{\eta C}{2}\|W^{k+1} - W^k\|^2 \to 0.$$

Split the last sum into $m < N$ and $m \ge N$. For fixed $N$, the first part tends to zero as $k \to \infty$ because $\tau^{k-1-m} \to 0$. For the second, $|r^m| < \epsilon$ for $m \ge N$, so

$$\|\sum_{m=N}^{k-1} (\prod_{j=m+1}^{k-1} \mathbf{A}^j)r^m\| \le \sum_{m=N}^{k-1} \tau^{k-1-m}\|r^k\| \le \epsilon \sum_{m=N}^{k-1} \tau^{k-1-m} \le \frac{\epsilon}{1-\tau}.$$

Thus the last term vanishes, proving convergence of $W^k$. $\qquad\square$

## B    OPTIMIZATION PROBLEM FOR IMAGE RECONSTRUCTION

This appendix formulates the optimization problems for image reconstruction, covering both denoising and deblurring.

Image denoising and deblurring are classical problems in image processing, extensively studied in the literature Buades et al. (2005); Dabov et al. (2007); Micchelli et al. (2011); Fergus et al. (2006); Krishnan & Fergus (2009); Beck & Teboulle (2009); Li et al. (2015). They are commonly modeled as

$$\hat{\mathbf{f}} := \mathbf{A}\mathbf{f} + \boldsymbol{\epsilon}$$

where $\hat{\mathbf{f}} \in \mathbb{R}^{n \times n}$ is the observed corrupted image, $\mathbf{A}$ is a transform operator, $\mathbf{f} \in \mathbb{R}^{n \times n}$ is the unknown clean image, and $\boldsymbol{\epsilon}$ represents additive noise. When $\mathbf{A}$ is the identity, the task reduces to denoising Buades et al. (2005); Dabov et al. (2007); Micchelli et al. (2011), whereas if $\mathbf{A}$ is a blurring operator, it corresponds to deblurring Fergus et al. (2006); Krishnan & Fergus (2009); Beck & Teboulle (2009); Li et al. (2015). The statistical nature of the noise $\boldsymbol{\epsilon}$ depends on the specific application: for instance, Gaussian noise is commonly used for natural images Micchelli et al. (2011), while Poisson noise is typical in medical imaging Guo et al. (2022). In this work, we focus on Gaussian blurring with additive Gaussian noise.

We model a grayscale image as a function $f : \mathbb{R}^2 \to \mathbb{R}$ mapping pixel coordinates to intensity values and aim to recover $f$ from the corrupted observation $\hat{\mathbf{f}}$ using neural network-based approximators.

### B.1    PROBLEM FORMALIZATION FOR SGDL

Let $\mathcal{N}_D(\Theta; \mathbf{x})$ denote the SGDL network with parameters $\Theta := \{\mathbf{W}_j, \mathbf{b}_j\}_{j=1}^D$, and define the associated image matrix $\mathbf{N}_\Theta \in \mathbb{R}^{n \times n}$ by $[\mathbf{N}_\Theta]_{i,j} := \mathcal{N}_D(\Theta; \mathbf{x}_{i,j})$. To suppress noise and stabilize the ill-posed problem, we adopt the Rudin–Osher–Fatemi (ROF) total variation model Rudin et al. (1992), leading to the objective

$$\mathcal{G}(\Theta) := \frac{1}{2}\|\hat{\mathbf{f}} - \mathbf{A}\mathbf{N}_\Theta\|_{\mathrm{F}}^2 + \lambda\|\mathbf{B}\mathbf{N}_\Theta\|_{1,1} \qquad (18)$$

where $\mathbf{B}$ denotes the first-order difference operator, $\|\cdot\|_{\mathrm{F}}$ denotes the Frobenius norm and $\|\cdot\|_{1,1}$ denotes the entrywise $l_1$ norm. The parameter $\lambda > 0$ balances data fidelity and regularization.

Since $\|\mathbf{BN}_\Theta\|_{1,1}$ is non-differentiable term $\|\cdot\|_{1,1}$, we introduce an auxiliary variable $\mathbf{u}$ and penalize the deviation $\mathbf{u} - \mathbf{BN}\Theta$, yielding

$$\mathcal{L}(\mathbf{u}, \Theta) := \frac{1}{2}\|\hat{\mathbf{f}} - \mathbf{AN}_\Theta\|_{\mathrm{F}}^2 + \frac{\beta}{2}\|\mathbf{u} - \mathbf{BN}_\Theta\|_{\mathrm{F}}^2 + \lambda\|\mathbf{u}\|_{1,1},$$

see, Fang et al. (2024); Shen et al. (2016); Wu & Xu (2022). The corresponding optimization problem is

$$\operatorname{argmin}\left\{\mathcal{L}(\mathbf{u}, \Theta) : \mathbf{u} \in \mathbb{R}^{2n \times n}, \mathbf{W}_j \in \mathbb{R}^{d_{j-1} \times d_j}, \mathbf{b}_j \in \mathbb{R}^{d_j}, \text{ for } j \in \mathbb{N}_D\right\}. \tag{19}$$

We update $\mathbf{u}$ via the proximity operator and $\Theta$ via gradient-based optimization, yielding the iterative scheme

$$\mathbf{u}^{k+1} = \operatorname{prox}_{\alpha\lambda/\beta\|\cdot\|_{1,1}}\left(\alpha\mathbf{BN}_{\Theta^k} + (1-\alpha)\mathbf{u}^k\right), \tag{20}$$

$$\text{Using gradient-based optimizer to minimize } \mathcal{L}(\mathbf{u}^{k+1}, \Theta) \text{ and obtain } \Theta^{k+1}. \tag{21}$$

### B.2 Problem Formalization for MGDL

We now present the MGDL framework for reconstructing $f$. Grade 1 follows the same setup as SGDL, except with a smaller hidden layer. Its objective is

$$\mathcal{L}_1(\mathbf{u}, \Theta_1) := \frac{1}{2}\|\hat{\mathbf{f}} - \mathbf{AN}_{\Theta_1}\|_{\mathrm{F}}^2 + \frac{\beta}{2}\|\mathbf{u} - \mathbf{BN}_{\Theta_1}\|_{\mathrm{F}}^2 + \lambda\|\mathbf{u}\|_{1,1},$$

with optimization problem

$$\operatorname{argmin}\left\{\mathcal{L}_1(\mathbf{u}, \Theta_1) : \mathbf{u} \in \mathbb{R}^{2n \times n}, \mathbf{W}_{1j} \in \mathbb{R}^{d_{1(j-1)} \times d_{1j}}, \mathbf{b}_{1j} \in \mathbb{R}^{d_{1j}}, \text{ for } j \in \mathbb{N}_{D_1}\right\}. \tag{22}$$

For grade $l \geq 2$, we learn a new network $\mathcal{N}_{D_l}$ composed with the trained subnetworks from earlier grades. Let $[\mathbf{g}_{\Theta_1^*}]_{i,j} := \mathcal{N}_{D_1}(\Theta_1^*; \mathbf{x}_{i,j})$ and define

$$[\mathbf{g}_{\Theta_l}]_{i,j} := \epsilon_l \mathcal{N}_{D_l}(\Theta_l; \cdot) \circ \mathcal{H}_{D_{l-1}-1}(\Theta_{l-1}^*; \cdot) \circ \ldots \circ \mathcal{H}_{D_1-1}(\Theta_1^*; \mathbf{x}_{i,j}) + [\mathbf{g}_{\Theta_{l-1}^*}]_{i,j}$$

where $\epsilon_l$ is a normalization factor. The grade-$l$ loss is then

$$\mathcal{L}_l(\mathbf{u}, \Theta_l) := \frac{1}{2}\|\hat{\mathbf{f}} - \mathbf{Ag}_{\Theta_l}\|_{\mathrm{F}}^2 + \frac{\beta}{2}\|\mathbf{u} - \mathbf{Bg}_{\Theta_l}\|_{\mathrm{F}}^2 + \lambda\|\mathbf{u}\|_{1,1},$$

with minimization problem

$$\operatorname{argmin}\left\{\mathcal{L}_l(\mathbf{u}, \Theta_l) : \mathbf{u} \in \mathbb{R}^{2n \times n}, \mathbf{W}_{lj} \in \mathbb{R}^{d_{l(j-1)} \times d_{lj}}, \mathbf{b}_{lj} \in \mathbb{R}^{d_{lj}}, \text{ for } j \in \mathbb{N}_{D_l}\right\}. \tag{23}$$

Both equation 22 and equation 23 are solved using the proximity-gradient scheme equation 20–equation 21, with the loss replaced accordingly.

## C Problem setting for time series regression

We describe the problem setting for time series regression.

Given a univariate time series:

$$\{y_t \in \mathbb{R} : t = 0, 1, \ldots, N-1\},$$

the task is to predict the next $s$ future values from the past $d$ observations. Formally, we seek a regression function

$$[y_{t+1}, \ldots, y_{t+s}]^\top \approx f_\Theta([y_{t-d+1}, \ldots, y_t]^\top),$$

where $f_\Theta : \mathbb{R}^d \to \mathbb{R}^s$ is parameterized by $\Theta$.

The dataset is constructed using a sliding window of size $d$. For each index $i = 0, 1, \ldots, N-d-s$, the input-output pairs is defined as

$$\mathbf{x}_i := [y_i, y_{i+1}, \ldots, y_{i+d-1}]^\top \in \mathbb{R}^d, \quad \mathbf{z}_i := [y_{i+d}, \ldots, y_{i+d+s-1}]^\top \in \mathbb{R}^s. \tag{24}$$

**Synthetic Data**

In Section 8, the synthetic dataset is generated as

$$y_t = \sin\left(\frac{2\pi t}{50}\right) + 0.5\sin\left(\frac{2\pi t}{23}\right) + 0.3\sin\left(\frac{2\pi t}{10} + 0.5\sin\frac{2\pi t}{100}\right) + 0.01t, \quad t = 0, 1, \ldots, N-1$$

with $N := 2,000$. Input–output pairs are formed via equation 24 with $d = 64$ and $s = 1$.

The SGT architecture is

$$\text{Input} \rightarrow \text{Embedding}(64) \rightarrow \big(\text{Attention}\,(64, 1) + \text{MLP}(128)\big) \times 3 \rightarrow \text{Mean Pool} \rightarrow \text{Output}(1). \quad (25)$$

For MGT, the model is divided into three grades, each containing one transformer block.

**Financial Data**

In Section 8, we also use financial data: the daily closing prices of the S&P 500 Index from January 1, 2000, to August 22, 2025 ($N = 6,449$). Input-output pairs are formed using equation 24 with $d = 20$ and $s = 1$. Before training, the series is normalized using the training set mean $\mu$ and standard deviation $\sigma$:

$$\tilde{y}_t = \frac{y_t - \mu}{\sigma}, \quad t = 0, 1, \ldots, N-1,$$

with the same normalization applied to validation and test sets.

The SGT follows equation 25, but with six transformer blocks instead of three. For MGT, the model is split into six grades, each with one transformer block.

**Training**

For both synthetic and financial datasets, SGT and MGT are trained with the Adam optimizer and mini-batch size 128.

## D  NETWORK STRUCTURES

This Appendix details the network architectures used in the paper.

We begin with the fully connected network for SGDL, where each hidden layer has the same width. The architecture is

$$[n_{in}] \rightarrow [n_{hidden}] \times n_h \rightarrow [n_{out}], \quad (26)$$

where $n_{in}$ is the input dimension, $n_{hidden}$ the number of neurons per hidden layer, $n_h$ the number of hidden layers, and $n_{out}$ the output dimension. The parameters of the network are $(n_{in}, n_{out}, n_{hidden}, n_h)$.

In MGDL, the network expands grade by grade. Each grade uses $n_h$ hidden layers with $n_{hidden}$ neurons per layer. At grade $l$, the structure is

$$[n_{in}] \rightarrow [n_{hidden}]_F \times (l-1)n_h \rightarrow [n_{hidden}] \times n_h \rightarrow [n_{out}], \quad l = 1, 2, \ldots, L, \quad (27)$$

where $[n_{hidden}]_F \times (l-1)n_h$ are the layers trained in the first $l-1$ grades (kept fixed), and $[n_{hidden}] \times n_h$ are the layers trained at grade $l$. The network parameters are $(n_{in}, n_{out}, n_{hidden}, n_h, L)$.

For CIFAR-100, the SGDL architecture is

$$32 \times 32 \times 3 \rightarrow \big[\text{Conv}(64) \times 3 \rightarrow \text{AvgPool}\big] \times 2 \rightarrow \big[\text{Conv}(128) \times 3 \rightarrow \text{AvgPool}\big] \times 2$$
$$\rightarrow \text{Flatten} \rightarrow \text{Dense}(128) \rightarrow \text{Dense}(100). \quad (28)$$

The MGDL counterpart consists of four grades:

$$\text{G1: } 32 \times 32 \times 3 \rightarrow \text{Conv}(64) \times 3 \rightarrow \text{AvgPool} \rightarrow \text{Flatten} \rightarrow \text{Dense}(64) \rightarrow \text{Dense}(100)$$

$$\text{G2: } 16 \times 16 \times 64 \rightarrow \text{Conv}(64) \times 3 \rightarrow \text{AvgPool} \rightarrow \text{Flatten} \rightarrow \text{Dense}(64) \rightarrow \text{Dense}(100)$$

$$\text{G3: } 8 \times 8 \times 64 \rightarrow \text{Conv}(128) \times 3 \rightarrow \text{AvgPool} \rightarrow \text{Flatten} \rightarrow \text{Dense}(128) \rightarrow \text{Dense}(100) \quad (29)$$

$$\text{G4: } 4 \times 4 \times 128 \rightarrow \text{Conv}(128) \times 3 \rightarrow \text{AvgPool} \rightarrow \text{Flatten} \rightarrow \text{Dense}(128) \rightarrow \text{Dense}(100).$$

Here, $\text{Conv}(c) \times m$ denotes $m$ convolutional layers with $c$ channels, and $\text{Dense}(n)$ a fully connected layer with $n$ neurons. Convolutions use $3 \times 3$ kernels, and average pooling uses a $2 \times 2$ window.

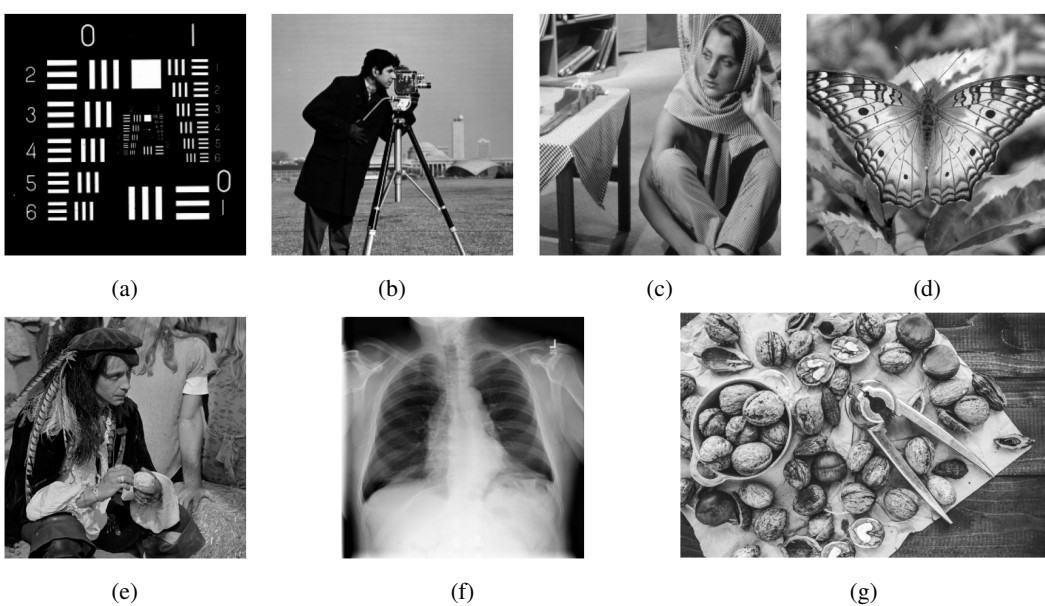

Figure 9: Clean images: (a) 'Resolution chart', (b) 'Cameraman', (c) 'Barbara', (d) 'Butterfly', (e) 'Pirate', (f) 'Chest', (g) 'Walnut'.

## E    SUPPORTING MATERIAL FOR SECTIONS 5-7

The experiments in Sections 5-7 were performed on X86_64 server with an AMD 7543 @ 2.8GHz (64 slots) supporting AVX512, 2 Nvidia Ampere A100 GPUs.

Image reconstruction quality is measured by the peak signal-to-noise ratio (PSNR), defined as

$$\text{PSNR} := 10\log_{10}\left((n \times 255^2)/\|\mathbf{v} - \hat{\mathbf{v}}\|_{\text{F}}^2\right) \tag{30}$$

where $\mathbf{v}$ is the ground-truth image, $\hat{\mathbf{v}}$ is the reconstructed image, $n$ is the number of pixels, and $\|\cdot\|_{\text{F}}$ is the Frobenius norm.

Supporting figures referenced in Section 5 include:

1. Figure 9: Clean images used in the paper.

2. Figures 10–11: Results for the 'Image Regression' in Section 5.

3. Figure 12: Noisy image used in the experiments. Figures 13–15: PSNR values during training and the denoised images produced by SGDL and MGDL. These figures correspond to the 'Image Denoising' in Section 5.

4. Figure 16: Blurred image used in the experiments. Figures 17–19: PSNR values during training and the deblurred images produced by SGDL and MGDL. These figures correspond to the 'Image Deblurring' in Section 5.

Figure 20 is the supporting figure referenced in Section 6, which presents the results for 'Image Regression'.

The supporting figures referenced in Section 7 include:

1. Figures 21-22: Results for the 'Synthetic data regression' in Section 7.

2. Figures 23-25: Results for the 'Image regression' in Section 7.

3. Figures 26-29: Results for the 'Image denoising' in Section 7.

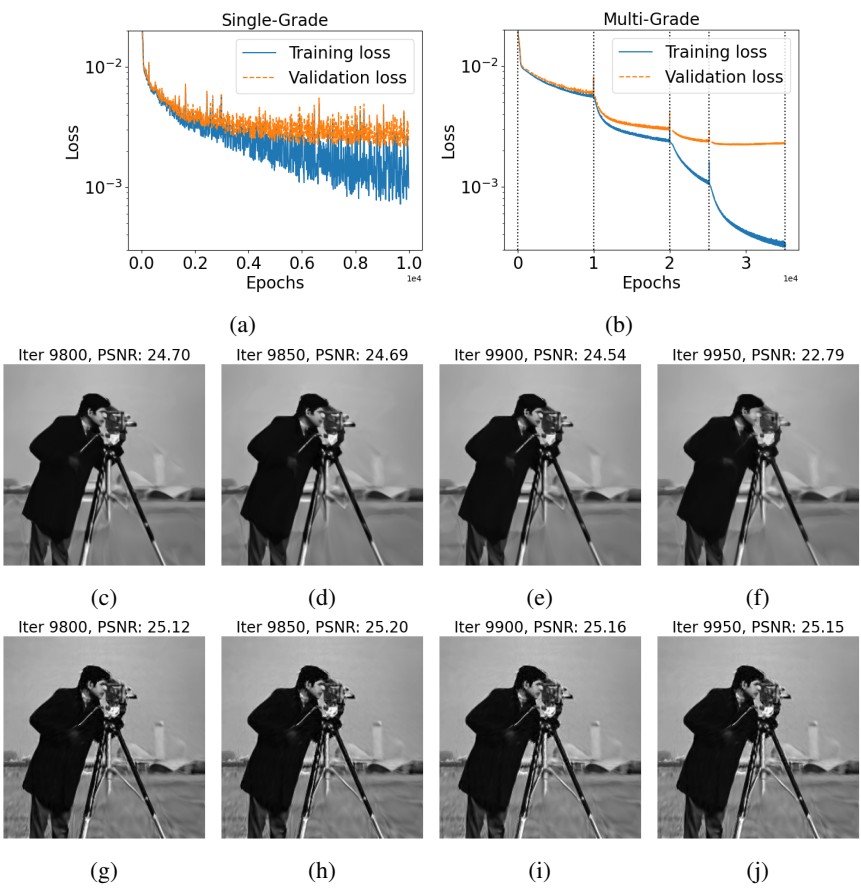

Figure 10: Comparison of SGDL and MGDL on the 'Cameraman' image. (a)–(b) show loss curves. (c)–(f) display SGDL reconstructions at iterations 9,800, 9,850, 9,900, and 9,950; (g)–(j) show the corresponding MGDL reconstructions at grade 4. PSNR values are given in the titles.

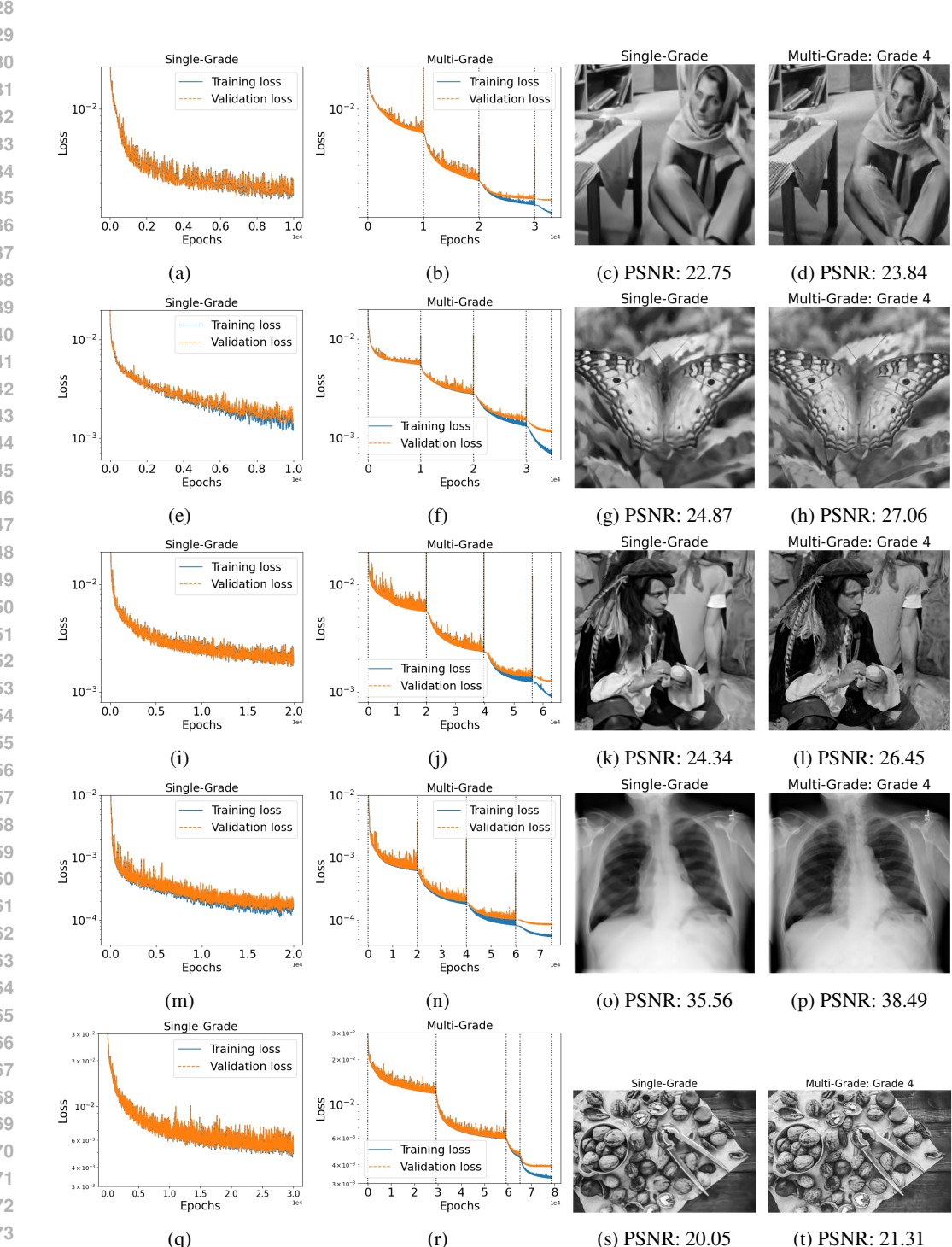

Figure 11: Comparison of SGDL and MGDL for image regression. (a)-(d) 'Barbara'; (e)-(h) 'Butterfly'; (i)-(l) 'Pirate'; (m)-(p) 'Chest'; (q)-(t) 'Walnut'.

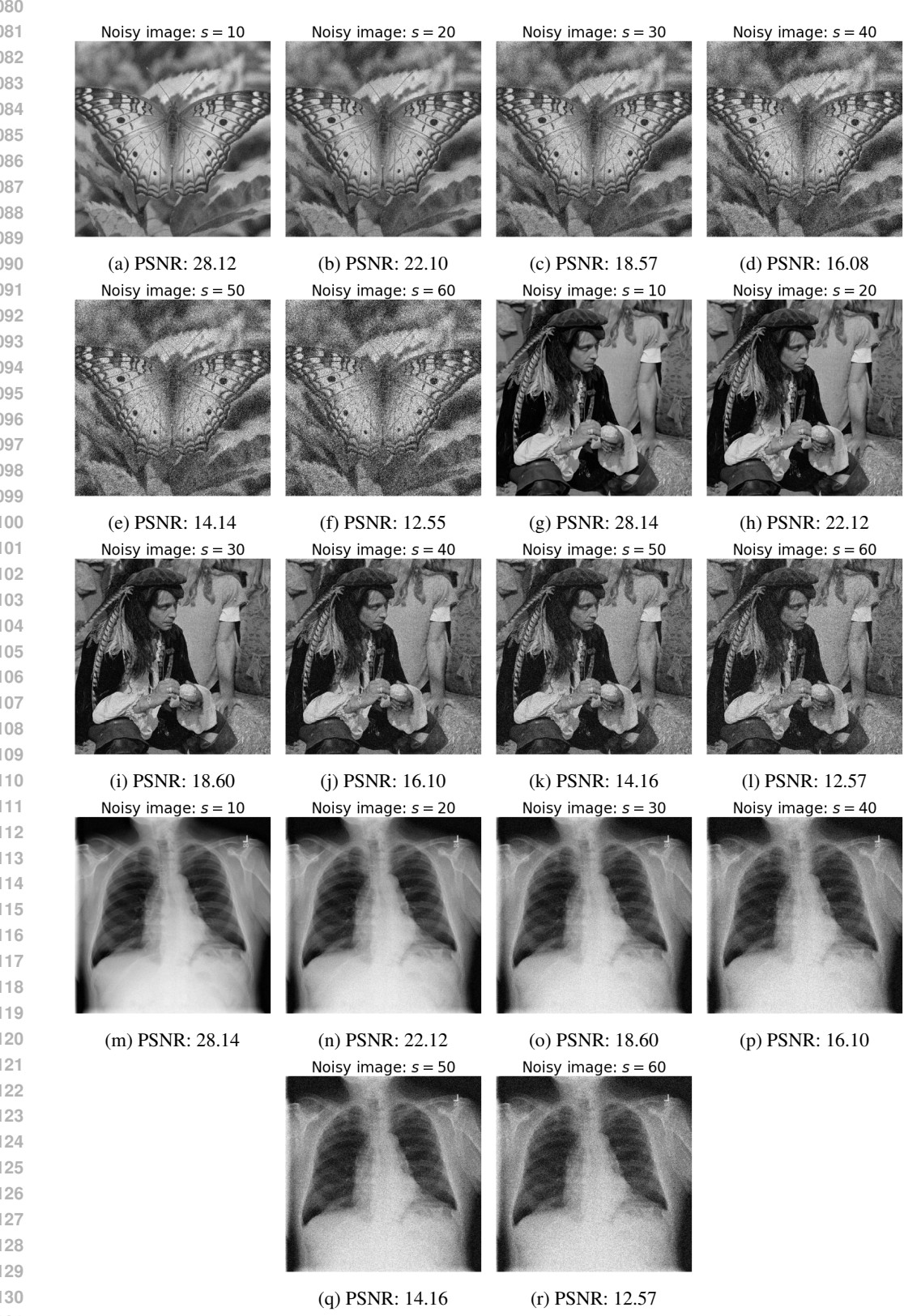

Figure 12: Noisy images at noise levels $s = 10, 20, 30, 40, 50, 60$: (a)–(f) 'Butterfly'; (g)–(l) 'Pirate'; (m)–(r) 'Chest'. PSNR values are given in the titles.

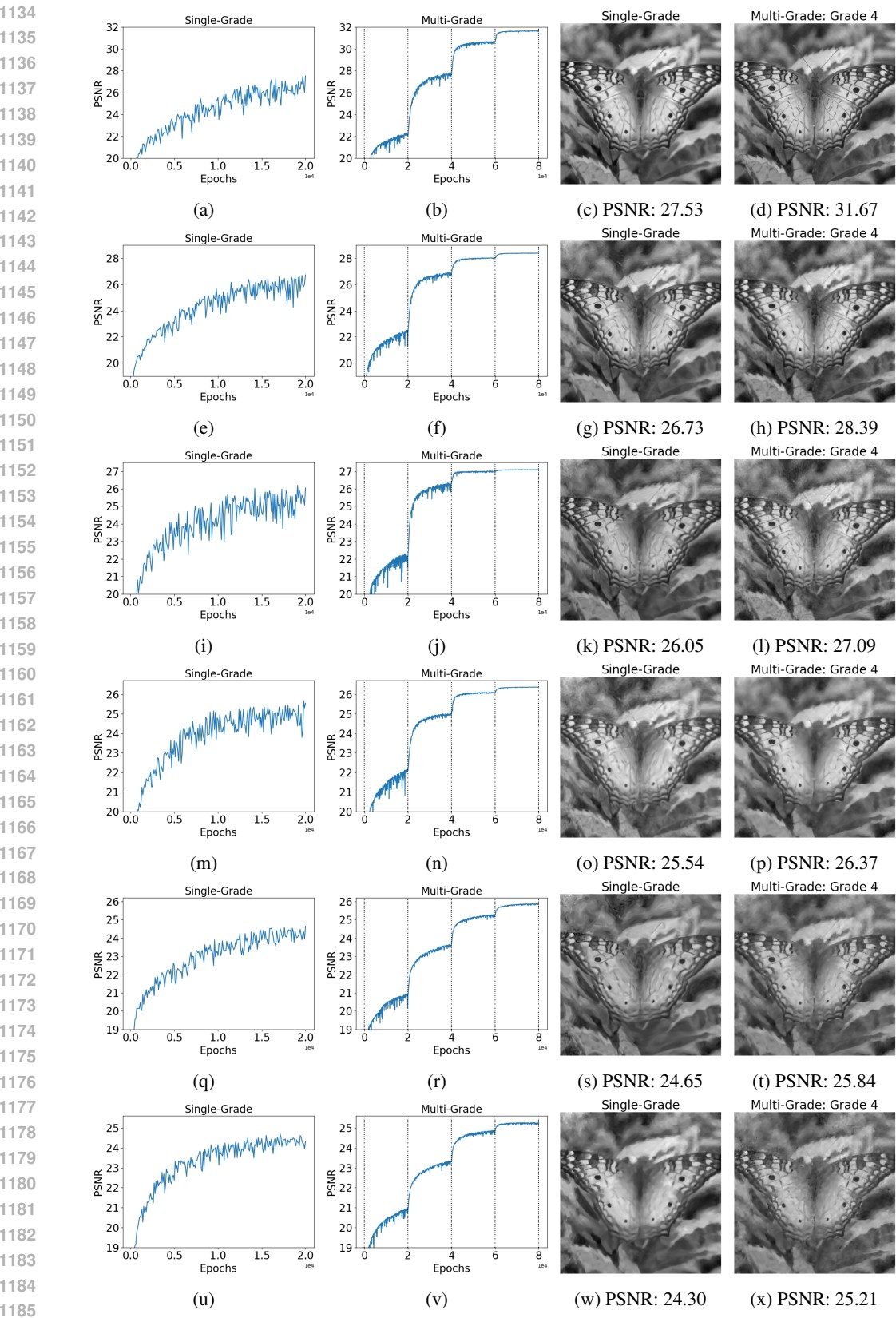

Figure 13: Comparison of SGDL and MGDL denoising results for the 'Butterfly' image. Rows 1–6 correspond to noise levels $s = 10, 20, 30, 40, 50, 60$, showing both the PSNR during training and the reconstructed images, with their PSNR values indicated in the subtitles.

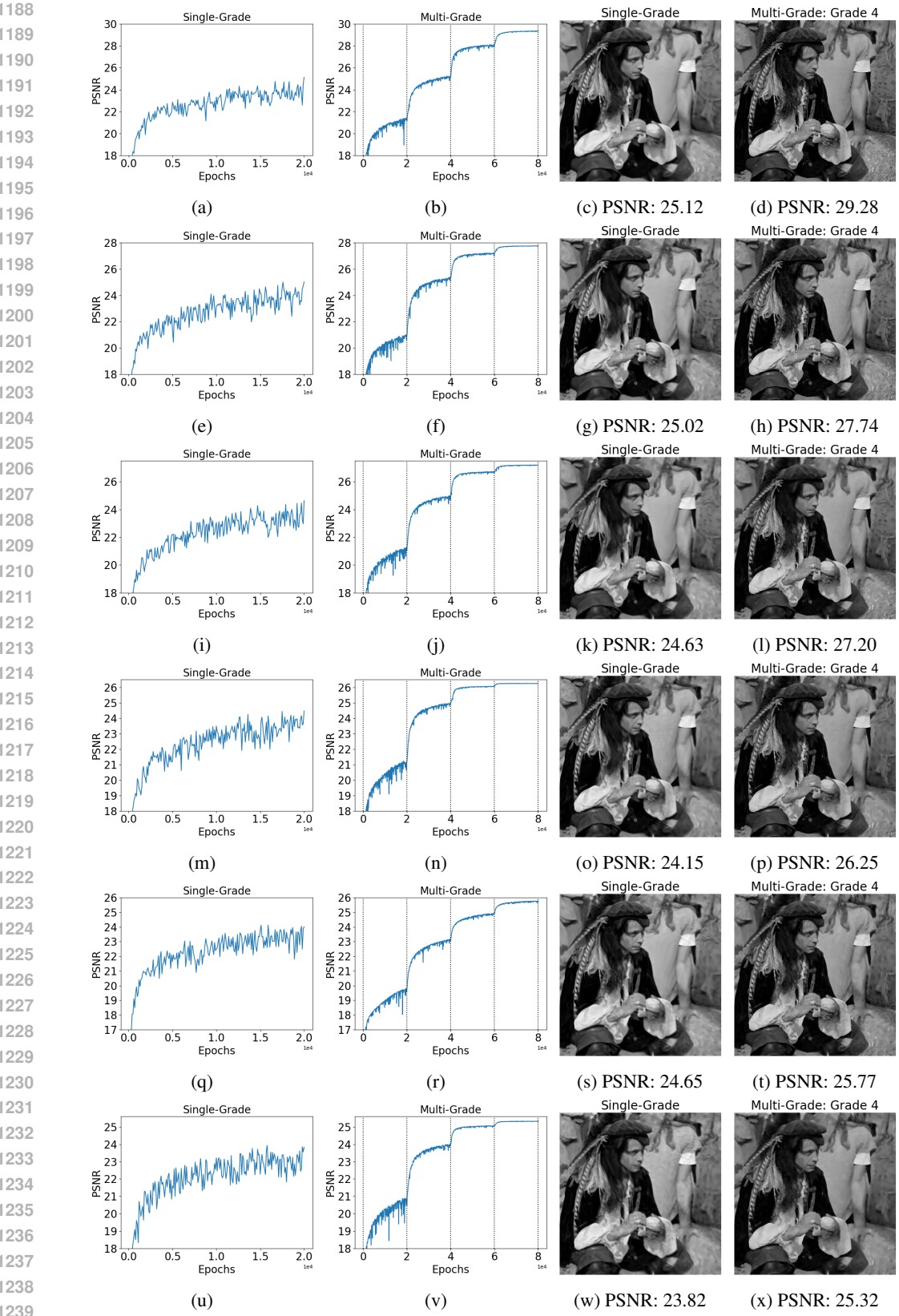

Figure 14: Comparison of SGDL and MGDL denoising results for the 'Butterfly' image. Rows 1–6 correspond to noise levels $s = 10, 20, 30, 40, 50, 60$, showing both the PSNR during training and the reconstructed images, with PSNR values indicated in the subtitles.

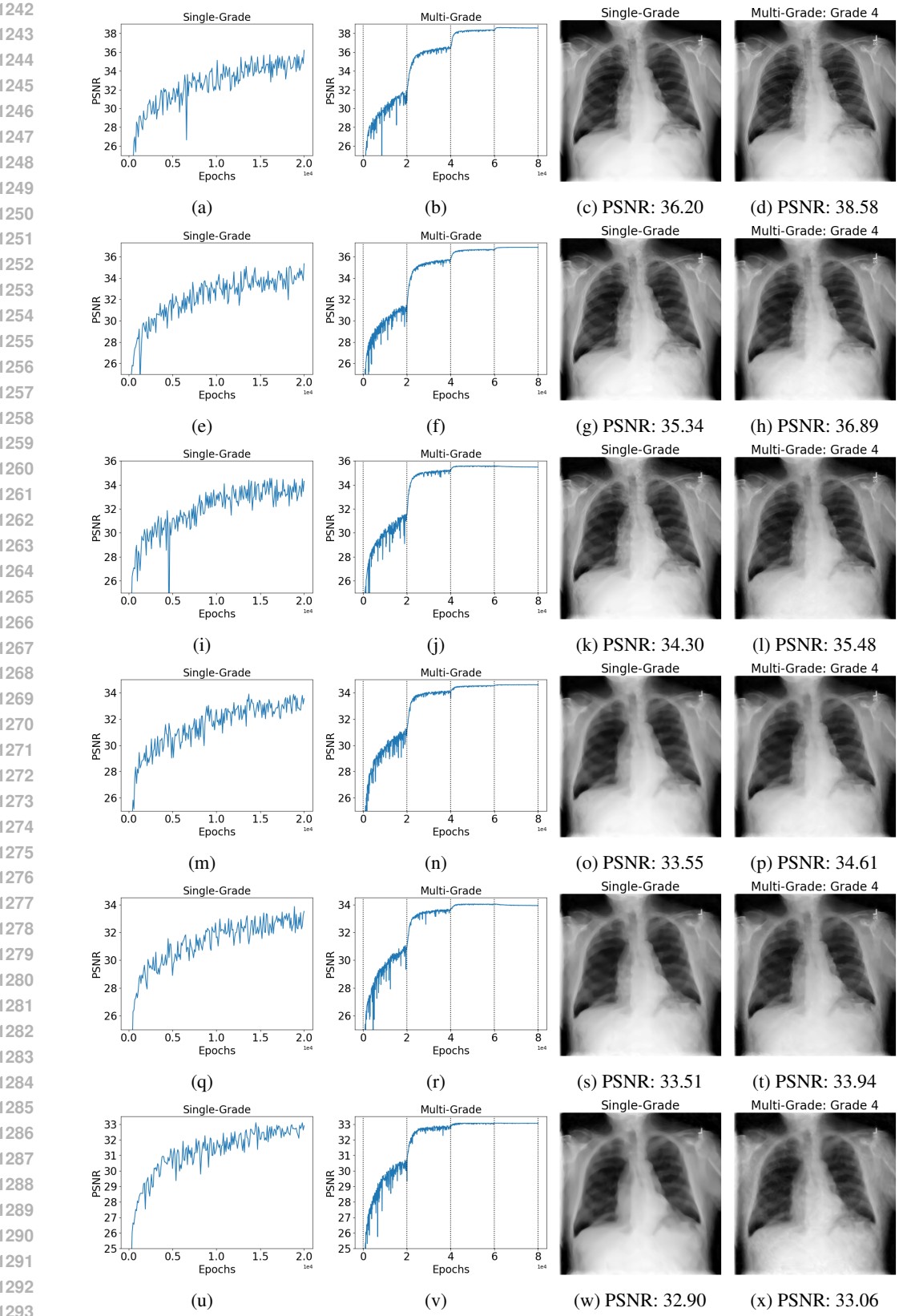

Figure 15: Comparison of SGDL and MGDL denoising results for the 'Chest' image. Rows 1–6 correspond to noise levels $s = 10, 20, 30, 40, 50, 60$, showing both the PSNR during training and the reconstructed images, with PSNR values indicated in the subtitles.

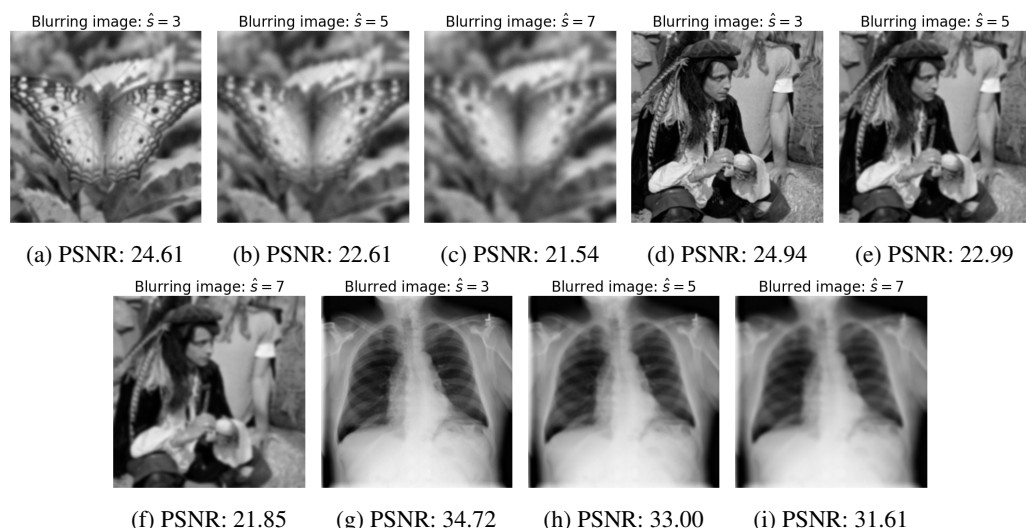

Figure 16: Images blurred with operators of standard deviations $\hat{s} = 3, 5, 7$: (a)-(c) 'Butterfly'; (d)-(f) 'Pirate'; (g)-(i) 'Chest'. The PSNR value is indicated in each title.

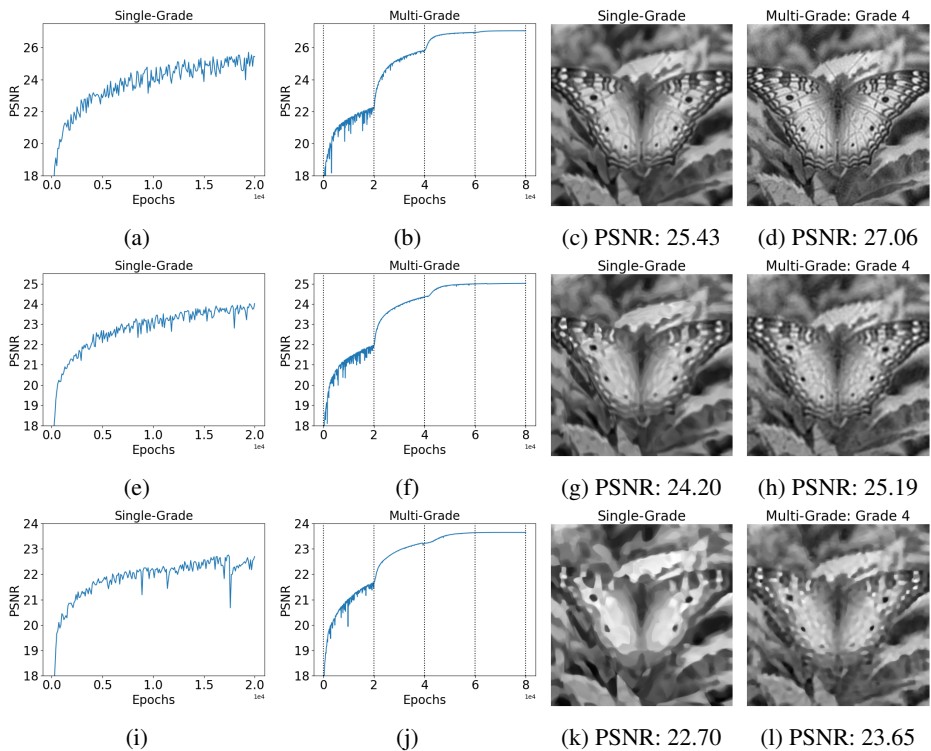

Figure 17: Comparison of SGDL and MGDL deblurring results for the 'Butterfly' image. Rows 1–3 correspond to blurring kernel standard deviations $\hat{s} = 3, 5, 7$.

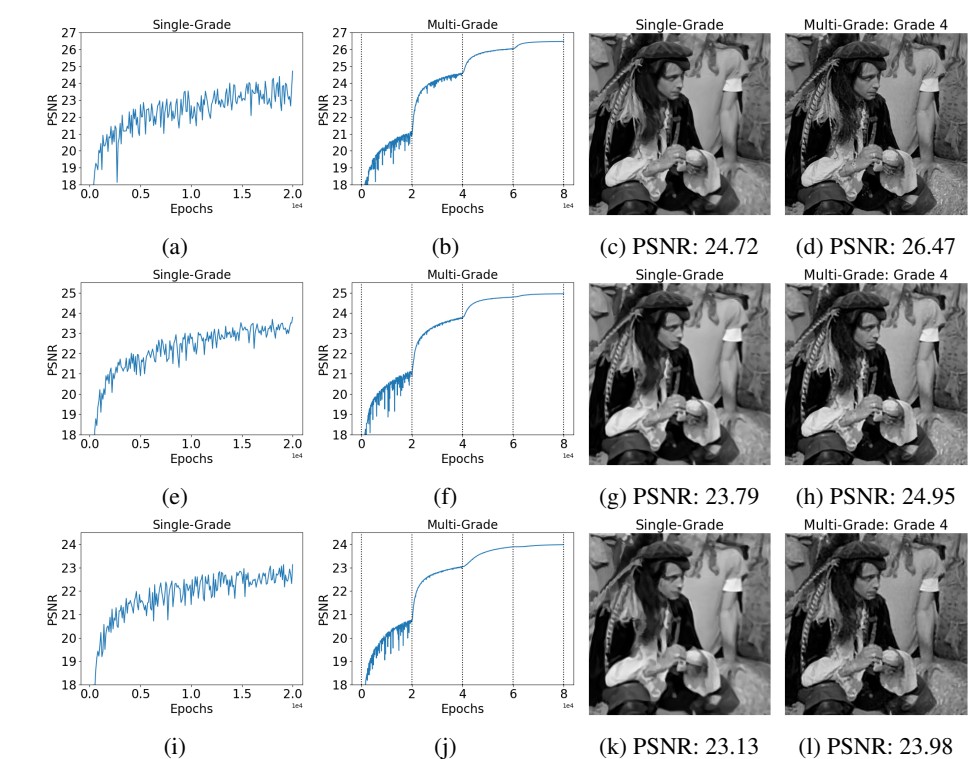

Figure 18: Comparison of SGDL and MGDL deblurring results for the 'Pirate' image. Rows 1–3 correspond to blurring kernel standard deviations $\hat{s} = 3, 5, 7$.

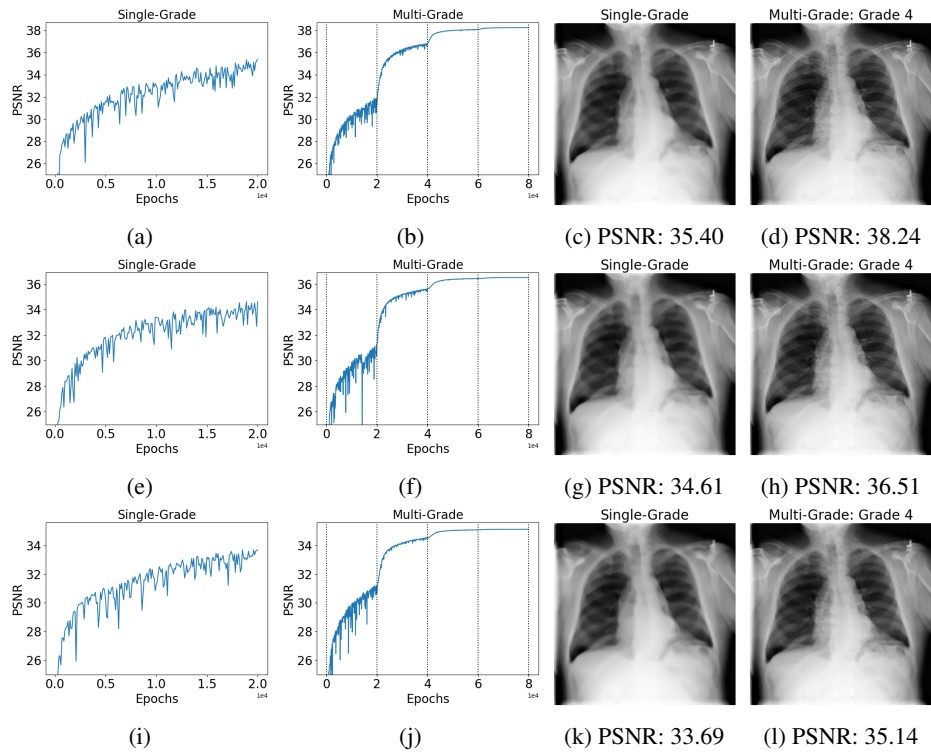

Figure 19: Comparison of SGDL and MGDL deblurring results for the 'Chest' image. Rows 1–3 correspond to blurring kernel standard deviations $\hat{s} = 3, 5, 7$.

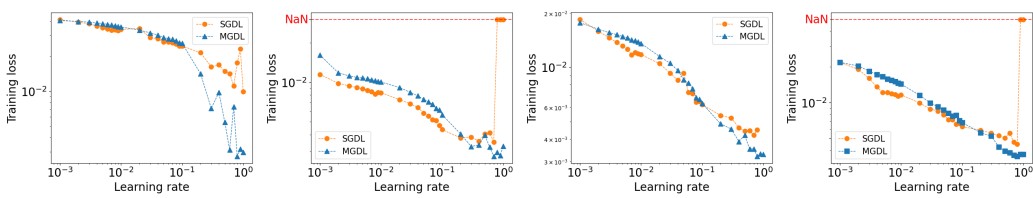

Figure 20: Impact of learning rate.

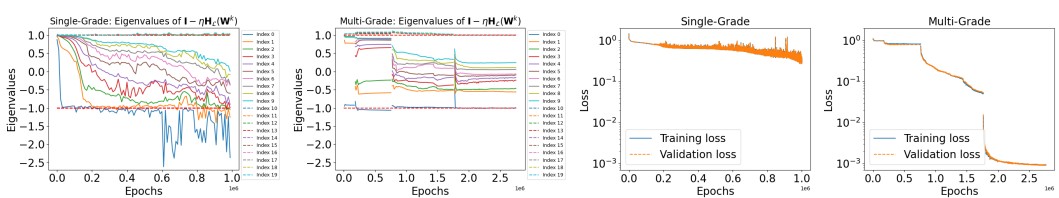

Figure 21: Training process of SGDL ($\eta = 0.005$) and MGDL (0.2) for Setting 2.

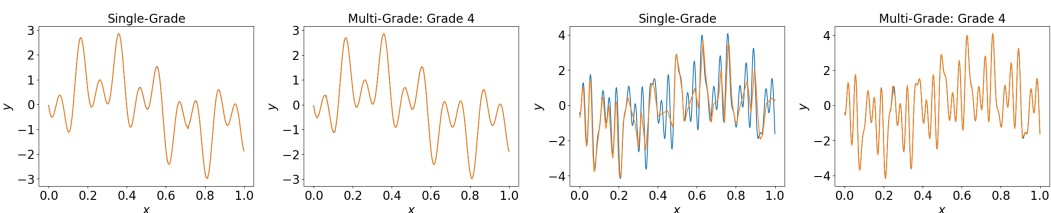

Figure 22: SGDL and MGDL predictions on synthetic data regression: Setting 1 (subfigures 1–2), Setting 2 (subfigures 3–4).

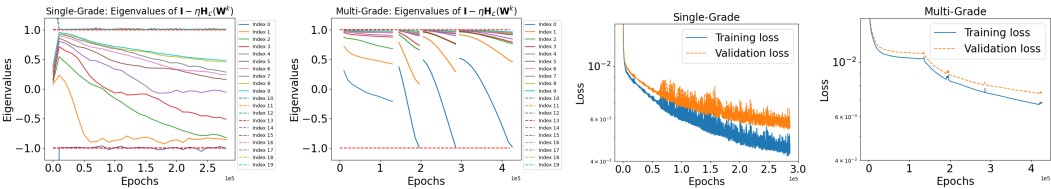

Figure 23: Training process of SGDL ($\eta = 0.1$) and MGDL ($\eta = 0.2$) for image 'Cameraman'.

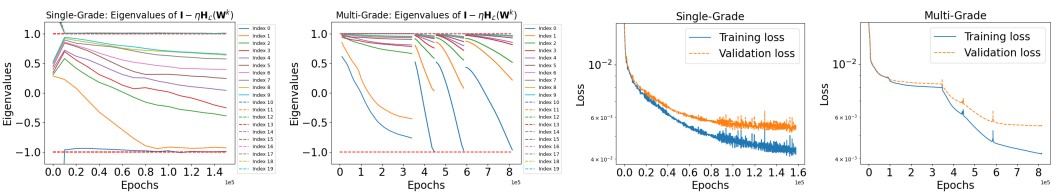

Figure 24: Training process of SGDL ($\eta = 0.08$) and MGDL ($\eta = 0.2$) for image 'Barbara'.

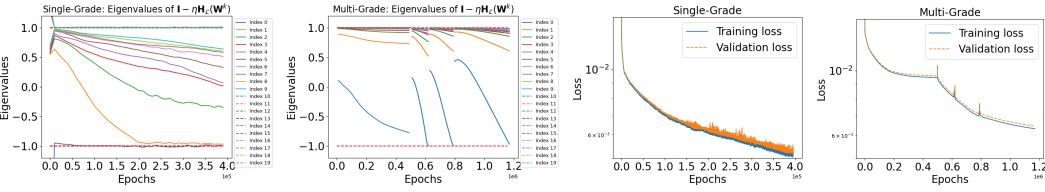

Figure 25: Training process of SGDL ($\eta = 0.05$) and MGDL ($\eta = 0.05$) for image 'Butterfly'.

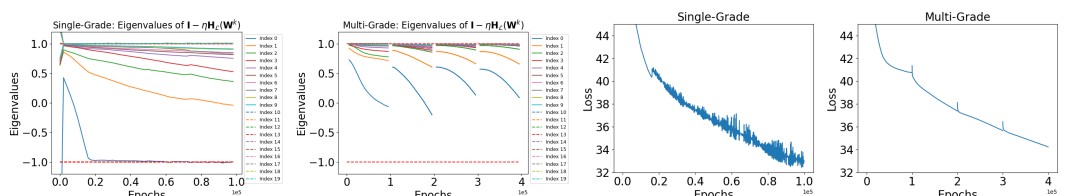

Figure 26: Training processes of SGDL and MGDL on the 'Butterfly' image (noise level 10).

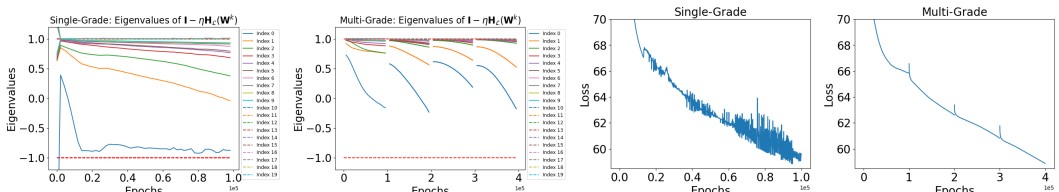

Figure 27: Training processes of SGDL and MGDL on the 'Butterfly' image (noise level 30).

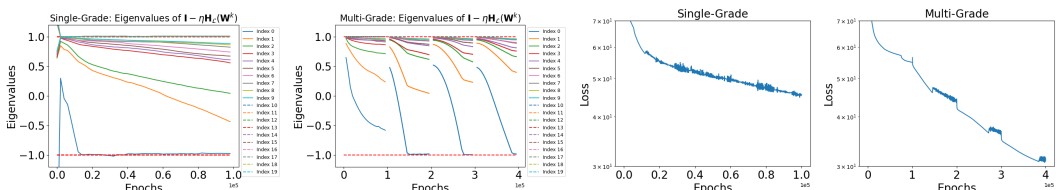

Figure 28: Training processes of SGDL and MGDL on the 'Barbara' image (noise level 10).

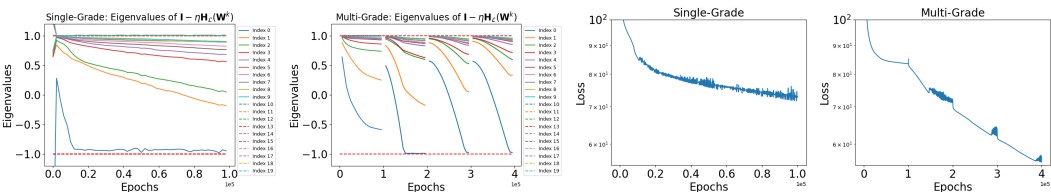

Figure 29: Training processes of SGDL and MGDL on the 'Barbara' image (noise level 30).