# SUPPLEMENTARY MATERIAL FOR:

# WHY MULTI-GRADE DEEP LEARNING OUTPERFORMS SINGLE-GRADE: THEORY AND PRACTICE

## GRADIENT AND HESSIAN MATRIX COMPUTATION

This supplementary material focuses on computing the gradient and the Hessian matrix of $\mathcal{L}$.

We define some notations. For $\ell \in \mathbb{N}_N, j \in \mathbb{N}_{D-1}$, we let

$$\mathbf{a}_{j\ell} := \mathcal{H}_j(\mathbf{x}_\ell), \quad \mathbf{a}_{0\ell} := \mathbf{x}_\ell \tag{1}$$

and the diagonal index matrix

$$[\mathbb{I}_{\mathbf{W}_j,\mathbf{b}_j,\mathbf{a}_{(j-1)\ell}}]_{ii} := \begin{cases} 1, & [\mathbf{W}_j^\top \mathbf{a}_{(j-1)\ell} + \mathbf{b}_j]_{ii} \geq 0 \\ 0, & [\mathbf{W}_j^\top \mathbf{a}_{(j-1)\ell} + \mathbf{b}_j]_{ii} < 0 \end{cases}, \quad i \in \mathbb{N}_{d_j} \tag{2}$$

and

$$\mathbf{e}_{D\ell} := \mathcal{N}_D\left(\{\mathbf{W}_j, \mathbf{b}_j\}_{j=1}^D; \mathbf{x}_\ell\right) - \mathbf{y}_\ell.$$

From the definition, $\mathbb{I}_{\mathbf{W}_j,\mathbf{b}_j,\mathbf{a}_{(j-1)\ell}}$ is a piecewise constant function with respect to $\mathbf{W}_j$ and $\mathbf{b}_j$, and therefore its gradient with respective to $W_j$ and $b_j$ are zero. Note that we do not consider the boundaries of each piece, as they have zero measure.

We will consider the network functions with one and four hidden layers, which will be used in the main paper.

## A SINGLE HIDDEN LAYER

We consider the network with a single hidden layer. In this case, $D = 2$. The network function is

$$\mathcal{N}_2\left(\{\mathbf{W}_j, \mathbf{b}_j\}_{j=1}^2; \mathbf{x}\right) := \mathbf{W}_2^\top \sigma\left(\mathbf{W}_1^\top \mathbf{x} + \mathbf{b}_1\right) + \mathbf{b}_2. \tag{3}$$

We first compute the gradient. Since we use the vectorization of the parameters in our computation, it is important to review the Kronecker product Schacke (2004); Van Loan (2000). If $\mathbf{A}$ is an $m \times n$ matrix and $\mathbf{B}$ is a $p \times q$ matrix, then the Kronecker product $\mathbf{A} \otimes \mathbf{B}$ is the $pm \times qn$ block matrix:

$$\mathbf{A} \otimes \mathbf{B} = \begin{bmatrix} a_{11}\mathbf{B} & \cdots & a_{1n}\mathbf{B} \\ \vdots & \ddots & \vdots \\ a_{m1}\mathbf{B} & \cdots & a_{mn}\mathbf{B} \end{bmatrix}.$$

Let $\mathbf{A}, \mathbf{B}$ and $\mathbf{V}$ be three matrices. The mixed Kronecker matrix-vector product can be written as:

$$(\mathbf{A} \otimes \mathbf{B})\operatorname{vec}(\mathbf{V}) = \operatorname{vec}(\mathbf{B}\mathbf{V}\mathbf{A}^T)$$

where the operator $\operatorname{vec}$ applied on a matrix denotes the vectorization of the matrix. For convenience, we let $\mathbf{z}_\ell = \mathbb{I}_{\mathbf{W}_1,\mathbf{b}_1,\mathbf{x}_\ell}\mathbf{W}_2$ for $\ell \in \mathbb{N}_N$.

**Lemma 1.** *Let $\mathcal{N}_2$ be the network function defined as* (3) *and $\mathcal{L}$ be the corresponding loss function for this network. Then, we have that*

$$\frac{\partial \mathcal{L}}{\partial W_1} = \frac{1}{N}\sum_{\ell=1}^N (\mathbf{z}_\ell \otimes \mathbf{x}_\ell)\mathbf{e}_{2\ell}, \quad \frac{\partial \mathcal{L}}{\partial b_1} = \frac{1}{N}\sum_{\ell=1}^N \mathbf{z}_\ell \mathbf{e}_{2\ell},$$

$$\frac{\partial \mathcal{L}}{\partial W_2} = \frac{1}{N}\sum_{\ell=1}^N \mathbf{a}_{1\ell}\mathbf{e}_{2\ell}, \quad \frac{\partial \mathcal{L}}{\partial b_2} = \frac{1}{N}\sum_{\ell=1}^N \mathbf{e}_{2\ell}.$$

*Proof.* Using the notations (2) and (1), we have rewrite function $\mathcal{N}_2$ as

$$\mathcal{N}_2\left(\{\mathbf{W}_j, \mathbf{b}_j\}_{j=1}^2 ; \mathbf{x}\right) = \mathbf{W}_2^\top \mathbb{I}_{\mathbf{W}_1, \mathbf{b}_1, \mathbf{x}} \mathbf{W}_1^\top \mathbf{x} + \mathbf{W}_2^\top \mathbb{I}_{\mathbf{W}_1, \mathbf{b}_1, \mathbf{x}} \mathbf{b}_1 + \mathbf{b}_2$$

$$= \left(\mathbf{z}^\top \otimes \mathbf{x}^\top\right) W_1 + \mathbf{z}^\top b_1 + b_2$$

and

$$\mathcal{N}_2\left(\{\mathbf{W}_j, \mathbf{b}_j\}_{j=1}^2 ; \mathbf{x}\right) = \mathbf{W}_2^\top \mathbf{a}_1 + \mathbf{b}_2.$$

Therefore,

$$\frac{\partial \mathcal{N}_2}{\partial W_1} = \mathbf{z} \otimes \mathbf{x}, \ \ \frac{\partial \mathcal{N}_2}{\partial b_1} = \mathbf{z}, \ \ \frac{\partial \mathcal{N}_2}{\partial W_2} = \mathbf{a}_1, \ \ \frac{\partial \mathcal{N}_2}{\partial b_2} = 1.$$

The lemma can be proved by using the chain's rule of gradient. $\square$

The hessian of $\mathcal{L}$ at $W$ is given by

$$\mathbf{H}_{\mathcal{L}}(W) := \begin{bmatrix} \frac{\partial^2 \mathcal{L}}{\partial W_1^2} & \frac{\partial^2 \mathcal{L}}{\partial b_1 \partial W_1} & \frac{\partial^2 \mathcal{L}}{\partial W_2 \partial W_1} & \frac{\partial^2 \mathcal{L}}{\partial b_2 \partial W_1} \\ \frac{\partial^2 \mathcal{L}}{\partial W_1 \partial b_1} & \frac{\partial^2 \mathcal{L}}{\partial b_1^2} & \frac{\partial^2 \mathcal{L}}{\partial W_2 \partial b_1} & \frac{\partial^2 \mathcal{L}}{\partial b_2 \partial b_1} \\ \frac{\partial^2 \mathcal{L}}{\partial W_1 \partial W_2} & \frac{\partial^2 \mathcal{L}}{\partial b_1 \partial b_2} & \frac{\partial^2 \mathcal{L}}{\partial W_2^2} & \frac{\partial^2 \mathcal{L}}{\partial b_2 \partial W_2} \\ \frac{\partial^2 \mathcal{L}}{\partial W_1 \partial b_2} & \frac{\partial^2 \mathcal{L}}{\partial b_1 \partial b_2} & \frac{\partial^2 \mathcal{L}}{\partial W_2 \partial b_2} & \frac{\partial^2 \mathcal{L}}{\partial b_2^2} \end{bmatrix}.$$

Since the Hessian matrix $\mathbf{H}_{\mathcal{L}}$ is symmetric, we compute only the upper triangular elements.

**Lemma 2.** *Let $\mathcal{N}_2$ be the network function defined as (3) and $\mathcal{L}$ be the corresponding loss function for this network. Then, we have that*

$$\frac{\partial^2 \mathcal{L}}{\partial W_1^2} = \frac{1}{N} \sum_{\ell=1}^N \left(\mathbf{z}_\ell \otimes \mathbf{x}_\ell\right) \left(\mathbf{z}_\ell \otimes \mathbf{x}_\ell\right)^\top, \quad \frac{\partial^2 \mathcal{L}}{\partial b_1 \partial W_1} = \frac{1}{N} \sum_{\ell=1}^N \left(\mathbf{z}_\ell \otimes \mathbf{x}_\ell\right) \mathbf{z}_\ell^\top$$

$$\frac{\partial^2 \mathcal{L}}{\partial W_2 \partial W_1} = \frac{1}{N} \sum_{\ell=1}^N \left(\mathbb{I}_{\mathbf{W}_1, \mathbf{b}_1, \mathbf{x}_\ell} \otimes \mathbf{x}_\ell\right) \mathbf{e}_{2\ell} + \left(\mathbf{z}_\ell \otimes \mathbf{x}_\ell\right) \mathbf{a}_{1\ell}^\top, \quad \frac{\partial^2 \mathcal{L}}{\partial b_2 \partial W_1} = \frac{1}{N} \sum_{\ell=1}^N \mathbf{z}_\ell \otimes \mathbf{x}_\ell.$$

$$\frac{\partial^2 \mathcal{L}}{\partial b_1^2} = \frac{1}{N} \sum_{\ell=1}^N \mathbf{z}_\ell \mathbf{z}_\ell^\top, \quad \frac{\partial^2 \mathcal{L}}{\partial W_2 \partial b_1} = \frac{1}{N} \sum_{\ell=1}^N \mathbb{I}_{\mathbf{W}_1, \mathbf{b}_1, \mathbf{x}_\ell} \mathbf{e}_{2\ell} + \mathbf{z}_\ell \mathbf{a}_{1\ell}^\top, \quad \frac{\partial^2 \mathcal{L}}{\partial b_2 \partial b_1} = \frac{1}{N} \sum_{\ell=1}^N \mathbf{z}_\ell$$

$$\frac{\partial^2 \mathcal{L}}{\partial W_2^2} = \frac{1}{N} \sum_{\ell=1}^N \mathbf{a}_{1\ell} \mathbf{a}_{1\ell}^\top, \quad \frac{\partial^2 \mathcal{L}}{\partial b_2 \partial W_2} = \frac{1}{N} \sum_{\ell=1}^N \mathbf{a}_{1\ell}, \quad \frac{\partial^2 \mathcal{L}}{\partial b_2^2} = 1.$$

*Proof.* We only prove $\frac{\partial^2 \mathcal{L}}{\partial W_1^2}$ and $\frac{\partial^2 \mathcal{L}}{\partial W_2 \partial W_1}$, as the proofs for the other terms follow in the same manner.

We first compute $\frac{\partial^2 \mathcal{L}}{\partial W_1^2}$. From Lemma 1, we have that

$$\frac{\partial^2 \mathcal{L}}{\partial W_1^2} = \frac{1}{N} \sum_{\ell=1}^N \frac{\partial(\mathbf{z}_\ell \otimes \mathbf{x}_\ell) \mathbf{e}_{2\ell}}{\partial W_1} = \frac{1}{N} \sum_{\ell=1}^N \frac{\partial(\mathbf{z}_\ell \otimes \mathbf{x}_\ell)}{\partial W_1} \mathbf{e}_{2\ell} + \left(\mathbf{z}_\ell \otimes \mathbf{x}_\ell\right) \left(\frac{\partial \mathbf{e}_{2\ell}}{\partial W_1}\right)^\top.$$

Since

$$\frac{\partial(\mathbf{z}_\ell \otimes \mathbf{x}_\ell)}{\partial W_1} = 0, \ \ \frac{\partial \mathbf{e}_{2\ell}}{\partial W_1} = \mathbf{z}_\ell \otimes \mathbf{x}_\ell,$$

we have that

$$\frac{\partial^2 \mathcal{L}}{\partial W_1^2} = \frac{1}{N} \sum_{\ell=1}^N \left(\mathbf{z}_\ell \otimes \mathbf{x}_\ell\right) \left(\mathbf{z}_\ell \otimes \mathbf{x}_\ell\right)^\top.$$

We now compute $\frac{\partial^2 \mathcal{L}}{\partial W_2 \partial W_1}$. Again from Lemma 1, we have that

$$\frac{\partial^2 \mathcal{L}}{\partial W_2 \partial W_1} = \frac{1}{N} \sum_{\ell=1}^N \frac{\partial(\mathbf{z}_\ell \otimes \mathbf{x}_\ell) \mathbf{e}_{2\ell}}{\partial W_2} = \frac{1}{N} \sum_{\ell=1}^N \frac{\partial(\mathbf{z}_\ell \otimes \mathbf{x}_\ell)}{\partial W_2} \mathbf{e}_{2\ell} + \left(\mathbf{z}_\ell \otimes \mathbf{x}_\ell\right) \left(\frac{\partial \mathbf{e}_{2\ell}}{\partial W_2}\right)^\top.$$

As $\mathbf{z}_\ell = \mathbb{I}_{\mathbf{W}_1, \mathbf{b}_1, \mathbf{x}_\ell} \mathbf{W}_2$, we have that

$$\frac{\partial(\mathbf{z}_\ell \otimes \mathbf{x}_\ell)}{\partial W_2} = \mathbb{I}_{\mathbf{W}_1, \mathbf{b}_1, \mathbf{x}_\ell} \otimes \mathbf{x}_\ell.$$

The definition of $\mathbf{e}_{2\ell}$ givens that $\frac{\partial \mathbf{e}_{2\ell}}{\partial W_2} = \mathbf{a}_{1\ell}$. Therefore,

$$\frac{\partial^2 \mathcal{L}}{\partial W_2 \partial W_1} = \frac{1}{N} \sum_{\ell=1}^{N} \left( \mathbb{I}_{\mathbf{W}_1, \mathbf{b}_1, \mathbf{x}_\ell} \otimes \mathbf{x}_\ell \right) \mathbf{e}_{2\ell} + \left( \mathbf{z}_\ell \otimes \mathbf{x}_\ell \right) \mathbf{a}_{1\ell}^\top.$$

$\square$

FOUR HIDDEN LAYERS

We consider a network with four hidden layers. In this case, $D = 5$. The network function is

$$\mathcal{N}_5 \left( \{\mathbf{W}_j, \mathbf{b}_j\}_{j=1}^5 ; \mathbf{x} \right) := \mathbf{W}_5^\top \sigma \left( \mathbf{W}_4^\top \sigma \left( \mathbf{W}_3^\top \sigma \left( \mathbf{W}_2^\top \sigma \left( \mathbf{W}_1^\top \mathbf{x} + \mathbf{b}_1 \right) + \mathbf{b}_2 \right) + \mathbf{b}_3 \right) + \mathbf{b}_4 \right) + \mathbf{b}_5. \tag{4}$$

The Hessian $\mathbf{H}_\mathcal{L}(W)$ is given by

$$\begin{bmatrix}
\frac{\partial^2 \mathcal{L}}{\partial W_1^2} & \frac{\partial^2 \mathcal{L}}{\partial b_1 \partial W_1} & \frac{\partial^2 \mathcal{L}}{\partial W_2 \partial W_1} & \frac{\partial^2 \mathcal{L}}{\partial b_2 \partial W_1} & \frac{\partial^2 \mathcal{L}}{\partial W_3 \partial W_1} & \frac{\partial^2 \mathcal{L}}{\partial b_3 \partial W_1} & \frac{\partial^2 \mathcal{L}}{\partial W_4 \partial W_1} & \frac{\partial^2 \mathcal{L}}{\partial b_4 \partial W_1} & \frac{\partial^2 \mathcal{L}}{\partial W_5 \partial W_1} & \frac{\partial^2 \mathcal{L}}{\partial b_5 \partial W_1} \\
\frac{\partial^2 \mathcal{L}}{\partial W_1 \partial b_1} & \frac{\partial^2 \mathcal{L}}{\partial b_1^2} & \frac{\partial^2 \mathcal{L}}{\partial W_2 \partial b_1} & \frac{\partial^2 \mathcal{L}}{\partial b_2 \partial b_1} & \frac{\partial^2 \mathcal{L}}{\partial W_3 \partial b_1} & \frac{\partial^2 \mathcal{L}}{\partial b_3 \partial b_1} & \frac{\partial^2 \mathcal{L}}{\partial W_4 \partial b_1} & \frac{\partial^2 \mathcal{L}}{\partial b_4 \partial b_1} & \frac{\partial^2 \mathcal{L}}{\partial W_5 \partial b_1} & \frac{\partial^2 \mathcal{L}}{\partial b_5 \partial b_1} \\
\frac{\partial^2 \mathcal{L}}{\partial W_1 \partial W_2} & \frac{\partial^2 \mathcal{L}}{\partial b_1 \partial W_2} & \frac{\partial^2 \mathcal{L}}{\partial W_2^2} & \frac{\partial^2 \mathcal{L}}{\partial b_2 \partial W_2} & \frac{\partial^2 \mathcal{L}}{\partial W_3 \partial W_2} & \frac{\partial^2 \mathcal{L}}{\partial b_3 \partial W_2} & \frac{\partial^2 \mathcal{L}}{\partial W_4 \partial W_2} & \frac{\partial^2 \mathcal{L}}{\partial b_4 \partial W_2} & \frac{\partial^2 \mathcal{L}}{\partial W_5 \partial W_2} & \frac{\partial^2 \mathcal{L}}{\partial b_5 \partial W_2} \\
\frac{\partial^2 \mathcal{L}}{\partial W_1 \partial b_2} & \frac{\partial^2 \mathcal{L}}{\partial b_1 \partial b_2} & \frac{\partial^2 \mathcal{L}}{\partial W_2 \partial b_2} & \frac{\partial^2 \mathcal{L}}{\partial b_2^2} & \frac{\partial^2 \mathcal{L}}{\partial W_3 \partial b_2} & \frac{\partial^2 \mathcal{L}}{\partial b_3 \partial b_2} & \frac{\partial^2 \mathcal{L}}{\partial W_4 \partial b_2} & \frac{\partial^2 \mathcal{L}}{\partial b_4 \partial b_2} & \frac{\partial^2 \mathcal{L}}{\partial W_5 \partial b_2} & \frac{\partial^2 \mathcal{L}}{\partial b_5 \partial b_2} \\
\frac{\partial^2 \mathcal{L}}{\partial W_1 \partial W_3} & \frac{\partial^2 \mathcal{L}}{\partial b_1 \partial W_3} & \frac{\partial^2 \mathcal{L}}{\partial W_2 \partial W_3} & \frac{\partial^2 \mathcal{L}}{\partial b_2 \partial W_3} & \frac{\partial^2 \mathcal{L}}{\partial W_3^2} & \frac{\partial^2 \mathcal{L}}{\partial b_3 \partial W_3} & \frac{\partial^2 \mathcal{L}}{\partial W_4 \partial W_3} & \frac{\partial^2 \mathcal{L}}{\partial b_4 \partial W_3} & \frac{\partial^2 \mathcal{L}}{\partial W_5 \partial W_3} & \frac{\partial^2 \mathcal{L}}{\partial b_5 \partial W_3} \\
\frac{\partial^2 \mathcal{L}}{\partial W_1 \partial b_3} & \frac{\partial^2 \mathcal{L}}{\partial b_1 \partial b_3} & \frac{\partial^2 \mathcal{L}}{\partial W_2 \partial b_3} & \frac{\partial^2 \mathcal{L}}{\partial b_2 \partial b_3} & \frac{\partial^2 \mathcal{L}}{\partial W_3 \partial b_3} & \frac{\partial^2 \mathcal{L}}{\partial b_3^2} & \frac{\partial^2 \mathcal{L}}{\partial W_4 \partial b_3} & \frac{\partial^2 \mathcal{L}}{\partial b_4 \partial b_3} & \frac{\partial^2 \mathcal{L}}{\partial W_5 \partial b_3} & \frac{\partial^2 \mathcal{L}}{\partial b_5 \partial b_3} \\
\frac{\partial^2 \mathcal{L}}{\partial W_1 \partial W_4} & \frac{\partial^2 \mathcal{L}}{\partial b_1 \partial W_4} & \frac{\partial^2 \mathcal{L}}{\partial W_2 \partial W_4} & \frac{\partial^2 \mathcal{L}}{\partial b_2 \partial W_4} & \frac{\partial^2 \mathcal{L}}{\partial W_3 \partial W_4} & \frac{\partial^2 \mathcal{L}}{\partial b_3 \partial W_4} & \frac{\partial^2 \mathcal{L}}{\partial W_4^2} & \frac{\partial^2 \mathcal{L}}{\partial b_4 \partial W_4} & \frac{\partial^2 \mathcal{L}}{\partial W_5 \partial W_4} & \frac{\partial^2 \mathcal{L}}{\partial b_5 \partial W_4} \\
\frac{\partial^2 \mathcal{L}}{\partial W_1 \partial b_4} & \frac{\partial^2 \mathcal{L}}{\partial b_1 \partial b_4} & \frac{\partial^2 \mathcal{L}}{\partial W_2 \partial b_4} & \frac{\partial^2 \mathcal{L}}{\partial b_2 \partial b_4} & \frac{\partial^2 \mathcal{L}}{\partial W_3 \partial b_4} & \frac{\partial^2 \mathcal{L}}{\partial b_3 \partial b_4} & \frac{\partial^2 \mathcal{L}}{\partial W_4 \partial b_4} & \frac{\partial^2 \mathcal{L}}{\partial b_4^2} & \frac{\partial^2 \mathcal{L}}{\partial W_5 \partial b_4} & \frac{\partial^2 \mathcal{L}}{\partial b_5 \partial b_4} \\
\frac{\partial^2 \mathcal{L}}{\partial W_1 \partial W_5} & \frac{\partial^2 \mathcal{L}}{\partial b_1 \partial W_5} & \frac{\partial^2 \mathcal{L}}{\partial W_2 \partial W_5} & \frac{\partial^2 \mathcal{L}}{\partial b_2 \partial W_5} & \frac{\partial^2 \mathcal{L}}{\partial W_3 \partial W_5} & \frac{\partial^2 \mathcal{L}}{\partial b_3 \partial W_5} & \frac{\partial^2 \mathcal{L}}{\partial W_4 \partial W_5} & \frac{\partial^2 \mathcal{L}}{\partial b_4 \partial W_5} & \frac{\partial^2 \mathcal{L}}{\partial W_5^2} & \frac{\partial^2 \mathcal{L}}{\partial b_5 \partial W_5} \\
\frac{\partial^2 \mathcal{L}}{\partial W_1 \partial b_5} & \frac{\partial^2 \mathcal{L}}{\partial b_1 \partial b_5} & \frac{\partial^2 \mathcal{L}}{\partial W_2 \partial b_5} & \frac{\partial^2 \mathcal{L}}{\partial b_2 \partial b_5} & \frac{\partial^2 \mathcal{L}}{\partial W_3 \partial b_5} & \frac{\partial^2 \mathcal{L}}{\partial b_3 \partial b_5} & \frac{\partial^2 \mathcal{L}}{\partial W_4 \partial b_5} & \frac{\partial^2 \mathcal{L}}{\partial b_4 \partial b_5} & \frac{\partial^2 \mathcal{L}}{\partial W_5 \partial b_5} & \frac{\partial^2 \mathcal{L}}{\partial b_5^2}
\end{bmatrix}.$$

For convenience, we let

$$\mathbf{z}_\ell := \mathbb{I}_{\mathbf{W}_1, \mathbf{b}_1, \mathbf{x}_\ell} \mathbf{W}_2 \mathbb{I}_{\mathbf{W}_2, \mathbf{b}_2, \mathbf{a}_{1\ell}} \mathbf{W}_3 \mathbb{I}_{\mathbf{W}_3, \mathbf{b}_3, \mathbf{a}_{2\ell}} \mathbf{W}_4 \mathbb{I}_{\mathbf{W}_4, \mathbf{b}_4, \mathbf{a}_{3\ell}} \mathbf{W}_5$$

and

$$\mathbf{z}_{1\ell} := \mathbb{I}_{\mathbf{W}_1, \mathbf{b}_1, \mathbf{x}_\ell} \mathbf{W}_2, \quad \mathbf{z}_{2\ell} := \mathbb{I}_{\mathbf{W}_2, \mathbf{b}_2, \mathbf{a}_{1\ell}} \mathbf{W}_3, \quad \mathbf{z}_{3\ell} := \mathbb{I}_{\mathbf{W}_3, \mathbf{b}_3, \mathbf{a}_{2\ell}} \mathbf{W}_4, \quad \mathbf{z}_{4\ell} := \mathbb{I}_{\mathbf{W}_4, \mathbf{b}_4, \mathbf{a}_{3\ell}} \mathbf{W}_5$$

and

$$\widetilde{\mathbf{z}}_{5\ell} = \mathbf{z}_{3\ell} \mathbf{z}_{4\ell}, \quad \widetilde{\mathbf{z}}_{6\ell} = \mathbf{z}_{2\ell} \mathbf{z}_{3\ell} \mathbf{z}_{4\ell}, \quad \widetilde{\mathbf{z}}_{7\ell} = \mathbf{z}_{1\ell} \mathbf{z}_{2\ell}, \quad \widetilde{\mathbf{z}}_{8\ell} = \mathbf{z}_{1\ell} \mathbf{z}_{2\ell} \mathbf{z}_{3\ell}, \quad \widetilde{\mathbf{z}}_{9\ell} = \mathbf{z}_{2\ell} \mathbf{z}_{3\ell}.$$

We first compute the gradient of $\mathcal{L}$.

$$\frac{\partial \mathcal{L}}{\partial W_1} = \frac{1}{N} \sum_{\ell=1}^{N} (\mathbf{z}_\ell \otimes \mathbf{x}_\ell) \mathbf{e}_{5\ell}, \quad \frac{\partial \mathcal{L}}{\partial b_1} = \frac{1}{N} \sum_{\ell=1}^{N} \mathbf{z}_\ell \mathbf{e}_{5\ell}, \quad \frac{\partial \mathcal{L}}{\partial W_2} = \frac{1}{N} \sum_{\ell=1}^{N} (\widetilde{\mathbf{z}}_{6\ell} \otimes \mathbf{a}_{1\ell}) \mathbf{e}_{5\ell}$$

$$\frac{\partial \mathcal{L}}{\partial b_2} = \frac{1}{N} \sum_{\ell=1}^{N} \widetilde{\mathbf{z}}_{6\ell} \mathbf{e}_{5\ell}, \quad \frac{\partial \mathcal{L}}{\partial W_3} = \frac{1}{N} \sum_{\ell=1}^{N} (\widetilde{\mathbf{z}}_{5\ell} \otimes \mathbf{a}_{2\ell}) \mathbf{e}_{5\ell}, \quad \frac{\partial \mathcal{L}}{\partial b_3} = \frac{1}{N} \sum_{\ell=1}^{N} \widetilde{\mathbf{z}}_{5\ell} \mathbf{e}_{5\ell}$$

$$\frac{\partial \mathcal{L}}{\partial W_4} = \frac{1}{N} \sum_{\ell=1}^{N} (\mathbf{z}_{4\ell} \otimes \mathbf{a}_{3\ell}) \mathbf{e}_{5\ell}, \quad \frac{\partial \mathcal{L}}{\partial b_4} = \frac{1}{N} \sum_{\ell=1}^{N} \mathbf{z}_{4\ell} \mathbf{e}_{5\ell}, \quad \frac{\partial \mathcal{L}}{\partial W_5} = \frac{1}{N} \sum_{\ell=1}^{N} \mathbf{a}_{4\ell} \mathbf{e}_{5\ell}, \quad \frac{\partial \mathcal{L}}{\partial b_5} = \frac{1}{N} \sum_{\ell=1}^{N} \mathbf{e}_{5\ell}$$

We then compute the Hessian of $\mathcal{L}$.

$$\frac{\partial^2 \mathcal{L}}{\partial W_1^2} = \frac{1}{N} \sum_{\ell=1}^{N} (\mathbf{z}_\ell \otimes \mathbf{x}_\ell) \otimes (\mathbf{z}_\ell \otimes \mathbf{x}_\ell)^\top, \quad \frac{\partial^2 \mathcal{L}}{\partial b_1 \partial W_1} = \frac{1}{N} \sum_{\ell=1}^{N} (\mathbf{z}_\ell \otimes \mathbf{x}_\ell) \otimes \mathbf{z}_\ell^\top$$

$$\frac{\partial^2 \mathcal{L}}{\partial W_2 \partial W_1} = \frac{1}{N} \sum_{\ell=1}^{N} \left( \widetilde{\mathbf{z}}_{6\ell}^\top \otimes \mathbb{I}_{\mathbf{W}_1, \mathbf{b}_1, \mathbf{x}_\ell} \otimes \mathbf{x}_\ell \right) \mathbf{e}_{5\ell} + (\mathbf{z}_\ell \otimes \mathbf{x}_\ell) \left( \widetilde{\mathbf{z}}_{6\ell} \otimes \mathbf{a}_{1\ell} \right)^\top$$

$$\frac{\partial^2 \mathcal{L}}{\partial b_2 \partial W_1} = \frac{1}{N} \sum_{\ell=1}^{N} (\mathbf{z}_\ell \otimes \mathbf{x}_\ell) \widetilde{\mathbf{z}}_{6\ell}^\top$$

$$\frac{\partial^2 \mathcal{L}}{\partial W_3 \partial W_1} = \frac{1}{N} \sum_{\ell=1}^{N} \left( \widetilde{\mathbf{z}}_{5\ell}^\top \otimes (\mathbf{z}_{1\ell} \mathbb{I}_{\mathbf{W}_2, \mathbf{b}_2, \mathbf{a}_{1\ell}}) \otimes \mathbf{x}_\ell \right) \mathbf{e}_{5\ell} + (\mathbf{z}_\ell \otimes \mathbf{x}_\ell) \left( \widetilde{\mathbf{z}}_{5\ell} \otimes \mathbf{a}_{2\ell} \right)^\top$$

$$\frac{\partial^2 \mathcal{L}}{\partial b_3 \partial W_1} = \frac{1}{N} \sum_{\ell=1}^{N} (\mathbf{z}_\ell \otimes \mathbf{x}_\ell) \widetilde{\mathbf{z}}_{5\ell}^\top$$

$$\frac{\partial^2 \mathcal{L}}{\partial W_4 \partial W_1} = \frac{1}{N} \sum_{\ell=1}^{N} \left( \mathbf{z}_{4\ell}^\top \otimes (\widetilde{\mathbf{z}}_{7\ell} \mathbb{I}_{\mathbf{W}_3, \mathbf{b}_3, \mathbf{a}_{2\ell}}) \otimes \mathbf{x}_\ell \right) \mathbf{e}_{5\ell} + (\mathbf{z}_\ell \otimes \mathbf{x}_\ell) \left( \mathbf{z}_{4\ell} \otimes \mathbf{a}_{3\ell} \right)^\top$$

$$\frac{\partial^2 \mathcal{L}}{\partial b_4 \partial W_1} = \frac{1}{N} \sum_{\ell=1}^{N} (\mathbf{z}_\ell \otimes \mathbf{x}_\ell) \mathbf{z}_{4\ell}^\top, \quad \frac{\partial^2 \mathcal{L}}{\partial W_5 \partial W_1} = \frac{1}{N} \sum_{\ell=1}^{N} ((\widetilde{\mathbf{z}}_{8\ell} \mathbb{I}_{\mathbf{W}_4, \mathbf{b}_4, \mathbf{a}_{3\ell}}) \otimes \mathbf{x}_\ell) \mathbf{e}_{5\ell} + (\mathbf{z}_\ell \otimes \mathbf{x}_\ell) \mathbf{a}_{4\ell}^\top$$

$$\frac{\partial^2 \mathcal{L}}{\partial b_5 \partial W_1} = \frac{1}{N} \sum_{\ell=1}^{N} \mathbf{z}_\ell \otimes \mathbf{x}_\ell, \quad \frac{\partial^2 \mathcal{L}}{\partial b_1^2} = \frac{1}{N} \sum_{\ell=1}^{N} \mathbf{z}_\ell \mathbf{z}_\ell^\top$$

$$\frac{\partial^2 \mathcal{L}}{\partial W_2 \partial b_1} = \frac{1}{N} \sum_{\ell=1}^{N} \left( \widetilde{\mathbf{z}}_{6\ell}^\top \otimes \mathbb{I}_{\mathbf{W}_1, \mathbf{b}_1, \mathbf{x}_\ell} \right) \mathbf{e}_{5\ell} + \mathbf{z}_\ell \left( \widetilde{\mathbf{z}}_{6\ell} \otimes \mathbf{a}_{1\ell} \right)^\top$$

$$\frac{\partial^2 \mathcal{L}}{\partial b_2 \partial b_1} = \frac{1}{N} \sum_{\ell=1}^{N} \mathbf{z}_\ell \widetilde{\mathbf{z}}_{6\ell}^\top, \quad \frac{\partial^2 \mathcal{L}}{\partial W_3 \partial b_1} = \frac{1}{N} \sum_{\ell=1}^{N} \left( \widetilde{\mathbf{z}}_{5\ell}^\top \otimes (\mathbf{z}_{1\ell} \mathbb{I}_{\mathbf{W}_2, \mathbf{b}_2, \mathbf{a}_{1\ell}}) \right) \mathbf{e}_{5\ell} + \mathbf{z}_\ell \left( \widetilde{\mathbf{z}}_{5\ell} \otimes \mathbf{a}_{2\ell} \right)^\top$$

$$\frac{\partial^2 \mathcal{L}}{\partial b_3 \partial b_1} = \frac{1}{N} \sum_{\ell=1}^{N} \mathbf{z}_\ell \widetilde{\mathbf{z}}_{5\ell}^\top, \quad \frac{\partial^2 \mathcal{L}}{\partial W_4 \partial b_1} = \frac{1}{N} \sum_{\ell=1}^{N} \left( \mathbf{z}_{4\ell}^\top \otimes (\widetilde{\mathbf{z}}_{7\ell} \mathbb{I}_{\mathbf{W}_3, \mathbf{b}_3, \mathbf{a}_{2\ell}}) \right) \mathbf{e}_{5\ell} + \mathbf{z}_\ell \left( \mathbf{z}_{4\ell} \otimes \mathbf{a}_{3\ell} \right)^\top$$

$$\frac{\partial^2 \mathcal{L}}{\partial b_4 \partial b_1} = \frac{1}{N} \sum_{\ell=1}^{N} \mathbf{z}_\ell \mathbf{z}_{4\ell}^\top, \quad \frac{\partial^2 \mathcal{L}}{\partial W_5 \partial b_1} = \frac{1}{N} \sum_{\ell=1}^{N} (\widetilde{\mathbf{z}}_{8\ell} \mathbb{I}_{\mathbf{W}_4, \mathbf{b}_4, \mathbf{a}_{3\ell}}) \mathbf{e}_{5\ell} + \mathbf{z}_\ell \mathbf{a}_{4\ell}^\top$$

$$\frac{\partial^2 \mathcal{L}}{\partial b_5 \partial b_1} = \frac{1}{N} \sum_{\ell=1}^{N} \mathbf{z}_\ell, \quad \frac{\partial^2 \mathcal{L}}{\partial W_2^2} = \frac{1}{N} \sum_{\ell=1}^{N} (\widetilde{\mathbf{z}}_{6\ell} \otimes \mathbf{a}_{1\ell}) (\widetilde{\mathbf{z}}_{6\ell} \otimes \mathbf{a}_{1\ell})^\top$$

$$\frac{\partial^2 \mathcal{L}}{\partial b_2 \partial W_2} = \frac{1}{N} \sum_{\ell=1}^{N} (\widetilde{\mathbf{z}}_{6\ell} \otimes \mathbf{a}_{1\ell}) \widetilde{\mathbf{z}}_{6\ell}^\top$$

$$\frac{\partial^2 \mathcal{L}}{\partial W_3 \partial W_2} = \frac{1}{N} \sum_{\ell=1}^{N} \left( \widetilde{\mathbf{z}}_{5\ell}^\top \otimes \mathbb{I}_{\mathbf{W}_2, \mathbf{b}_2, \mathbf{a}_{1\ell}} \otimes \mathbf{a}_{1\ell} \right) \mathbf{e}_{5\ell} + (\widetilde{\mathbf{z}}_{6\ell} \otimes \mathbf{a}_{1\ell}) (\widetilde{\mathbf{z}}_{5\ell} \otimes \mathbf{a}_{2\ell})^\top$$

$$\frac{\partial^2 \mathcal{L}}{\partial b_3 \partial W_2} = \frac{1}{N} \sum_{\ell=1}^{N} (\widetilde{\mathbf{z}}_{6\ell} \otimes \mathbf{a}_{1\ell}) \widetilde{\mathbf{z}}_{5\ell}^\top$$

$$\frac{\partial^2 \mathcal{L}}{\partial W_4 \partial W_2} = \frac{1}{N} \sum_{\ell=1}^{N} \left( \mathbf{z}_{4\ell}^\top \otimes (\mathbf{z}_{3\ell} \mathbb{I}_{\mathbf{W}_3, \mathbf{b}_3, \mathbf{a}_{2\ell}}) \otimes \mathbf{a}_{1\ell} \right) \mathbf{e}_{5\ell} + (\widetilde{\mathbf{z}}_{6\ell} \otimes \mathbf{a}_{1\ell}) \left( \mathbf{z}_{4\ell} \otimes \mathbf{a}_{3\ell} \right)^\top$$

$$\frac{\partial^2 \mathcal{L}}{\partial b_4 \partial W_2} = \frac{1}{N} \sum_{\ell=1}^{N} (\widetilde{\mathbf{z}}_{6\ell} \otimes \mathbf{a}_{1\ell}) \mathbf{z}_{4\ell}^\top$$

$$\frac{\partial^2 \mathcal{L}}{\partial W_5 \partial W_2} = \frac{1}{N} \sum_{\ell=1}^{N} ((\widetilde{\mathbf{z}}_{9\ell} \mathbb{I}_{\mathbf{W}_4, \mathbf{b}_4, \mathbf{a}_{3\ell}}) \otimes \mathbf{a}_{1\ell}) \mathbf{e}_{5\ell} + (\widetilde{\mathbf{z}}_{6\ell} \otimes \mathbf{a}_{1\ell}) \mathbf{a}_{4\ell}^\top$$

$$\frac{\partial^2 \mathcal{L}}{\partial b_5 \partial W_2} = \frac{1}{N} \sum_{\ell=1}^{N} \widetilde{\mathbf{z}}_{6\ell} \otimes \mathbf{a}_{1\ell}, \quad \frac{\partial^2 \mathcal{L}}{\partial b_2^2} = \frac{1}{N} \sum_{\ell=1}^{N} \widetilde{\mathbf{z}}_{6\ell} \widetilde{\mathbf{z}}_{6\ell}^{\top}$$

$$\frac{\partial^2 \mathcal{L}}{\partial W_3 \partial b_2} = \frac{1}{N} \sum_{\ell=1}^{N} \left( \widetilde{\mathbf{z}}_{5\ell}^{\top} \otimes \mathbb{I}_{\mathbf{W}_2, \mathbf{b}_2, \mathbf{a}_{1\ell}} \right) \mathbf{e}_{5\ell} + \widetilde{\mathbf{z}}_{6\ell} (\widetilde{\mathbf{z}}_{5\ell} \otimes \mathbf{a}_{2\ell})^{\top}, \quad \frac{\partial^2 \mathcal{L}}{\partial b_3 \partial b_2} = \frac{1}{N} \sum_{\ell=1}^{N} \widetilde{\mathbf{z}}_{6\ell} \widetilde{\mathbf{z}}_{5\ell}^{\top}$$

$$\frac{\partial^2 \mathcal{L}}{\partial W_4 \partial b_2} = \frac{1}{N} \sum_{\ell=1}^{N} \left( \mathbf{z}_{4\ell}^{\top} \otimes (\mathbf{z}_{3\ell} \mathbb{I}_{\mathbf{W}_3, \mathbf{b}_3, \mathbf{a}_{2\ell}}) \right) \mathbf{e}_{5\ell} + \widetilde{\mathbf{z}}_{6\ell} \left( \mathbf{z}_{4\ell} \otimes \mathbf{a}_{3\ell} \right)^{\top}, \quad \frac{\partial^2 \mathcal{L}}{\partial b_4 \partial b_2} = \frac{1}{N} \sum_{\ell=1}^{N} \widetilde{\mathbf{z}}_{6\ell} \mathbf{z}_{4\ell}^{\top}$$

$$\frac{\partial^2 \mathcal{L}}{\partial W_5 \partial b_2} = \frac{1}{N} \sum_{\ell=1}^{N} \left( \widetilde{\mathbf{z}}_{9\ell} \mathbb{I}_{\mathbf{W}_4, \mathbf{b}_4, \mathbf{a}_{4\ell}} \right) \mathbf{e}_{5\ell} + \widetilde{\mathbf{z}}_{6\ell} \mathbf{a}_{4\ell}^{\top}, \quad \frac{\partial^2 \mathcal{L}}{\partial b_5 \partial b_2} = \frac{1}{N} \sum_{\ell=1}^{N} \widetilde{\mathbf{z}}_{6\ell}$$

$$\frac{\partial^2 \mathcal{L}}{\partial W_3^2} = \frac{1}{N} \sum_{\ell=1}^{N} (\widetilde{\mathbf{z}}_{5\ell} \otimes \mathbf{a}_{2\ell}) \otimes (\widetilde{\mathbf{z}}_{5\ell} \otimes \mathbf{a}_{2\ell})^{\top}, \quad \frac{\partial^2 \mathcal{L}}{\partial b_3 \partial W_3} = \frac{1}{N} \sum_{\ell=1}^{N} (\widetilde{\mathbf{z}}_{5\ell} \otimes \mathbf{a}_{2\ell}) \otimes \widetilde{\mathbf{z}}_{5\ell}^{\top}$$

$$\frac{\partial^2 \mathcal{L}}{\partial W_4 \partial W_3} = \frac{1}{N} \sum_{\ell=1}^{N} \left( \mathbf{z}_{4\ell}^{\top} \otimes \mathbb{I}_{\mathbf{W}_3, \mathbf{b}_3, \mathbf{a}_{2\ell}} \otimes \mathbf{a}_{2\ell} \right) \mathbf{e}_{5\ell} + (\widetilde{\mathbf{z}}_{5\ell} \otimes \mathbf{a}_{2\ell}) \left( \mathbf{z}_{4\ell} \otimes \mathbf{a}_{3\ell} \right)^{\top}$$

$$\frac{\partial^2 \mathcal{L}}{\partial b_4 \partial W_3} = \frac{1}{N} \sum_{\ell=1}^{N} (\widetilde{\mathbf{z}}_{5\ell} \otimes \mathbf{a}_{2\ell}) \mathbf{z}_{4\ell}^{\top}, \quad \frac{\partial^2 \mathcal{L}}{\partial W_5 \partial W_3} = \frac{1}{N} \sum_{\ell=1}^{N} ((\mathbf{z}_{3\ell} \mathbb{I}_{\mathbf{W}_4, \mathbf{b}_4, \mathbf{a}_{3\ell}}) \otimes \mathbf{a}_{2\ell}) \mathbf{e}_{5\ell} + (\widetilde{\mathbf{z}}_{5\ell} \otimes \mathbf{a}_{2\ell}) \mathbf{a}_{4\ell}^{\top}$$

$$\frac{\partial^2 \mathcal{L}}{\partial b_5 \partial W_3} = \frac{1}{N} \sum_{\ell=1}^{N} \widetilde{\mathbf{z}}_{5\ell} \otimes \mathbf{a}_{2\ell}, \quad \frac{\partial^2 \mathcal{L}}{\partial b_3^2} = \frac{1}{N} \sum_{\ell=1}^{N} \widetilde{\mathbf{z}}_{5\ell} \widetilde{\mathbf{z}}_{5\ell}^{\top}$$

$$\frac{\partial^2 \mathcal{L}}{\partial W_4 \partial b_3} = \frac{1}{N} \sum_{\ell=1}^{N} \left( \mathbf{z}_{4\ell}^{\top} \otimes \mathbb{I}_{\mathbf{W}_3, \mathbf{b}_3, \mathbf{a}_{2\ell}} \right) \mathbf{e}_{5\ell} + \widetilde{\mathbf{z}}_{5\ell} (\mathbf{z}_{4\ell} \otimes \mathbf{a}_{3\ell})^{\top}, \quad \frac{\partial^2 \mathcal{L}}{\partial b_4 \partial b_3} = \frac{1}{N} \sum_{\ell=1}^{N} \widetilde{\mathbf{z}}_{5\ell} \mathbf{z}_{4\ell}^{\top}$$

$$\frac{\partial^2 \mathcal{L}}{\partial W_5 \partial b_3} = \frac{1}{N} \sum_{\ell=1}^{N} (\mathbf{z}_{3\ell} \mathbb{I}_{\mathbf{W}_4, \mathbf{b}_4, \mathbf{a}_{3\ell}}) \mathbf{e}_{5\ell} + \widetilde{\mathbf{z}}_{5\ell} \mathbf{a}_{4\ell}^{\top}, \quad \frac{\partial^2 \mathcal{L}}{\partial b_5 \partial b_3} = \frac{1}{N} \sum_{\ell=1}^{N} \widetilde{\mathbf{z}}_{5\ell}$$

$$\frac{\partial^2 \mathcal{L}}{\partial W_4^2} = \frac{1}{N} \sum_{\ell=1}^{N} (\mathbf{z}_{4\ell} \otimes \mathbf{a}_{3\ell}) (\mathbf{z}_{4\ell} \otimes \mathbf{a}_{3\ell})^{\top}, \quad \frac{\partial^2 \mathcal{L}}{\partial b_4 \partial W_4} = \frac{1}{N} \sum_{\ell=1}^{N} (\mathbf{z}_{4\ell} \otimes \mathbf{a}_{3\ell}) \mathbf{z}_{4\ell}^{\top}$$

$$\frac{\partial^2 \mathcal{L}}{\partial W_5 \partial W_4} = \frac{1}{N} \sum_{\ell=1}^{N} (\mathbb{I}_{\mathbf{W}_4, \mathbf{b}_3, \mathbf{a}_{3\ell}} \otimes \mathbf{a}_{3\ell}) \mathbf{e}_{5,\ell} + (\mathbf{z}_{4\ell} \otimes \mathbf{a}_{3\ell}) \mathbf{a}_{4\ell}^{\top}, \quad \frac{\partial^2 \mathcal{L}}{\partial b_5 \partial W_4} = \frac{1}{N} \sum_{\ell=1}^{N} \mathbf{z}_{4\ell} \otimes \mathbf{a}_{3\ell}, \quad \frac{\partial^2 \mathcal{L}}{\partial b_4^2} = \frac{1}{N} \sum_{\ell=1}^{N} \mathbf{z}_{4\ell} \mathbf{z}_{4\ell}^{\top}$$

$$\frac{\partial^2 \mathcal{L}}{\partial W_5 \partial b_4} = \frac{1}{N} \sum_{\ell=1}^{N} (\mathbb{I}_{\mathbf{W}_4, \mathbf{b}_3, \mathbf{a}_{3\ell}}) \mathbf{e}_{5,\ell} + \mathbf{z}_{4\ell} \mathbf{a}_{4\ell}^{\top}, \quad \frac{\partial^2 \mathcal{L}}{\partial b_5 \partial b_4} = \frac{1}{N} \sum_{\ell=1}^{N} \mathbf{z}_{4\ell}$$

$$\frac{\partial^2 \mathcal{L}}{\partial W_5^2} = \frac{1}{N} \sum_{\ell=1}^{N} \mathbf{a}_{4\ell} \mathbf{a}_{4\ell}^{\top}, \quad \frac{\partial^2 \mathcal{L}}{\partial b_5 \partial W_5} = \frac{1}{N} \sum_{\ell=1}^{N} \mathbf{a}_{4\ell}, \quad \frac{\partial^2 \mathcal{L}}{\partial b_5^5} = 1.$$

## HESSIAN COMPUTATION OF IMAGE DENOISING PROBLEM

To compute the hessian matrix of $\mathcal{L}$, we write $\mathcal{L}$ in the elementwise form

$$\mathcal{L}(\Theta, \mathbf{u}) = \frac{1}{2} \sum_{s,t=1}^{n} \left( \mathcal{N}_D(\Theta; \mathbf{x}_{st}) - \hat{\mathbf{f}}_{st} \right)^2 + \lambda \sum_{s,t=1}^{n} (|\mathbf{u}_{st}^1| + |\mathbf{u}_{st}^2|) +$$

$$\frac{\beta}{2} \sum_{s,t=1}^{n} \left( (\mathcal{N}_D(\Theta; \mathbf{x}_{st}) - \mathcal{N}_D(\Theta; \mathbf{x}_{s(t-1)})) - \mathbf{u}_{st}^1 \right)^2 + \left( (\mathcal{N}_D(\Theta; \mathbf{x}_{st}) - \mathcal{N}_D(\Theta; \mathbf{x}_{(s-1)t})) - \mathbf{u}_{st}^2 \right)^2 \tag{5}$$

where we set $\mathcal{N}_D(\Theta; \mathbf{x}_{s0}) = \mathcal{N}_D(\Theta; \mathbf{x}_{s1})$ and $\mathcal{N}_D(\Theta; \mathbf{x}_{0t}) = \mathcal{N}_D(\Theta; \mathbf{x}_{1t})$ for $s, t \in \mathbb{N}_n$.

### ONE HIDDEN LAYER

We consider the case with one hidden layer and the network function is given by (3). Let

$$\mathbf{z}_{st} = \mathbb{I}_{\mathbf{W}_1, \mathbf{b}_1, \mathbf{x}_{st}} \mathbf{W}_2, \quad \mathbf{a}_{1st} = \sigma(\mathbf{W}_1^\top \mathbf{x}_{st} + \mathbf{b}_1), \quad \mathbf{e}_{1st} = \mathcal{N}_2\left(\{\mathbf{W}_j, \mathbf{b}_j\}_{j=1}^2 ; \mathbf{x}_{st}\right) - \hat{\mathbf{f}}_{st}$$

$$\mathbf{e}_{2st} = \mathcal{N}_2\left(\{\mathbf{W}_j, \mathbf{b}_j\}_{j=1}^2 ; \mathbf{x}_{st}\right) - \mathcal{N}_2\left(\{\mathbf{W}_j, \mathbf{b}_j\}_{j=1}^2 ; \mathbf{x}_{s(t-1)}\right) - \mathbf{u}_{st}^1$$

$$\mathbf{e}_{3st} = \mathcal{N}_2\left(\{\mathbf{W}_j, \mathbf{b}_j\}_{j=1}^2 ; \mathbf{x}_{st}\right) - \mathcal{N}_2\left(\{\mathbf{W}_j, \mathbf{b}_j\}_{j=1}^2 ; \mathbf{x}_{(s-1)t}\right) - \mathbf{u}_{st}^2.$$

We first consider the gradient of $\mathcal{L}$.

$$\frac{\partial \mathcal{L}}{\partial W_1} = \sum_{s,t=1}^n (\mathbf{z}_{st} \otimes \mathbf{x}_{st})\mathbf{e}_{1st} + \left((\mathbf{z}_{st} \otimes \mathbf{x}_{st} - \mathbf{z}_{s(t-1)} \otimes \mathbf{x}_{s(t-1)})\mathbf{e}_{2st} + \right.$$

$$(\mathbf{z}_{st} \otimes \mathbf{x}_{st} - \mathbf{z}_{(s-1)t} \otimes \mathbf{x}_{(s-1)t})\mathbf{e}_{3st})$$

$$\frac{\partial \mathcal{L}}{\partial b_1} = \sum_{s,t=1}^n \mathbf{z}_{st}\mathbf{e}_{1st} + \beta\left((\mathbf{z}_{st} - \mathbf{z}_{s(t-1)})\mathbf{e}_{2st} + (\mathbf{z}_{st} - \mathbf{z}_{(s-1)t})\mathbf{e}_{3st}\right)$$

$$\frac{\partial \mathcal{L}}{\partial W_2} = \sum_{s,t=1}^n \mathbf{a}_{1st}\mathbf{e}_{1st} + \beta\left((\mathbf{a}_{1st} - \mathbf{a}_{1s(t-1)})\mathbf{e}_{2st} + (\mathbf{a}_{1st} - \mathbf{a}_{1(s-1)t})\mathbf{e}_{3st}\right), \quad \frac{\partial \mathcal{L}}{\partial b_2} = \sum_{s,t=1}^n \mathbf{e}_{1st}.$$

We then consider Hessian matrix of $\mathcal{L}$.

$$\frac{\partial^2 \mathcal{L}}{\partial W_1^2} = \sum_{s,t=1}^n (\mathbf{z}_{st} \otimes \mathbf{x}_{st})(\mathbf{z}_{st} \otimes \mathbf{x}_{st})^\top$$

$$+ \beta\left((\mathbf{z}_{st} \otimes \mathbf{x}_{st} - \mathbf{z}_{s(t-1)} \otimes \mathbf{x}_{s(t-1)})(\mathbf{z}_{st} \otimes \mathbf{x}_{st} - \mathbf{z}_{s(t-1)} \otimes \mathbf{x}_{s(t-1)})^\top\right.$$

$$+ (\mathbf{z}_{st} \otimes \mathbf{x}_{st} - \mathbf{z}_{(s-1)t} \otimes \mathbf{x}_{(s-1)t})(\mathbf{z}_{st} \otimes \mathbf{x}_{st} - \mathbf{z}_{(s-1)t} \otimes \mathbf{x}_{(s-1)t})^\top)$$

$$\frac{\partial^2 \mathcal{L}}{\partial b_1 \partial W_1} = \sum_{s,t=1}^n (\mathbf{z}_{st} \otimes \mathbf{x}_{st})\mathbf{z}_{st}^\top + \beta\left((\mathbf{z}_{st} \otimes \mathbf{x}_{st} - \mathbf{z}_{s(t-1)} \otimes \mathbf{x}_{s(t-1)})(\mathbf{z}_{st} - \mathbf{z}_{s(t-1)})^\top\right.$$

$$+ (\mathbf{z}_{st} \otimes \mathbf{x}_{st} - \mathbf{z}_{(s-1)t} \otimes \mathbf{x}_{(s-1)t})(\mathbf{z}_{st} - \mathbf{z}_{(s-1)t})^\top)$$

$$\frac{\partial^2 \mathcal{L}}{\partial W_2 \partial W_1} = \sum_{s,t=1}^n \left(\mathbb{I}_{\mathbf{W}_1, \mathbf{b}_1, \mathbf{x}_{st}} \otimes \mathbf{x}_{st}\right)\mathbf{e}_{1st} + (\mathbf{z}_{st} \otimes \mathbf{x}_{st})\mathbf{a}_{1st}^\top$$

$$+ \beta\left((\mathbb{I}_{\mathbf{W}_1, \mathbf{b}_1, \mathbf{x}_{st}} \otimes \mathbf{x}_{st} - \mathbb{I}_{\mathbf{W}_1, \mathbf{b}_1, \mathbf{x}_{s(t-1)}} \otimes \mathbf{x}_{s(t-1)})\mathbf{e}_{2st}\right.$$

$$+ (\mathbf{z}_{st} \otimes \mathbf{x}_{st} - \mathbf{z}_{s(t-1)} \otimes \mathbf{x}_{s(t-1)})(\mathbf{a}_{1st} - \mathbf{a}_{1s(t-1)})^\top$$

$$+ (\mathbb{I}_{\mathbf{W}_1, \mathbf{b}_1, \mathbf{x}_{st}} \otimes \mathbf{x}_{st} - \mathbb{I}_{\mathbf{W}_1, \mathbf{b}_1, \mathbf{x}_{(s-1)t}} \otimes \mathbf{x}_{(s-1)t})\mathbf{e}_{3st}$$

$$+ (\mathbf{z}_{st} \otimes \mathbf{x}_{st} - \mathbf{z}_{(s-1)t} \otimes \mathbf{x}_{(s-1)t})(\mathbf{a}_{1st} - \mathbf{a}_{1(s-1)t})^\top)$$

$$\frac{\partial^2 \mathcal{L}}{\partial b_2 \partial W_1} = \sum_{s,t=1}^n (\mathbf{z}_{st} \otimes \mathbf{x}_{st})$$

$$\frac{\partial^2 \mathcal{L}}{\partial b_1^2} = \sum_{s,t=1}^n \mathbf{z}_{st}\mathbf{z}_{st}^\top + \beta\left((\mathbf{z}_{st} - \mathbf{z}_{s(t-1)})(\mathbf{z}_{st} - \mathbf{z}_{s(t-1)})^\top + (\mathbf{z}_{st} - \mathbf{z}_{(s-1)t})(\mathbf{z}_{st} - \mathbf{z}_{(s-1)t})^\top\right)$$

$$\frac{\partial^2 \mathcal{L}}{\partial W_2 \partial b_1} = \sum_{s,t=1}^n \mathbb{I}_{\mathbf{W}_1, \mathbf{b}_1, \mathbf{x}_{st}}\mathbf{e}_{1st} + \mathbf{z}_{st}\mathbf{a}_{1st}^\top$$

$$+ \beta\left((\mathbb{I}_{\mathbf{W}_1, \mathbf{b}_1, \mathbf{x}_{st}} - \mathbb{I}_{\mathbf{W}_1, \mathbf{b}_1, \mathbf{x}_{s(t-1)}})\mathbf{e}_{2st} + (\mathbf{z}_{st} - \mathbf{z}_{s(t-1)})(\mathbf{a}_{1st} - \mathbf{a}_{1s(t-1)})^\top\right.$$

$$+ (\mathbb{I}_{\mathbf{W}_1, \mathbf{b}_1, \mathbf{x}_{st}} - \mathbb{I}_{\mathbf{W}_1, \mathbf{b}_1, \mathbf{x}_{(s-1)t}})\mathbf{e}_{3st} + (\mathbf{z}_{st} - \mathbf{z}_{(s-1)t})(\mathbf{a}_{1st} - \mathbf{a}_{1(s-1)t})^\top)$$

$$\frac{\partial^2 \mathcal{L}}{\partial b_2 \partial b_1} = \sum_{s,t=1}^{n} \mathbf{z}_{st}$$

$$\frac{\partial^2 \mathcal{L}}{\partial W_2^2} = \sum_{s,t=1}^{n} \mathbf{a}_{1st}\mathbf{a}_{1st}^{\top} + \beta\big((\mathbf{a}_{1st} - \mathbf{a}_{1s(t-1)})(\mathbf{a}_{1st} - \mathbf{a}_{1s(t-1)})^{\top} +$$

$$(\mathbf{a}_{1st} - \mathbf{a}_{1(s-1)t})(\mathbf{a}_{1st} - \mathbf{a}_{1(s-1)t})^{\top}\big)$$

$$\frac{\partial^2 \mathcal{L}}{\partial b_2 \partial W_2} = \sum_{s,t=1}^{n} \mathbf{a}_{1st}, \quad \frac{\partial^2 \mathcal{L}}{\partial b_2^2} = n^2.$$

FOUR HIDDEN LAYERS

We consider the case with four hidden layers and the network function is given by (4).

Let
$$\mathbf{z}_{st} := \mathbb{I}_{\mathbf{W}_1,\mathbf{b}_1,\mathbf{x}_{st}} \mathbf{W}_2 \mathbb{I}_{\mathbf{W}_2,\mathbf{b}_2,\mathbf{a}_{1st}} \mathbf{W}_3 \mathbb{I}_{\mathbf{W}_3,\mathbf{b}_3,\mathbf{a}_{2st}} \mathbf{W}_4 \mathbb{I}_{\mathbf{W}_4,\mathbf{b}_4,\mathbf{a}_{3st}} \mathbf{W}_5$$
and
$$\mathbf{z}_{1st} := \mathbb{I}_{\mathbf{W}_1,\mathbf{b}_1,\mathbf{x}_{st}} \mathbf{W}_2, \quad \mathbf{z}_{2st} := \mathbb{I}_{\mathbf{W}_2,\mathbf{b}_2,\mathbf{a}_{1st}} \mathbf{W}_3, \quad \mathbf{z}_{3st} := \mathbb{I}_{\mathbf{W}_3,\mathbf{b}_3,\mathbf{a}_{2st}} \mathbf{W}_4, \quad \mathbf{z}_{4st} := \mathbb{I}_{\mathbf{W}_4,\mathbf{b}_4,\mathbf{a}_{3st}} \mathbf{W}_5$$
and
$$\widetilde{\mathbf{z}}_{5st} = \mathbf{z}_{3st}\mathbf{z}_{4st}, \quad \widetilde{\mathbf{z}}_{6st} = \mathbf{z}_{2st}\mathbf{z}_{3st}\mathbf{z}_{4st}, \quad \widetilde{\mathbf{z}}_{7st} = \mathbf{z}_{1st}\mathbf{z}_{2st}, \quad \widetilde{\mathbf{z}}_{8st} = \mathbf{z}_{1st}\mathbf{z}_{2st}\mathbf{z}_{3st}, \quad \widetilde{\mathbf{z}}_{9st} = \mathbf{z}_{2st}\mathbf{z}_{3st}.$$

$$\mathbf{e}_{1st} = \mathcal{N}_4\left(\{\mathbf{W}_j, \mathbf{b}_j\}_{j=1}^{4}; \mathbf{x}_{st}\right) - \hat{\mathbf{f}}_{st}$$

$$\mathbf{e}_{2st} = \mathcal{N}_4\left(\{\mathbf{W}_j, \mathbf{b}_j\}_{j=1}^{4}; \mathbf{x}_{st}\right) - \mathcal{N}_4\left(\{\mathbf{W}_j, \mathbf{b}_j\}_{j=1}^{4}; \mathbf{x}_{s(t-1)}\right) - \mathbf{u}_{st}^{1}$$

$$\mathbf{e}_{3st} = \mathcal{N}_4\left(\{\mathbf{W}_j, \mathbf{b}_j\}_{j=1}^{4}; \mathbf{x}_{st}\right) - \mathcal{N}_4\left(\{\mathbf{W}_j, \mathbf{b}_j\}_{j=1}^{4}; \mathbf{x}_{(s-1)t}\right) - \mathbf{u}_{st}^{2}.$$

We first compute the gradient of $\mathcal{L}$.

$$\frac{\partial \mathcal{L}}{\partial W_1} = \sum_{s,t=1}^{n} (\mathbf{z}_{st} \otimes \mathbf{x}_{st})\mathbf{e}_{1st} + \beta\big((\mathbf{z}_{st} \otimes \mathbf{x}_{st} - \mathbf{z}_{s(t-1)} \otimes \mathbf{x}_{s(t-1)})\mathbf{e}_{2st} +$$

$$(\mathbf{z}_{st} \otimes \mathbf{x}_{st} - \mathbf{z}_{(s-1)t} \otimes \mathbf{x}_{(s-1)t})\mathbf{e}_{3st}\big)$$

$$\frac{\partial \mathcal{L}}{\partial b_1} = \sum_{s,t=1}^{n} \mathbf{z}_{st}\mathbf{e}_{1st} + \beta\left((\mathbf{z}_{st} - \mathbf{z}_{s(t-1)})\mathbf{e}_{2st} + (\mathbf{z}_{st} - \mathbf{z}_{(s-1)t})\mathbf{e}_{3st}\right)$$

$$\frac{\partial \mathcal{L}}{\partial W_2} = \sum_{s,t=1}^{n} (\widetilde{\mathbf{z}}_{6st} \otimes \mathbf{a}_{1st})\mathbf{e}_{1st} + \beta\big((\widetilde{\mathbf{z}}_{6st} \otimes \mathbf{a}_{1st} - \widetilde{\mathbf{z}}_{6s(t-1)} \otimes \mathbf{a}_{1s(t-1)})\mathbf{e}_{2st} +$$

$$(\widetilde{\mathbf{z}}_{6st} \otimes \mathbf{a}_{1st} - \widetilde{\mathbf{z}}_{6(s-1)t} \otimes \mathbf{a}_{1(s-1)t})\mathbf{e}_{3st}\big)$$

$$\frac{\partial \mathcal{L}}{\partial b_2} = \sum_{s,t=1}^{n} \widetilde{\mathbf{z}}_{6st}\mathbf{e}_{1st} + \beta\big((\widetilde{\mathbf{z}}_{6st} - \widetilde{\mathbf{z}}_{6s(t-1)})\mathbf{e}_{2st} + (\widetilde{\mathbf{z}}_{6st} - \widetilde{\mathbf{z}}_{6(s-1)t})\mathbf{e}_{3st}\big)$$

$$\frac{\partial \mathcal{L}}{\partial W_3} = \sum_{s,t=1}^{n} (\widetilde{\mathbf{z}}_{5st} \otimes \mathbf{a}_{2st})\mathbf{e}_{1st} + \beta\big((\widetilde{\mathbf{z}}_{5st} \otimes \mathbf{a}_{2st} - \widetilde{\mathbf{z}}_{5s(t-1)} \otimes \mathbf{a}_{2s(t-1)})\mathbf{e}_{2st} +$$

$$(\widetilde{\mathbf{z}}_{5st} \otimes \mathbf{a}_{2st} - \widetilde{\mathbf{z}}_{5(s-1)t} \otimes \mathbf{a}_{2(s-1)t})\mathbf{e}_{3st}\big)$$

$$\frac{\partial \mathcal{L}}{\partial b_3} = \sum_{s,t=1}^{n} \widetilde{\mathbf{z}}_{5st}\mathbf{e}_{1st} + \beta\big((\widetilde{\mathbf{z}}_{5st} - \widetilde{\mathbf{z}}_{5s(t-1)})\mathbf{e}_{2st} + (\widetilde{\mathbf{z}}_{5st} - \widetilde{\mathbf{z}}_{5(s-1)t})\mathbf{e}_{3st}\big)$$

$$\frac{\partial \mathcal{L}}{\partial W_4} = \sum_{s,t=1}^{n} (\widetilde{\mathbf{z}}_{4st} \otimes \mathbf{a}_{3st})\mathbf{e}_{1st} + \beta\big((\widetilde{\mathbf{z}}_{4st} \otimes \mathbf{a}_{3st} - \widetilde{\mathbf{z}}_{4s(t-1)} \otimes \mathbf{a}_{3s(t-1)})\mathbf{e}_{2st} +$$

$$(\widetilde{\mathbf{z}}_{4st} \otimes \mathbf{a}_{3st} - \widetilde{\mathbf{z}}_{4(s-1)t} \otimes \mathbf{a}_{3(s-1)t})\mathbf{e}_{3st}\big)$$

$$\frac{\partial \mathcal{L}}{\partial b_4} = \sum_{s,t=1}^{n} \widetilde{\mathbf{z}}_{4st}\mathbf{e}_{1st} + \beta\big((\widetilde{\mathbf{z}}_{4st} - \widetilde{\mathbf{z}}_{4s(t-1)})\mathbf{e}_{2st} + (\widetilde{\mathbf{z}}_{4st} - \widetilde{\mathbf{z}}_{4(s-1)t})\mathbf{e}_{3st}\big)$$

$$\frac{\partial \mathcal{L}}{\partial W_5} = \sum_{s,t=1}^{n} \mathbf{a}_{4st}\mathbf{e}_{1st} + \beta\big((\mathbf{a}_{4st} - \mathbf{a}_{4s(t-1)})\mathbf{e}_{2st} + (\mathbf{a}_{4st} - \mathbf{a}_{4(s-1)t})\mathbf{e}_{3st}\big), \quad \frac{\partial \mathcal{L}}{\partial b_5} = \sum_{s,t=1}^{n} \mathbf{e}_{1st}$$

We then consider Hessian matrix of $\mathcal{L}$.

$$\frac{\partial^2 \mathcal{L}}{\partial W_1^2} = \sum_{s,t=1}^{n} (\mathbf{z}_{st} \otimes \mathbf{x}_{st})(\mathbf{z}_{st} \otimes \mathbf{x}_{st})^\top$$

$$+ \beta\big((\mathbf{z}_{st} \otimes \mathbf{x}_{st} - \mathbf{z}_{s(t-1)} \otimes \mathbf{x}_{s(t-1)})(\mathbf{z}_{st} \otimes \mathbf{x}_{st} - \mathbf{z}_{s(t-1)} \otimes \mathbf{x}_{s(t-1)})^\top$$

$$+ (\mathbf{z}_{st} \otimes \mathbf{x}_{st} - \mathbf{z}_{(s-1)t} \otimes \mathbf{x}_{(s-1)t})(\mathbf{z}_{st} \otimes \mathbf{x}_{st} - \mathbf{z}_{(s-1)t} \otimes \mathbf{x}_{(s-1)t})^\top\big)$$

$$\frac{\partial^2 \mathcal{L}}{\partial b_1 \partial W_1} = \sum_{s,t=1}^{n} (\mathbf{z}_{st} \otimes \mathbf{x}_{st})\mathbf{z}_{st}^\top + \beta\big((\mathbf{z}_{st} \otimes \mathbf{x}_{st} - \mathbf{z}_{s(t-1)} \otimes \mathbf{x}_{s(t-1)})(\mathbf{z}_{st} - \mathbf{z}_{s(t-1)})^\top$$

$$+ (\mathbf{z}_{st} \otimes \mathbf{x}_{st} - \mathbf{z}_{(s-1)t} \otimes \mathbf{x}_{(s-1)t})(\mathbf{z}_{st} - \mathbf{z}_{(s-1)t})^\top\big)$$

$$\frac{\partial^2 \mathcal{L}}{\partial W_2 \partial W_1} = \sum_{s,t=1}^{n} \left(\widetilde{\mathbf{z}}_{6st}^\top \otimes \mathbb{I}_{\mathbf{W}_1,\mathbf{b}_1,\mathbf{x}_{st}} \otimes \mathbf{x}_{st}\right)\mathbf{e}_{1st} + (\mathbf{z}_{st} \otimes \mathbf{x}_{st})(\widetilde{\mathbf{z}}_{6st} \otimes \mathbf{a}_{1st})^\top$$

$$+ \beta\big((\widetilde{\mathbf{z}}_{6st}^\top \otimes \mathbb{I}_{\mathbf{W}_1,\mathbf{b}_1,\mathbf{x}_{st}} \otimes \mathbf{x}_{st} - \widetilde{\mathbf{z}}_{6s(t-1)}^\top \otimes \mathbb{I}_{\mathbf{W}_1,\mathbf{b}_1,\mathbf{x}_{s(t-1)}} \otimes \mathbf{x}_{s(t-1)})\mathbf{e}_{2st}$$

$$+ (\mathbf{z}_{st} \otimes \mathbf{x}_{st} - \mathbf{z}_{s(t-1)} \otimes \mathbf{x}_{s(t-1)})(\widetilde{\mathbf{z}}_{6st} \otimes \mathbf{a}_{1st} - \widetilde{\mathbf{z}}_{6s(t-1)} \otimes \mathbf{a}_{1s(t-1)})^\top$$

$$+ (\widetilde{\mathbf{z}}_{6st}^\top \otimes \mathbb{I}_{\mathbf{W}_1,\mathbf{b}_1,\mathbf{x}_{st}} \otimes \mathbf{x}_{st} - \widetilde{\mathbf{z}}_{6(s-1)t}^\top \otimes \mathbb{I}_{\mathbf{W}_1,\mathbf{b}_1,\mathbf{x}_{(s-1)t}} \otimes \mathbf{x}_{(s-1)t})\mathbf{e}_{3st}$$

$$+ (\mathbf{z}_{st} \otimes \mathbf{x}_{st} - \mathbf{z}_{(s-1)t} \otimes \mathbf{x}_{(s-1)t})(\widetilde{\mathbf{z}}_{6st} \otimes \mathbf{a}_{1st} - \widetilde{\mathbf{z}}_{6(s-1)t} \otimes \mathbf{a}_{1(s-1)t})^\top\big)$$

$$\frac{\partial^2 \mathcal{L}}{\partial b_2 \partial W_1} = \sum_{s,t=1}^{n} (\mathbf{z}_{st} \otimes \mathbf{x}_{st})\widetilde{\mathbf{z}}_{6st}^\top + \beta\big((\mathbf{z}_{st} \otimes \mathbf{x}_{st} - \mathbf{z}_{s(t-1)} \otimes \mathbf{x}_{s(t-1)})(\widetilde{\mathbf{z}}_{6st} - \widetilde{\mathbf{z}}_{6s(t-1)})^\top$$

$$+ (\mathbf{z}_{st} \otimes \mathbf{x}_{st} - \mathbf{z}_{(s-1)t} \otimes \mathbf{x}_{(s-1)t})(\widetilde{\mathbf{z}}_{6st} - \widetilde{\mathbf{z}}_{6(s-1)t})^\top\big)$$

$$\frac{\partial^2 \mathcal{L}}{\partial W_3 \partial W_1} = \sum_{s,t=1}^{n} \left(\widetilde{\mathbf{z}}_{5st}^\top \otimes (\mathbf{z}_{1st}\mathbb{I}_{\mathbf{W}_2,\mathbf{b}_2,\mathbf{a}_{1st}}) \otimes \mathbf{x}_{st}\right)\mathbf{e}_{1st} + (\mathbf{z}_{st} \otimes \mathbf{x}_{st})(\widetilde{\mathbf{z}}_{5st} \otimes \mathbf{a}_{2st})^\top$$

$$+ \beta\big((\widetilde{\mathbf{z}}_{5st}^\top \otimes (\mathbf{z}_{1st}\mathbb{I}_{\mathbf{W}_2,\mathbf{b}_2,\mathbf{a}_{1st}}) \otimes \mathbf{x}_{st} - \widetilde{\mathbf{z}}_{5s(t-1)}^\top \otimes (\mathbf{z}_{1s(t-1)}\mathbb{I}_{\mathbf{W}_2,\mathbf{b}_2,\mathbf{a}_{1s(t-1)}}) \otimes \mathbf{x}_{s(t-1)})\mathbf{e}_{2st}$$

$$+ (\mathbf{z}_{st} \otimes \mathbf{x}_{st} - \mathbf{z}_{s(t-1)} \otimes \mathbf{x}_{s(t-1)})(\widetilde{\mathbf{z}}_{5st} \otimes \mathbf{a}_{2st} - \widetilde{\mathbf{z}}_{5s(t-1)} \otimes \mathbf{a}_{2s(t-1)})^\top$$

$$+ (\widetilde{\mathbf{z}}_{5st}^\top \otimes (\mathbf{z}_{1st}\mathbb{I}_{\mathbf{W}_2,\mathbf{b}_2,\mathbf{a}_{1st}}) \otimes \mathbf{x}_{st} - \widetilde{\mathbf{z}}_{5(s-1)t}^\top \otimes (\mathbf{z}_{1(s-1)t}\mathbb{I}_{\mathbf{W}_2,\mathbf{b}_2,\mathbf{a}_{1(s-1)t}}) \otimes \mathbf{x}_{(s-1)t})\mathbf{e}_{3st}$$

$$+ (\mathbf{z}_{st} \otimes \mathbf{x}_{st} - \mathbf{z}_{(s-1)t} \otimes \mathbf{x}_{(s-1)t})(\widetilde{\mathbf{z}}_{5st} \otimes \mathbf{a}_{2st} - \widetilde{\mathbf{z}}_{5(s-1)t} \otimes \mathbf{a}_{2(s-1)t})^\top\big)$$

$$\frac{\partial^2 \mathcal{L}}{\partial b_3 \partial W_1} = \sum_{s,t=1}^{n} (\mathbf{z}_{st} \otimes \mathbf{x}_{st})\widetilde{\mathbf{z}}_{5st}^\top + \beta\big((\mathbf{z}_{st} \otimes \mathbf{x}_{st} - \mathbf{z}_{s(t-1)} \otimes \mathbf{x}_{s(t-1)})(\widetilde{\mathbf{z}}_{5st} - \widetilde{\mathbf{z}}_{5s(t-1)})^\top$$

$$+ (\mathbf{z}_{st} \otimes \mathbf{x}_{st} - \mathbf{z}_{(s-1)t} \otimes \mathbf{x}_{(s-1)t})(\widetilde{\mathbf{z}}_{5st} - \widetilde{\mathbf{z}}_{5(s-1)t})^\top\big)$$

$$\frac{\partial^2 \mathcal{L}}{\partial W_4 \partial W_1} = \sum_{s,t=1}^{n} \left( \mathbf{z}_{4st}^{\top} \otimes (\widetilde{\mathbf{z}}_{7st} \mathbb{I}_{\mathbf{W}_3, \mathbf{b}_3, \mathbf{a}_{2st}}) \otimes \mathbf{x}_{st} \right) \mathbf{e}_{1st} + (\mathbf{z}_{st} \otimes \mathbf{x}_{st})(\mathbf{z}_{4st} \otimes \mathbf{a}_{3st})^{\top}$$

$$+ \beta \big( (\mathbf{z}_{4st}^{\top} \otimes (\widetilde{\mathbf{z}}_{7st} \mathbb{I}_{\mathbf{W}_3, \mathbf{b}_3, \mathbf{a}_{2st}}) \otimes \mathbf{x}_{st} - \mathbf{z}_{4s(t-1)}^{\top} \otimes (\widetilde{\mathbf{z}}_{7s(t-1)} \mathbb{I}_{\mathbf{W}_3, \mathbf{b}_3, \mathbf{a}_{2s(t-1)}}) \otimes \mathbf{x}_{s(t-1)}) \mathbf{e}_{2st}$$

$$+ (\mathbf{z}_{st} \otimes \mathbf{x}_{st} - \mathbf{z}_{s(t-1)} \otimes \mathbf{x}_{s(t-1)})(\mathbf{z}_{4st} \otimes \mathbf{a}_{3st} - \mathbf{z}_{4s(t-1)} \otimes \mathbf{a}_{3s(t-1)})^{\top}$$

$$+ (\mathbf{z}_{4st}^{\top} \otimes (\widetilde{\mathbf{z}}_{7st} \mathbb{I}_{\mathbf{W}_3, \mathbf{b}_3, \mathbf{a}_{2st}}) \otimes \mathbf{x}_{st} - \mathbf{z}_{4(s-1)t}^{\top} \otimes (\widetilde{\mathbf{z}}_{7(s-1)t} \mathbb{I}_{\mathbf{W}_3, \mathbf{b}_3, \mathbf{a}_{2(s-1)t}}) \otimes \mathbf{x}_{(s-1)t}) \mathbf{e}_{3st}$$

$$+ (\mathbf{z}_{st} \otimes \mathbf{x}_{st} - \mathbf{z}_{(s-1)t} \otimes \mathbf{x}_{(s-1)t})(\mathbf{z}_{4st} \otimes \mathbf{a}_{3st} - \mathbf{z}_{4(s-1)t} \otimes \mathbf{a}_{3(s-1)t})^{\top} \big)$$

$$\frac{\partial^2 \mathcal{L}}{\partial b_4 \partial W_1} = \sum_{s,t=1}^{n} (\mathbf{z}_{st} \otimes \mathbf{x}_{st}) \mathbf{z}_{4st}^{\top} + \beta \big( (\mathbf{z}_{st} \otimes \mathbf{x}_{st} - \mathbf{z}_{s(t-1)} \otimes \mathbf{x}_{s(t-1)})(\mathbf{z}_{4st} - \mathbf{z}_{4s(t-1)})^{\top}$$

$$+ (\mathbf{z}_{st} \otimes \mathbf{x}_{st} - \mathbf{z}_{(s-1)t} \otimes \mathbf{x}_{(s-1)t})(\mathbf{z}_{4st} - \mathbf{z}_{4(s-1)t})^{\top} \big)$$

$$\frac{\partial^2 \mathcal{L}}{\partial W_5 \partial W_1} = \sum_{s,t=1}^{n} ((\widetilde{\mathbf{z}}_{8st} \mathbb{I}_{\mathbf{W}_4, \mathbf{b}_4, \mathbf{a}_{3st}}) \otimes \mathbf{x}_{st}) \mathbf{e}_{1st} + (\mathbf{z}_{st} \otimes \mathbf{x}_{st}) \mathbf{a}_{4st}^{\top}$$

$$+ \beta \big( ((\widetilde{\mathbf{z}}_{8st} \mathbb{I}_{\mathbf{W}_4, \mathbf{b}_4, \mathbf{a}_{3st}}) \otimes \mathbf{x}_{st} - (\widetilde{\mathbf{z}}_{8s(t-1)} \mathbb{I}_{\mathbf{W}_4, \mathbf{b}_4, \mathbf{a}_{3s(t-1)}}) \otimes \mathbf{x}_{s(t-1)}) \mathbf{e}_{2st}$$

$$+ (\mathbf{z}_{st} \otimes \mathbf{x}_{st} - \mathbf{z}_{s(t-1)} \otimes \mathbf{x}_{s(t-1)})(\mathbf{a}_{4st} - \mathbf{a}_{4s(t-1)})^{\top}$$

$$+ ((\widetilde{\mathbf{z}}_{8st} \mathbb{I}_{\mathbf{W}_4, \mathbf{b}_4, \mathbf{a}_{3st}}) \otimes \mathbf{x}_{st} - (\widetilde{\mathbf{z}}_{8(s-1)t} \mathbb{I}_{\mathbf{W}_4, \mathbf{b}_4, \mathbf{a}_{3(s-1)t}}) \otimes \mathbf{x}_{(s-1)t}) \mathbf{e}_{3st}$$

$$+ (\mathbf{z}_{st} \otimes \mathbf{x}_{st} - \mathbf{z}_{(s-1)t} \otimes \mathbf{x}_{(s-1)t})(\mathbf{a}_{4st} - \mathbf{a}_{4(s-1)t})^{\top} \big)$$

$$\frac{\partial^2 \mathcal{L}}{\partial b_5 \partial W_1} = \sum_{s,t=1}^{n} \mathbf{z}_{st} \otimes \mathbf{x}_{st}$$

$$\frac{\partial^2 \mathcal{L}}{\partial b_1^2} = \sum_{s,t=1}^{n} \mathbf{z}_{st} \mathbf{z}_{st}^{\top} + \beta \big( (\mathbf{z}_{st} - \mathbf{z}_{s(t-1)})(\mathbf{z}_{st} - \mathbf{z}_{s(t-1)})^{\top} + (\mathbf{z}_{st} - \mathbf{z}_{(s-1)t})(\mathbf{z}_{st} - \mathbf{z}_{(s-1)t})^{\top} \big)$$

$$\frac{\partial^2 \mathcal{L}}{\partial W_2 \partial b_1} = \sum_{s,t=1}^{n} \left( \widetilde{\mathbf{z}}_{6st}^{\top} \otimes \mathbb{I}_{\mathbf{W}_1, \mathbf{b}_1, \mathbf{x}_{st}} \right) \mathbf{e}_{1st} + \mathbf{z}_{st} (\widetilde{\mathbf{z}}_{6st} \otimes \mathbf{a}_{1st})^{\top}$$

$$+ \beta \big( (\widetilde{\mathbf{z}}_{6st}^{\top} \otimes \mathbb{I}_{\mathbf{W}_1, \mathbf{b}_1, \mathbf{x}_{st}} - \widetilde{\mathbf{z}}_{6s(t-1)}^{\top} \otimes \mathbb{I}_{\mathbf{W}_1, \mathbf{b}_1, \mathbf{x}_{s(t-1)}}) \mathbf{e}_{2st}$$

$$+ (\mathbf{z}_{st} - \mathbf{z}_{s(t-1)})(\widetilde{\mathbf{z}}_{6st} \otimes \mathbf{a}_{1st} - \widetilde{\mathbf{z}}_{6s(t-1)} \otimes \mathbf{a}_{1s(t-1)})^{\top}$$

$$+ (\widetilde{\mathbf{z}}_{6st}^{\top} \otimes \mathbb{I}_{\mathbf{W}_1, \mathbf{b}_1, \mathbf{x}_{st}} - \widetilde{\mathbf{z}}_{6(s-1)t}^{\top} \otimes \mathbb{I}_{\mathbf{W}_1, \mathbf{b}_1, \mathbf{x}_{(s-1)t}}) \mathbf{e}_{3st}$$

$$+ (\mathbf{z}_{st} - \mathbf{z}_{(s-1)t})(\widetilde{\mathbf{z}}_{6st} \otimes \mathbf{a}_{1st} - \widetilde{\mathbf{z}}_{6(s-1)t} \otimes \mathbf{a}_{1(s-1)t})^{\top} \big)$$

$$\frac{\partial^2 \mathcal{L}}{\partial b_2 \partial b_1} = \sum_{s,t=1}^{n} \mathbf{z}_{st} \widetilde{\mathbf{z}}_{6st}^{\top} + \beta \big( (\mathbf{z}_{st} - \mathbf{z}_{s(t-1)})(\widetilde{\mathbf{z}}_{6st} - \widetilde{\mathbf{z}}_{6s(t-1)})^{\top} + (\mathbf{z}_{st} - \mathbf{z}_{(s-1)t})(\widetilde{\mathbf{z}}_{6st} - \widetilde{\mathbf{z}}_{6(s-1)t})^{\top} \big)$$

$$\frac{\partial^2 \mathcal{L}}{\partial W_3 \partial b_1} = \sum_{s,t=1}^{n} \left( \widetilde{\mathbf{z}}_{5st}^{\top} \otimes (\mathbf{z}_{1st} \mathbb{I}_{\mathbf{W}_2, \mathbf{b}_2, \mathbf{a}_{1st}}) \right) \mathbf{e}_{1st} + \mathbf{z}_{st} (\widetilde{\mathbf{z}}_{5st} \otimes \mathbf{a}_{2st})^{\top}$$

$$+ \beta \big( (\widetilde{\mathbf{z}}_{5st}^{\top} \otimes (\mathbf{z}_{1st} \mathbb{I}_{\mathbf{W}_2, \mathbf{b}_2, \mathbf{a}_{1st}}) - \widetilde{\mathbf{z}}_{5s(t-1)}^{\top} \otimes (\mathbf{z}_{1s(t-1)} \mathbb{I}_{\mathbf{W}_2, \mathbf{b}_2, \mathbf{a}_{1s(t-1)}})) \mathbf{e}_{2st}$$

$$+ (\mathbf{z}_{st} - \mathbf{z}_{s(t-1)})(\widetilde{\mathbf{z}}_{5st} \otimes \mathbf{a}_{2st} - \widetilde{\mathbf{z}}_{5s(t-1)} \otimes \mathbf{a}_{2s(t-1)})^{\top}$$

$$+ (\widetilde{\mathbf{z}}_{5st}^{\top} \otimes (\mathbf{z}_{1st} \mathbb{I}_{\mathbf{W}_2, \mathbf{b}_2, \mathbf{a}_{1st}}) - \widetilde{\mathbf{z}}_{5(s-1)t}^{\top} \otimes (\mathbf{z}_{1(s-1)t} \mathbb{I}_{\mathbf{W}_2, \mathbf{b}_2, \mathbf{a}_{1(s-1)t}})) \mathbf{e}_{3st}$$

$$+ (\mathbf{z}_{st} - \mathbf{z}_{(s-1)t})(\widetilde{\mathbf{z}}_{5st} \otimes \mathbf{a}_{2st} - \widetilde{\mathbf{z}}_{5(s-1)t} \otimes \mathbf{a}_{2(s-1)t})^{\top} \big)$$

$$\frac{\partial^2 \mathcal{L}}{\partial b_3 \partial b_1} = \sum_{s,t=1}^{n} \mathbf{z}_{st} \widetilde{\mathbf{z}}_{5st}^{\top} + \beta\big((\mathbf{z}_{st} - \mathbf{z}_{s(t-1)})(\widetilde{\mathbf{z}}_{5st} - \widetilde{\mathbf{z}}_{5s(t-1)})^{\top} + (\mathbf{z}_{st} - \mathbf{z}_{(s-1)t})(\widetilde{\mathbf{z}}_{5st} - \widetilde{\mathbf{z}}_{5(s-1)t})^{\top}\big)$$

$$\frac{\partial^2 \mathcal{L}}{\partial W_4 \partial b_1} = \sum_{s,t=1}^{n} \Big(\mathbf{z}_{4st}^{\top} \otimes (\widetilde{\mathbf{z}}_{7st} \mathbb{I}_{\mathbf{W}_3,\mathbf{b}_3,\mathbf{a}_{2st}})\Big) \mathbf{e}_{1st} + \mathbf{z}_{st}\big(\mathbf{z}_{4st} \otimes \mathbf{a}_{3st}\big)^{\top}$$
$$+ \beta\big((\mathbf{z}_{4st}^{\top} \otimes (\widetilde{\mathbf{z}}_{7st} \mathbb{I}_{\mathbf{W}_3,\mathbf{b}_3,\mathbf{a}_{2st}}) - \mathbf{z}_{4s(t-1)}^{\top} \otimes (\widetilde{\mathbf{z}}_{7s(t-1)} \mathbb{I}_{\mathbf{W}_3,\mathbf{b}_3,\mathbf{a}_{2s(t-1)}}))\mathbf{e}_{2st}$$
$$+ (\mathbf{z}_{st} - \mathbf{z}_{s(t-1)})(\mathbf{z}_{4st} \otimes \mathbf{a}_{3st} - \mathbf{z}_{4s(t-1)} \otimes \mathbf{a}_{3s(t-1)})^{\top}$$
$$+ (\mathbf{z}_{4st}^{\top} \otimes (\widetilde{\mathbf{z}}_{7st} \mathbb{I}_{\mathbf{W}_3,\mathbf{b}_3,\mathbf{a}_{2st}}) - \mathbf{z}_{4(s-1)t}^{\top} \otimes (\widetilde{\mathbf{z}}_{7(s-1)t} \mathbb{I}_{\mathbf{W}_3,\mathbf{b}_3,\mathbf{a}_{2(s-1)t}}))\mathbf{e}_{3st}$$
$$+ (\mathbf{z}_{st} - \mathbf{z}_{(s-1)t})(\mathbf{z}_{4st} \otimes \mathbf{a}_{3st} - \mathbf{z}_{4(s-1)t} \otimes \mathbf{a}_{3(s-1)t})^{\top}\big)$$

$$\frac{\partial^2 \mathcal{L}}{\partial b_4 \partial b_1} = \sum_{s,t=1}^{n} \mathbf{z}_{st} \mathbf{z}_{4st}^{\top} + \beta\big((\mathbf{z}_{st} - \mathbf{z}_{s(t-1)})(\mathbf{z}_{4st} - \mathbf{z}_{4s(t-1)})^{\top} + (\mathbf{z}_{st} - \mathbf{z}_{(s-1)t})(\mathbf{z}_{4st} - \mathbf{z}_{4(s-1)t})^{\top}\big)$$

$$\frac{\partial^2 \mathcal{L}}{\partial W_5 \partial b_1} = \sum_{s,t=1}^{n} \widetilde{\mathbf{z}}_{8st} \mathbb{I}_{\mathbf{W}_4,\mathbf{b}_4,\mathbf{a}_{3st}} \mathbf{e}_{1st} + \mathbf{z}_{st} \mathbf{a}_{4st}^{\top}$$
$$+ \beta\big((\widetilde{\mathbf{z}}_{8st} \mathbb{I}_{\mathbf{W}_4,\mathbf{b}_4,\mathbf{a}_{3st}} - \widetilde{\mathbf{z}}_{8s(t-1)} \mathbb{I}_{\mathbf{W}_4,\mathbf{b}_4,\mathbf{a}_{3s(t-1)}})\mathbf{e}_{2st} + (\mathbf{z}_{st} - \mathbf{z}_{s(t-1)})(\mathbf{a}_{4st} - \mathbf{a}_{4s(t-1)})^{\top}$$
$$+ (\widetilde{\mathbf{z}}_{8st} \mathbb{I}_{\mathbf{W}_4,\mathbf{b}_4,\mathbf{a}_{3st}} - \widetilde{\mathbf{z}}_{8(s-1)t} \mathbb{I}_{\mathbf{W}_4,\mathbf{b}_4,\mathbf{a}_{3(s-1)t}})\mathbf{e}_{3st} + (\mathbf{z}_{st} - \mathbf{z}_{(s-1)t})(\mathbf{a}_{4st} - \mathbf{a}_{4(s-1)t})^{\top}\big)$$

$$\frac{\partial^2 \mathcal{L}}{\partial b_5 \partial b_1} = \sum_{s,t=1}^{n} \mathbf{z}_{st}$$

$$\frac{\partial^2 \mathcal{L}}{\partial W_2^2} = \sum_{s,t=1}^{n} (\widetilde{\mathbf{z}}_{6st} \otimes \mathbf{a}_{1st})(\widetilde{\mathbf{z}}_{6st} \otimes \mathbf{a}_{1st})^{\top}$$
$$+ \beta\big((\widetilde{\mathbf{z}}_{6st} \otimes \mathbf{a}_{1st} - \widetilde{\mathbf{z}}_{6s(t-1)} \otimes \mathbf{a}_{1s(t-1)})(\widetilde{\mathbf{z}}_{6st} \otimes \mathbf{a}_{1st} - \widetilde{\mathbf{z}}_{6s(t-1)} \otimes \mathbf{a}_{1s(t-1)})^{\top}$$
$$+ (\widetilde{\mathbf{z}}_{6st} \otimes \mathbf{a}_{1st} - \widetilde{\mathbf{z}}_{6(s-1)t} \otimes \mathbf{a}_{1(s-1)t})(\widetilde{\mathbf{z}}_{6st} \otimes \mathbf{a}_{1st} - \widetilde{\mathbf{z}}_{6(s-1)t} \otimes \mathbf{a}_{1(s-1)t})^{\top}\big)$$

$$\frac{\partial^2 \mathcal{L}}{\partial b_2 \partial W_2} = \sum_{s,t=1}^{n} (\widetilde{\mathbf{z}}_{6st} \otimes \mathbf{a}_{1st})\widetilde{\mathbf{z}}_{6st}^{\top} + \beta\big((\widetilde{\mathbf{z}}_{6st} \otimes \mathbf{a}_{1st} - \widetilde{\mathbf{z}}_{6s(t-1)} \otimes \mathbf{a}_{1s(t-1)})(\widetilde{\mathbf{z}}_{6st} - \widetilde{\mathbf{z}}_{6s(t-1)})^{\top}$$
$$+ (\widetilde{\mathbf{z}}_{6st} \otimes \mathbf{a}_{1st} - \widetilde{\mathbf{z}}_{6(s-1)t} \otimes \mathbf{a}_{1(s-1)t})(\widetilde{\mathbf{z}}_{6st} - \widetilde{\mathbf{z}}_{6(s-1)t})^{\top}\big)$$

$$\frac{\partial^2 \mathcal{L}}{\partial W_3 \partial W_2} = \sum_{s,t=1}^{n} \Big(\widetilde{\mathbf{z}}_{5st}^{\top} \otimes \mathbb{I}_{\mathbf{W}_2,\mathbf{b}_2,\mathbf{a}_{1st}} \otimes \mathbf{a}_{1st}\Big) \mathbf{e}_{1st} + (\widetilde{\mathbf{z}}_{6st} \otimes \mathbf{a}_{1st})(\widetilde{\mathbf{z}}_{5st} \otimes \mathbf{a}_{2st})^{\top}$$
$$+ \beta\big((\widetilde{\mathbf{z}}_{5st}^{\top} \otimes \mathbb{I}_{\mathbf{W}_2,\mathbf{b}_2,\mathbf{a}_{1st}} \otimes \mathbf{a}_{1st} - \widetilde{\mathbf{z}}_{5s(t-1)}^{\top} \otimes \mathbb{I}_{\mathbf{W}_2,\mathbf{b}_2,\mathbf{a}_{1s(t-1)}} \otimes \mathbf{a}_{1s(t-1)})\mathbf{e}_{2st}$$
$$+ (\widetilde{\mathbf{z}}_{6st} \otimes \mathbf{a}_{1st} - \widetilde{\mathbf{z}}_{6s(t-1)} \otimes \mathbf{a}_{1s(t-1)})(\widetilde{\mathbf{z}}_{5st} \otimes \mathbf{a}_{2st} - \widetilde{\mathbf{z}}_{5s(t-1)} \otimes \mathbf{a}_{2s(t-1)})^{\top}$$
$$+ (\widetilde{\mathbf{z}}_{5st}^{\top} \otimes \mathbb{I}_{\mathbf{W}_2,\mathbf{b}_2,\mathbf{a}_{1st}} \otimes \mathbf{a}_{1st} - \widetilde{\mathbf{z}}_{5(s-1)t}^{\top} \otimes \mathbb{I}_{\mathbf{W}_2,\mathbf{b}_2,\mathbf{a}_{1(s-1)t}} \otimes \mathbf{a}_{1(s-1)t})\mathbf{e}_{3st}$$
$$+ (\widetilde{\mathbf{z}}_{6st} \otimes \mathbf{a}_{1st} - \widetilde{\mathbf{z}}_{6s(t-1)} \otimes \mathbf{a}_{1s(t-1)})(\widetilde{\mathbf{z}}_{5st} \otimes \mathbf{a}_{2st} - \widetilde{\mathbf{z}}_{5(s-1)t} \otimes \mathbf{a}_{2(s-1)t})^{\top}\big)$$

$$\frac{\partial^2 \mathcal{L}}{\partial b_3 \partial W_2} = \sum_{s,t=1}^{n} (\widetilde{\mathbf{z}}_{6st} \otimes \mathbf{a}_{1st})\widetilde{\mathbf{z}}_{5st}^{\top} + \beta\big((\widetilde{\mathbf{z}}_{6st} \otimes \mathbf{a}_{1st} - \widetilde{\mathbf{z}}_{6s(t-1)} \otimes \mathbf{a}_{1s(t-1)})(\widetilde{\mathbf{z}}_{5st} - \widetilde{\mathbf{z}}_{5s(t-1)})^{\top}$$
$$+ (\widetilde{\mathbf{z}}_{6st} \otimes \mathbf{a}_{1st} - \widetilde{\mathbf{z}}_{6s(t-1)} \otimes \mathbf{a}_{1s(t-1)})(\widetilde{\mathbf{z}}_{5st} - \widetilde{\mathbf{z}}_{5(s-1)t})$$

$$\frac{\partial^2 \mathcal{L}}{\partial W_4 \partial W_2} = \sum_{s,t=1}^{n} \left( \mathbf{z}_{4st}^\top \otimes (\mathbf{z}_{3st} \mathbb{I}_{\mathbf{W}_3,\mathbf{b}_3,\mathbf{a}_{2st}}) \otimes \mathbf{a}_{1st} \right) \mathbf{e}_{1st} + (\widetilde{\mathbf{z}}_{6st} \otimes \mathbf{a}_{1st})(\mathbf{z}_{4st} \otimes \mathbf{a}_{3st})^\top$$

$$+ \beta \big( (\mathbf{z}_{4st}^\top \otimes (\mathbf{z}_{3st} \mathbb{I}_{\mathbf{W}_3,\mathbf{b}_3,\mathbf{a}_{2st}}) \otimes \mathbf{a}_{1st} - \mathbf{z}_{4s(t-1)}^\top \otimes (\mathbf{z}_{3s(t-1)} \mathbb{I}_{\mathbf{W}_3,\mathbf{b}_3,\mathbf{a}_{2s(t-1)}}) \otimes \mathbf{a}_{1s(t-1)}) \mathbf{e}_{2st}$$

$$+ (\widetilde{\mathbf{z}}_{6st} \otimes \mathbf{a}_{1st} - \widetilde{\mathbf{z}}_{6s(t-1)} \otimes \mathbf{a}_{1s(t-1)})(\mathbf{z}_{4st} \otimes \mathbf{a}_{3st} - \mathbf{z}_{4s(t-1)} \otimes \mathbf{a}_{3s(t-1)})^\top$$

$$+ (\mathbf{z}_{4st}^\top \otimes (\mathbf{z}_{3st} \mathbb{I}_{\mathbf{W}_3,\mathbf{b}_3,\mathbf{a}_{2st}}) \otimes \mathbf{a}_{1st} - \mathbf{z}_{4(s-1)t}^\top \otimes (\mathbf{z}_{3(s-1)t} \mathbb{I}_{\mathbf{W}_3,\mathbf{b}_3,\mathbf{a}_{3(s-1)t}}) \otimes \mathbf{a}_{1(s-1)t}) \mathbf{e}_{3st}$$

$$+ (\widetilde{\mathbf{z}}_{6st} \otimes \mathbf{a}_{1st} - \widetilde{\mathbf{z}}_{6s(t-1)} \otimes \mathbf{a}_{1s(t-1)})(\mathbf{z}_{4st} \otimes \mathbf{a}_{3st} - \mathbf{z}_{4(s-1)t} \otimes \mathbf{a}_{3(s-1)t})^\top \big)$$

$$\frac{\partial^2 \mathcal{L}}{\partial b_4 \partial W_2} = \sum_{s,t=1}^{n} (\widetilde{\mathbf{z}}_{6st} \otimes \mathbf{a}_{1st}) \mathbf{z}_{4st}^\top + \beta \big( (\widetilde{\mathbf{z}}_{6st} \otimes \mathbf{a}_{1st} - \widetilde{\mathbf{z}}_{6s(t-1)} \otimes \mathbf{a}_{1s(t-1)})(\mathbf{z}_{4st} - \mathbf{z}_{4s(t-1)})^\top$$

$$+ (\widetilde{\mathbf{z}}_{6st} \otimes \mathbf{a}_{1st} - \widetilde{\mathbf{z}}_{6s(t-1)} \otimes \mathbf{a}_{1s(t-1)})(\mathbf{z}_{4st} - \mathbf{z}_{4(s-1)t})^\top \big)$$

$$\frac{\partial^2 \mathcal{L}}{\partial W_5 \partial W_2} = \sum_{s,t=1}^{n} ((\widetilde{\mathbf{z}}_{9st} \mathbb{I}_{\mathbf{W}_4,\mathbf{b}_4,\mathbf{a}_{3st}}) \otimes \mathbf{a}_{1st}) \mathbf{e}_{1st} + (\widetilde{\mathbf{z}}_{6st} \otimes \mathbf{a}_{1st}) \mathbf{a}_{4st}^\top$$

$$+ \beta \big( ((\widetilde{\mathbf{z}}_{9st} \mathbb{I}_{\mathbf{W}_4,\mathbf{b}_4,\mathbf{a}_{3st}}) \otimes \mathbf{a}_{1st} - (\widetilde{\mathbf{z}}_{9s(t-1)} \mathbb{I}_{\mathbf{W}_3,\mathbf{b}_3,\mathbf{a}_{2s(t-1)}}) \otimes \mathbf{a}_{1s(t-1)}) \mathbf{e}_{2st}$$

$$+ (\widetilde{\mathbf{z}}_{6st} \otimes \mathbf{a}_{1st} - \widetilde{\mathbf{z}}_{6s(t-1)} \otimes \mathbf{a}_{1s(t-1)})(\mathbf{a}_{4st} - \mathbf{a}_{4s(t-1)})^\top$$

$$+ ((\widetilde{\mathbf{z}}_{9st} \mathbb{I}_{\mathbf{W}_4,\mathbf{b}_4,\mathbf{a}_{3st}}) \otimes \mathbf{a}_{1st} - (\widetilde{\mathbf{z}}_{9(s-1)t} \mathbb{I}_{\mathbf{W}_4,\mathbf{b}_4,\mathbf{a}_{4(s-1)t}}) \otimes \mathbf{a}_{1(s-1)t}) \mathbf{e}_{3st}$$

$$+ (\widetilde{\mathbf{z}}_{6st} \otimes \mathbf{a}_{1st} - \widetilde{\mathbf{z}}_{6s(t-1)} \otimes \mathbf{a}_{1s(t-1)})(\mathbf{a}_{4st} - \mathbf{a}_{4(s-1)t})^\top \big)$$

$$\frac{\partial^2 \mathcal{L}}{\partial W_5 \partial W_2} = \sum_{s,t=1}^{n} \widetilde{\mathbf{z}}_{6st} \otimes \mathbf{a}_{1st}$$

$$\frac{\partial^2 \mathcal{L}}{\partial b_2^2} = \sum_{s,t=1}^{n} \widetilde{\mathbf{z}}_{6st} \widetilde{\mathbf{z}}_{6st}^\top + \beta \big( (\widetilde{\mathbf{z}}_{6st} - \widetilde{\mathbf{z}}_{6s(t-1)})(\widetilde{\mathbf{z}}_{6st} - \widetilde{\mathbf{z}}_{6s(t-1)})^\top + (\widetilde{\mathbf{z}}_{6st} - \widetilde{\mathbf{z}}_{6(s-1)t})(\widetilde{\mathbf{z}}_{6st} - \widetilde{\mathbf{z}}_{6(s-1)t})^\top \big)$$

$$\frac{\partial^2 \mathcal{L}}{\partial W_3 \partial b_2} = \sum_{s,t=1}^{n} \left( \widetilde{\mathbf{z}}_{5st}^\top \otimes \mathbb{I}_{\mathbf{W}_2,\mathbf{b}_2,\mathbf{a}_{1st}} \right) \mathbf{e}_{1st} + \widetilde{\mathbf{z}}_{6st} (\widetilde{\mathbf{z}}_{5st} \otimes \mathbf{a}_{2st})^\top$$

$$+ \beta \big( (\widetilde{\mathbf{z}}_{5st}^\top \otimes \mathbb{I}_{\mathbf{W}_2,\mathbf{b}_2,\mathbf{a}_{1st}} - \widetilde{\mathbf{z}}_{5s(t-1)}^\top \otimes \mathbb{I}_{\mathbf{W}_2,\mathbf{b}_2,\mathbf{a}_{1s(t-1)}}) \mathbf{e}_{2st}$$

$$+ (\widetilde{\mathbf{z}}_{6st} - \widetilde{\mathbf{z}}_{6s(t-1)})(\widetilde{\mathbf{z}}_{5st} \otimes \mathbf{a}_{2st} - \widetilde{\mathbf{z}}_{5s(t-1)} \otimes \mathbf{a}_{2s(t-1)})^\top$$

$$+ (\widetilde{\mathbf{z}}_{5st}^\top \otimes \mathbb{I}_{\mathbf{W}_2,\mathbf{b}_2,\mathbf{a}_{1st}} - \widetilde{\mathbf{z}}_{5(s-1)t}^\top \otimes \mathbb{I}_{\mathbf{W}_2,\mathbf{b}_2,\mathbf{a}_{1(s-1)t}}) \mathbf{e}_{3st}$$

$$+ (\widetilde{\mathbf{z}}_{6st} - \widetilde{\mathbf{z}}_{6s(t-1)})(\widetilde{\mathbf{z}}_{5st} \otimes \mathbf{a}_{2st} - \widetilde{\mathbf{z}}_{5(s-1)t} \otimes \mathbf{a}_{2(s-1)t})^\top \big)$$

$$\frac{\partial^2 \mathcal{L}}{\partial b_3 \partial b_2} = \sum_{s,t=1}^{n} \widetilde{\mathbf{z}}_{6st} \widetilde{\mathbf{z}}_{5st}^\top + \beta \big( (\widetilde{\mathbf{z}}_{6st} - \widetilde{\mathbf{z}}_{6s(t-1)})(\widetilde{\mathbf{z}}_{5st} - \widetilde{\mathbf{z}}_{5s(t-1)})^\top$$

$$+ (\widetilde{\mathbf{z}}_{6st} - \widetilde{\mathbf{z}}_{6s(t-1)})(\widetilde{\mathbf{z}}_{5st} - \widetilde{\mathbf{z}}_{5(s-1)t})^\top$$

$$\frac{\partial^2 \mathcal{L}}{\partial W_4 \partial b_2} = \sum_{s,t=1}^{n} \left( \mathbf{z}_{4st}^\top \otimes (\mathbf{z}_{3st} \mathbb{I}_{\mathbf{W}_3,\mathbf{b}_3,\mathbf{a}_{2st}}) \right) \mathbf{e}_{1st} + \widetilde{\mathbf{z}}_{6st} (\mathbf{z}_{4st} \otimes \mathbf{a}_{3st})^\top$$

$$+ \beta \big( (\mathbf{z}_{4st}^\top \otimes (\mathbf{z}_{3st} \mathbb{I}_{\mathbf{W}_3,\mathbf{b}_3,\mathbf{a}_{2st}}) - \mathbf{z}_{4s(t-1)}^\top \otimes (\mathbf{z}_{3s(t-1)} \mathbb{I}_{\mathbf{W}_3,\mathbf{b}_3,\mathbf{a}_{2s(t-1)}})) \mathbf{e}_{2st}$$

$$+ (\widetilde{\mathbf{z}}_{6st} - \widetilde{\mathbf{z}}_{6s(t-1)})(\mathbf{z}_{4st} \otimes \mathbf{a}_{3st} - \mathbf{z}_{4s(t-1)} \otimes \mathbf{a}_{3s(t-1)})^\top$$

$$+ (\mathbf{z}_{4st}^\top \otimes (\mathbf{z}_{3st} \mathbb{I}_{\mathbf{W}_3,\mathbf{b}_3,\mathbf{a}_{2st}}) - \mathbf{z}_{4(s-1)t}^\top \otimes (\mathbf{z}_{3(s-1)t} \mathbb{I}_{\mathbf{W}_3,\mathbf{b}_3,\mathbf{a}_{3(s-1)t}})) \mathbf{e}_{3st}$$

$$+ (\widetilde{\mathbf{z}}_{6st} - \widetilde{\mathbf{z}}_{6s(t-1)})(\mathbf{z}_{4st} \otimes \mathbf{a}_{3st} - \mathbf{z}_{4(s-1)t} \otimes \mathbf{a}_{3(s-1)t})^\top \big)$$

$$\frac{\partial^2 \mathcal{L}}{\partial b_4 \partial b_2} = \sum_{s,t=1}^{n} \widetilde{\mathbf{z}}_{6st} \mathbf{z}_{4st}^\top + \beta\big((\widetilde{\mathbf{z}}_{6st} - \widetilde{\mathbf{z}}_{6s(t-1)})(\mathbf{z}_{4st} - \mathbf{z}_{4s(t-1)})^\top$$
$$+ (\widetilde{\mathbf{z}}_{6st} - \widetilde{\mathbf{z}}_{6s(t-1)})(\mathbf{z}_{4st} - \mathbf{z}_{4(s-1)t})^\top\big)$$

$$\frac{\partial^2 \mathcal{L}}{\partial W_5 \partial b_2} = \sum_{s,t=1}^{n} \left(\widetilde{\mathbf{z}}_{9st} \mathbb{I}_{\mathbf{W}_4, \mathbf{b}_4, \mathbf{a}_{3st}}\right) \mathbf{e}_{1st} + \widetilde{\mathbf{z}}_{6st} \mathbf{a}_{4st}^\top$$
$$+ \beta\big((\widetilde{\mathbf{z}}_{9st} \mathbb{I}_{\mathbf{W}_4, \mathbf{b}_4, \mathbf{a}_{3st}} - \widetilde{\mathbf{z}}_{9s(t-1)} \mathbb{I}_{\mathbf{W}_3, \mathbf{b}_3, \mathbf{a}_{2s(t-1)}}) \mathbf{e}_{2st}$$
$$+ (\widetilde{\mathbf{z}}_{6st} - \widetilde{\mathbf{z}}_{6s(t-1)})(\mathbf{a}_{4st} - \mathbf{a}_{4s(t-1)})^\top$$
$$+ (\widetilde{\mathbf{z}}_{9st} \mathbb{I}_{\mathbf{W}_4, \mathbf{b}_4, \mathbf{a}_{3st}} - \widetilde{\mathbf{z}}_{9(s-1)t} \mathbb{I}_{\mathbf{W}_4, \mathbf{b}_4, \mathbf{a}_{4(s-1)t}}) \mathbf{e}_{3st}$$
$$+ (\widetilde{\mathbf{z}}_{6st} - \widetilde{\mathbf{z}}_{6s(t-1)})(\mathbf{a}_{4st} - \mathbf{a}_{4(s-1)t})^\top\big)$$

$$\frac{\partial^2 \mathcal{L}}{\partial W_5 \partial b_2} = \sum_{s,t=1}^{n} \widetilde{\mathbf{z}}_{6st}$$

$$\frac{\partial^2 \mathcal{L}}{\partial W_3^2} = \sum_{s,t=1}^{n} (\widetilde{\mathbf{z}}_{5st} \otimes \mathbf{a}_{2st})(\widetilde{\mathbf{z}}_{5st} \otimes \mathbf{a}_{2st})^\top$$
$$+ \beta\big((\widetilde{\mathbf{z}}_{5st} \otimes \mathbf{a}_{2st} - \widetilde{\mathbf{z}}_{5s(t-1)} \otimes \mathbf{a}_{2s(t-1)})(\widetilde{\mathbf{z}}_{5st} \otimes \mathbf{a}_{2st} - \widetilde{\mathbf{z}}_{5s(t-1)} \otimes \mathbf{a}_{2s(t-1)})^\top$$
$$+ (\widetilde{\mathbf{z}}_{5st} \otimes \mathbf{a}_{2st} - \widetilde{\mathbf{z}}_{5(s-1)t} \otimes \mathbf{a}_{2(s-1)t})(\widetilde{\mathbf{z}}_{5st} \otimes \mathbf{a}_{2st} - \widetilde{\mathbf{z}}_{5(s-1)t} \otimes \mathbf{a}_{2(s-1)t})^\top\big)$$

$$\frac{\partial^2 \mathcal{L}}{\partial b_3 \partial W_3} = \sum_{s,t=1}^{n} (\widetilde{\mathbf{z}}_{5st} \otimes \mathbf{a}_{2st}) \widetilde{\mathbf{z}}_{5st}^\top + \beta\big((\widetilde{\mathbf{z}}_{5st} \otimes \mathbf{a}_{2st} - \widetilde{\mathbf{z}}_{5s(t-1)} \otimes \mathbf{a}_{2s(t-1)})(\widetilde{\mathbf{z}}_{5st} - \widetilde{\mathbf{z}}_{5s(t-1)})^\top$$
$$+ (\widetilde{\mathbf{z}}_{5st} \otimes \mathbf{a}_{2st} - \widetilde{\mathbf{z}}_{5s(t-1)} \otimes \mathbf{a}_{2s(t-1)})(\widetilde{\mathbf{z}}_{5st} - \widetilde{\mathbf{z}}_{5(s-1)t})^\top\big)$$

$$\frac{\partial^2 \mathcal{L}}{\partial W_4 \partial W_3} = \sum_{s,t=1}^{n} \left(\mathbf{z}_{4st}^\top \otimes \mathbb{I}_{\mathbf{W}_3, \mathbf{b}_3, \mathbf{a}_{2st}} \otimes \mathbf{a}_{2st}\right) \mathbf{e}_{1st} + (\widetilde{\mathbf{z}}_{5st} \otimes \mathbf{a}_{2st})(\mathbf{z}_{4st} \otimes \mathbf{a}_{3st})^\top$$
$$+ \beta\big((\mathbf{z}_{4st}^\top \otimes \mathbb{I}_{\mathbf{W}_3, \mathbf{b}_3, \mathbf{a}_{2st}} \otimes \mathbf{a}_{2st} - \mathbf{z}_{4s(t-1)}^\top \otimes \mathbb{I}_{\mathbf{W}_3, \mathbf{b}_3, \mathbf{a}_{2s(t-1)}} \otimes \mathbf{a}_{2s(t-1)}) \mathbf{e}_{2st}$$
$$+ (\widetilde{\mathbf{z}}_{5st} \otimes \mathbf{a}_{2st} - \widetilde{\mathbf{z}}_{5s(t-1)} \otimes \mathbf{a}_{2s(t-1)})(\mathbf{z}_{4st} \otimes \mathbf{a}_{3st} - \mathbf{z}_{4s(t-1)} \otimes \mathbf{a}_{3s(t-1)})^\top$$
$$+ (\mathbf{z}_{4st}^\top \otimes \mathbb{I}_{\mathbf{W}_3, \mathbf{b}_3, \mathbf{a}_{2st}} \otimes \mathbf{a}_{2st} - \mathbf{z}_{4(s-1)t}^\top \otimes \mathbb{I}_{\mathbf{W}_3, \mathbf{b}_3, \mathbf{a}_{2(s-1)t}} \otimes \mathbf{a}_{2(s-1)t}) \mathbf{e}_{3st}$$
$$+ (\widetilde{\mathbf{z}}_{5st} \otimes \mathbf{a}_{2st} - \widetilde{\mathbf{z}}_{5(s-1)t} \otimes \mathbf{a}_{2(s-1)t})(\mathbf{z}_{4st} \otimes \mathbf{a}_{3st} - \mathbf{z}_{4(s-1)t} \otimes \mathbf{a}_{3(s-1)t})^\top\big)$$

$$\frac{\partial^2 \mathcal{L}}{\partial b_4 \partial W_3} = \sum_{s,t=1}^{n} (\widetilde{\mathbf{z}}_{5st} \otimes \mathbf{a}_{2st}) \mathbf{z}_{4st}^\top + \beta\big((\widetilde{\mathbf{z}}_{5st} \otimes \mathbf{a}_{2st} - \widetilde{\mathbf{z}}_{5s(t-1)} \otimes \mathbf{a}_{2s(t-1)})(\mathbf{z}_{4st} - \mathbf{z}_{4s(t-1)})^\top$$
$$+ (\widetilde{\mathbf{z}}_{5st} \otimes \mathbf{a}_{2st} - \widetilde{\mathbf{z}}_{5(s-1)t} \otimes \mathbf{a}_{2(s-1)t})(\mathbf{z}_{4st} - \mathbf{z}_{4(s-1)t})^\top\big)$$

$$\frac{\partial^2 \mathcal{L}}{\partial W_5 \partial W_3} = \sum_{s,t=1}^{n} ((\mathbf{z}_{3st} \mathbb{I}_{\mathbf{W}_4, \mathbf{b}_4, \mathbf{a}_{3st}}) \otimes \mathbf{a}_{2st}) \mathbf{e}_{1st} + (\widetilde{\mathbf{z}}_{5st} \otimes \mathbf{a}_{2st}) \mathbf{a}_{4st}^\top$$
$$+ \beta\big(((\mathbf{z}_{3st} \mathbb{I}_{\mathbf{W}_4, \mathbf{b}_4, \mathbf{a}_{3st}}) \otimes \mathbf{a}_{2st} - (\mathbf{z}_{3s(t-1)} \mathbb{I}_{\mathbf{W}_4, \mathbf{b}_4, \mathbf{a}_{3s(t-1)}}) \otimes \mathbf{a}_{2s(t-1)}) \mathbf{e}_{2st}$$
$$+ (\widetilde{\mathbf{z}}_{5st} \otimes \mathbf{a}_{2st} - \widetilde{\mathbf{z}}_{5s(t-1)} \otimes \mathbf{a}_{2s(t-1)})(\mathbf{a}_{4st} - \mathbf{a}_{4s(t-1)})^\top$$
$$+ ((\mathbf{z}_{3st} \mathbb{I}_{\mathbf{W}_4, \mathbf{b}_4, \mathbf{a}_{3st}}) \otimes \mathbf{a}_{2st} - (\mathbf{z}_{3(s-1)t} \mathbb{I}_{\mathbf{W}_4, \mathbf{b}_4, \mathbf{a}_{3(s-1)t}}) \otimes \mathbf{a}_{2(s-1)t}) \mathbf{e}_{3st}$$
$$+ (\widetilde{\mathbf{z}}_{5st} \otimes \mathbf{a}_{2st} - \widetilde{\mathbf{z}}_{5s(t-1)} \otimes \mathbf{a}_{2s(t-1)})(\mathbf{a}_{4st} - \mathbf{a}_{4(s-1)t})^\top\big)$$

$$\frac{\partial^2 \mathcal{L}}{\partial b_5 \partial W_3} = \sum_{s,t=1}^{n} \widetilde{\mathbf{z}}_{5st} \otimes \mathbf{a}_{2st}$$

$$\frac{\partial^2 \mathcal{L}}{\partial b_3^2} = \sum_{s,t=1}^{n} \widetilde{\mathbf{z}}_{5st} \widetilde{\mathbf{z}}_{5st}^\top + \beta \big( (\widetilde{\mathbf{z}}_{5st} - \widetilde{\mathbf{z}}_{5s(t-1)})(\widetilde{\mathbf{z}}_{5st} - \widetilde{\mathbf{z}}_{5s(t-1)})^\top + (\widetilde{\mathbf{z}}_{5st} - \widetilde{\mathbf{z}}_{5(s-1)t})(\widetilde{\mathbf{z}}_{5st} - \widetilde{\mathbf{z}}_{5(s-1)t})^\top \big)$$

$$\begin{aligned}
\frac{\partial^2 \mathcal{L}}{\partial W_4 \partial b_3} &= \sum_{s,t=1}^{n} \left( \mathbf{z}_{4st}^\top \otimes \mathbb{I}_{\mathbf{W}_3, \mathbf{b}_3, \mathbf{a}_{2st}} \right) \mathbf{e}_{1st} + \widetilde{\mathbf{z}}_{5st} \left( \mathbf{z}_{4st} \otimes \mathbf{a}_{3st} \right)^\top \\
&\quad + \beta \big( (\mathbf{z}_{4st}^\top \otimes \mathbb{I}_{\mathbf{W}_3, \mathbf{b}_3, \mathbf{a}_{2st}} - \mathbf{z}_{4s(t-1)}^\top \otimes \mathbb{I}_{\mathbf{W}_3, \mathbf{b}_3, \mathbf{a}_{2s(t-1)}}) \mathbf{e}_{2st} \\
&\quad + (\widetilde{\mathbf{z}}_{5st} - \widetilde{\mathbf{z}}_{5s(t-1)})(\mathbf{z}_{4st} \otimes \mathbf{a}_{3st} - \mathbf{z}_{4s(t-1)} \otimes \mathbf{a}_{3s(t-1)})^\top \\
&\quad + (\mathbf{z}_{4st}^\top \otimes \mathbb{I}_{\mathbf{W}_3, \mathbf{b}_3, \mathbf{a}_{2st}} - \mathbf{z}_{4(s-1)t}^\top \otimes \mathbb{I}_{\mathbf{W}_3, \mathbf{b}_3, \mathbf{a}_{2(s-1)t}}) \mathbf{e}_{3st} \\
&\quad + (\widetilde{\mathbf{z}}_{5st} - \widetilde{\mathbf{z}}_{5(s-1)t})(\mathbf{z}_{4st} \otimes \mathbf{a}_{3st} - \mathbf{z}_{4(s-1)t} \otimes \mathbf{a}_{3(s-1)t})^\top \big)
\end{aligned}$$

$$\begin{aligned}
\frac{\partial^2 \mathcal{L}}{\partial b_4 \partial b_3} &= \sum_{s,t=1}^{n} \widetilde{\mathbf{z}}_{5st} \mathbf{z}_{4st}^\top + \beta \big( (\widetilde{\mathbf{z}}_{5st} - \widetilde{\mathbf{z}}_{5s(t-1)})(\mathbf{z}_{4st} - \mathbf{z}_{4s(t-1)})^\top \\
&\quad + (\widetilde{\mathbf{z}}_{5st} - \widetilde{\mathbf{z}}_{5(s-1)t})(\mathbf{z}_{4st} - \mathbf{z}_{4(s-1)t})^\top \big)
\end{aligned}$$

$$\begin{aligned}
\frac{\partial^2 \mathcal{L}}{\partial W_5 \partial b_3} &= \sum_{s,t=1}^{n} \left( \mathbf{z}_{3st} \mathbb{I}_{\mathbf{W}_4, \mathbf{b}_4, \mathbf{a}_{3st}} \right) \mathbf{e}_{1st} + \widetilde{\mathbf{z}}_{5st} \mathbf{a}_{4st}^\top \\
&\quad + \beta \big( (\mathbf{z}_{3st} \mathbb{I}_{\mathbf{W}_4, \mathbf{b}_4, \mathbf{a}_{3st}} - \mathbf{z}_{3s(t-1)} \mathbb{I}_{\mathbf{W}_4, \mathbf{b}_4, \mathbf{a}_{3s(t-1)}}) \mathbf{e}_{2st} \\
&\quad + (\widetilde{\mathbf{z}}_{5st} - \widetilde{\mathbf{z}}_{5s(t-1)})(\mathbf{a}_{4st} - \mathbf{a}_{4s(t-1)})^\top \\
&\quad + (\mathbf{z}_{3st} \mathbb{I}_{\mathbf{W}_4, \mathbf{b}_4, \mathbf{a}_{3st}} - \mathbf{z}_{3(s-1)t} \mathbb{I}_{\mathbf{W}_4, \mathbf{b}_4, \mathbf{a}_{3(s-1)t}}) \mathbf{e}_{3st} \\
&\quad + (\widetilde{\mathbf{z}}_{5st} - \widetilde{\mathbf{z}}_{5s(t-1)})(\mathbf{a}_{4st} - \mathbf{a}_{4(s-1)t})^\top \big)
\end{aligned}$$

$$\frac{\partial^2 \mathcal{L}}{\partial b_5 \partial b_3} = \sum_{s,t=1}^{n} \widetilde{\mathbf{z}}_{5st}$$

$$\begin{aligned}
\frac{\partial^2 \mathcal{L}}{\partial W_4^2} &= \sum_{s,t=1}^{n} (\mathbf{z}_{4st} \otimes \mathbf{a}_{3st})(\mathbf{z}_{4st} \otimes \mathbf{a}_{3st})^\top \\
&\quad + \beta \big( (\mathbf{z}_{4st} \otimes \mathbf{a}_{3st} - \mathbf{z}_{4s(t-1)} \otimes \mathbf{a}_{3s(t-1)})(\mathbf{z}_{4st} \otimes \mathbf{a}_{3st} - \mathbf{z}_{4s(t-1)} \otimes \mathbf{a}_{3s(t-1)})^\top \\
&\quad + (\mathbf{z}_{4st} \otimes \mathbf{a}_{3st} - \mathbf{z}_{4(s-1)t} \otimes \mathbf{a}_{3(s-1)t})(\mathbf{z}_{4st} \otimes \mathbf{a}_{3st} - \mathbf{z}_{4(s-1)t} \otimes \mathbf{a}_{3(s-1)t})^\top \big)
\end{aligned}$$

$$\begin{aligned}
\frac{\partial^2 \mathcal{L}}{\partial b_4 \partial W_4} &= \sum_{s,t=1}^{n} (\mathbf{z}_{4st} \otimes \mathbf{a}_{3st}) \mathbf{z}_{4st}^\top \\
&\quad + \beta \big( (\mathbf{z}_{4st} \otimes \mathbf{a}_{3st} - \mathbf{z}_{4s(t-1)} \otimes \mathbf{a}_{3s(t-1)})(\mathbf{z}_{4st} - \mathbf{z}_{4s(t-1)})^\top \\
&\quad + (\mathbf{z}_{4st} \otimes \mathbf{a}_{3st} - \mathbf{z}_{4(s-1)t} \otimes \mathbf{a}_{3(s-1)t})(\mathbf{z}_{4st} - \mathbf{z}_{4(s-1)t})^\top \big)
\end{aligned}$$

$$\begin{aligned}
\frac{\partial^2 \mathcal{L}}{\partial W_5 \partial W_4} &= \sum_{s,t=1}^{n} (\mathbb{I}_{\mathbf{W}_4, \mathbf{b}_4, \mathbf{a}_{3st}} \otimes \mathbf{a}_{3st}) \mathbf{e}_{1st} + (\mathbf{z}_{4st} \otimes \mathbf{a}_{3st}) \mathbf{a}_{4st}^\top \\
&\quad + \beta \big( (\mathbb{I}_{\mathbf{W}_4, \mathbf{b}_4, \mathbf{a}_{3st}} \otimes \mathbf{a}_{3st} - \mathbb{I}_{\mathbf{W}_4, \mathbf{b}_4, \mathbf{a}_{3s(t-1)}} \otimes \mathbf{a}_{3s(t-1)}) \mathbf{e}_{2st} + (\mathbf{z}_{4st} \otimes \mathbf{a}_{3st} - \mathbf{z}_{4s(t-1)} \otimes \mathbf{a}_{3s(t-1)})(\mathbf{a}_{4st} - \mathbf{a}_{4s(t-1)})^\top \\
&\quad + (\mathbb{I}_{\mathbf{W}_4, \mathbf{b}_4, \mathbf{a}_{3st}} \otimes \mathbf{a}_{3st} - \mathbb{I}_{\mathbf{W}_4, \mathbf{b}_4, \mathbf{a}_{3(s-1)t}} \otimes \mathbf{a}_{3(s-1)t}) \mathbf{e}_{3st} + (\mathbf{z}_{4st} \otimes \mathbf{a}_{3st} - \mathbf{z}_{4s(t-1)} \otimes \mathbf{a}_{3s(t-1)})(\mathbf{a}_{4st} - \mathbf{a}_{4(s-1)t})^\top \big)
\end{aligned}$$

$$\frac{\partial^2 \mathcal{L}}{\partial b_5 \partial W_4} = \sum_{s,t=1}^{n} \mathbf{z}_{4st} \otimes \mathbf{a}_{3st}$$

$$\frac{\partial^2 \mathcal{L}}{\partial b_4^2} = \sum_{s,t=1}^{n} \mathbf{z}_{4st}\mathbf{z}_{4st}^{\top} + \beta\big((\mathbf{z}_{4st} - \mathbf{z}_{4s(t-1)})(\mathbf{z}_{4st} - \mathbf{z}_{4s(t-1)})^{\top} + (\mathbf{z}_{4st} - \mathbf{z}_{4(s-1)t})(\mathbf{z}_{4st} - \mathbf{z}_{4(s-1)t})^{\top}\big)$$

$$\frac{\partial^2 \mathcal{L}}{\partial W_5 \partial b_4} = \sum_{s,t=1}^{n} \mathbb{I}_{\mathbf{W}_4,\mathbf{b}_4,\mathbf{a}_{3st}}\mathbf{e}_{1st} + \mathbf{z}_{4st}\mathbf{a}_{4st}^{\top}$$

$$+ \beta\big((\mathbb{I}_{\mathbf{W}_4,\mathbf{b}_4,\mathbf{a}_{3st}} - \mathbb{I}_{\mathbf{W}_4,\mathbf{b}_4,\mathbf{a}_{3s(t-1)}})\mathbf{e}_{2st} + (\mathbf{z}_{4st} - \mathbf{z}_{4s(t-1)})(\mathbf{a}_{4st} - \mathbf{a}_{4s(t-1)})^{\top}$$

$$+ (\mathbb{I}_{\mathbf{W}_4,\mathbf{b}_4,\mathbf{a}_{3st}} - \mathbb{I}_{\mathbf{W}_4,\mathbf{b}_4,\mathbf{a}_{3(s-1)t}})\mathbf{e}_{3st} + (\mathbf{z}_{4st} - \mathbf{z}_{4s(t-1)})(\mathbf{a}_{4st} - \mathbf{a}_{4(s-1)t})^{\top}\big)$$

$$\frac{\partial^2 \mathcal{L}}{\partial b_5 \partial b_4} = \sum_{s,t=1}^{n} \mathbf{z}_{4st}$$

$$\frac{\partial^2 \mathcal{L}}{\partial W_5^2} = \sum_{s,t=1}^{n} \mathbf{a}_{4st}\mathbf{a}_{4st}^{\top} + \beta\big((\mathbf{a}_{4st} - \mathbf{a}_{4s(t-1)})(\mathbf{a}_{4st} - \mathbf{a}_{4s(t-1)})^{\top} + (\mathbf{a}_{4st} - \mathbf{a}_{4(s-1)t})(\mathbf{a}_{4st} - \mathbf{a}_{4(s-1)t})^{\top}\big)$$

$$\frac{\partial^2 \mathcal{L}}{\partial b_5 \partial W_5} = \sum_{s,t=1}^{n} \mathbf{a}_{4st}, \quad \frac{\partial^2 \mathcal{L}}{\partial b_5^2} = n^2.$$