# OpenReview forum: "Why Multi-Grade Deep Learning Outperforms Single-Grade: Theory and Practice"
_ICLR.cc/2026/Conference — Submitted to ICLR 2026_

### Official Review · Reviewer_zTvu · 2025-10-31

**Soundness:** 1
**Presentation:** 2
**Contribution:** 2
**Rating:** 2
**Confidence:** 3

**Summary:**

The paper introduces **Multi-Grade Deep Learning (MGDL)**, a training framework that decomposes end-to-end optimization into multiple shallow “grades,” each trained sequentially on the residual errors of previous stages. The authors claim this approach improves convergence stability and robustness to learning rates compared to standard **Single-Grade Deep Learning (SGDL)**. Theoretically, they prove that when each grade uses a single ReLU layer, the training reduces to a series of convex subproblems, and spectral analysis shows MGDL maintains Jacobian eigenvalues within (−1, 1), ensuring stable loss decay. Empirically, MGDL outperforms SGDL on image regression, denoising, CIFAR-10/100, and time-series prediction, showing smoother convergence and higher accuracy.

**Strengths:**

The paper presents an interesting and well-organized discussion of Multi-Grade Deep Learning (MGDL) as an alternative to standard end-to-end training. It combines both theoretical analysis and diverse experimental evidence, offering a clear connection between the proposed method’s mathematical properties, such as convergence guarantees and spectral stability, and its empirical performance across multiple domains including image regression, classification, and time-series prediction. This integration of theory and practice makes the study both insightful and convincing.

**Weaknesses:**

1. **Unconvincing Empirical Validation**: The paper aims to explain **why** MGDL outperforms SGDL, but **whether** this claim holds is still debatable. MGDL is not yet a widely recognized training paradigm, and the current small-scale, toy-level experiments are insufficient to establish its general superiority across deep learning. A single comparative experiment on a canonical benchmark, such as training a ResNet-50 on ImageNet, would be far more persuasive than multiple synthetic or low-complexity tasks.
2. **Limited and Disconnected Theoretical Contribution**: The main theoretical novelty is confined to Section 4, which shows that a single-layer ReLU network under MGDL yields a convex optimization problem. However, the connection between this convex proof and the empirical performance gains of MGDL remains unsubstantiated, leaving the theory detached from the observed advantages.
3. **Overlap with Layer-Wise Training Methods**: MGDL closely resembles established **layer-wise training** approaches. The paper should explicitly discuss this relationship and clarify whether MGDL introduces any substantial difference beyond conceptual rebranding.
4. **Lack of Standard Training Practices**: Section 6 omits common practices such as learning-rate warm-up and decay schedules, which are known to enhance stability and robustness. Their absence weakens the experimental validity of the claimed training stability improvements of MGDL.
5. **Minor Presentation Issues**: Minor issues include small font sizes in some figures (e.g., Figures 4–5) and typographical errors:
   - Line 408: duplicated “0.004”.
   - Line 461: “Although oth models fit the training data affectively” → “Although both models fit the training data effectively.”

**Questions:**

1. **Effectiveness in Fine-Tuning Scenarios**: Is MGDL also effective when applied to fine-tuning a pre-trained network?
2. **Loss of Layer Differentiation**: Previous studies such as [1] have shown that different layers in deep networks play distinct roles during end-to-end training. In contrast, MGDL freezes earlier grades and prevents such differentiation from emerging. Does this structural uniformity benefit generalization, or does it harm the representational hierarchy learned by deep models?
3. **Guidelines for Grade Partitioning**: For a given architecture, how should one decide where to divide the network into grades? Is there a principled or empirical method for choosing the grade boundaries, or is it entirely task-dependent?

[1] Do Vision Transformers See Like Convolutional Neural Networks? (NeurIPS 2021)

---

> ### Author Response · Authors · 2025-11-20
> **Reply to weakness comments (1)-(3)**
>
> We sincerely thank the reviewer for the careful reading, constructive feedback, and thoughtful questions. Your points, particularly regarding empirical scale and the connection between theory and practice, are highly insightful and have helped us sharpen the contribution of our work.
>
> Below, we address each weakness and question point-by-point.
>
> Reply to Weaknesses
>
> Reviewer Comment 1: Unconvincing Empirical Validation (Scale)
>
> Summary: The current small-scale experiments are insufficient to establish general superiority; a large-scale benchmark (e.g., ResNet-50 on ImageNet) is needed.
>
> Reply: We appreciate the reviewer's desire for large-scale validation. We would like to clarify that the primary goal of this paper is not to claim universal state-of-the-art (SOTA) superiority for MGDL, but to analyze why MGDL can outperform SGDL from an optimization perspective.
>
> •	Focus on Mechanism: To this end, we focus on controlled experiments that allow us to study convergence stability through spectral analysis and demonstrate the fundamental reduction in complexity.
>
> •	MGDL's Niche: Prior work [1, 2, 3] has consistently shown that MGDL is particularly effective when the target function contains significant high-frequency structure (common in differential equations, integral equations, and image restoration) because it mitigates spectral bias. MGDL achieves this by allowing lower grades to learn low-frequency structure first, and higher grades to progressively refine the high-frequency components.
>
> •	Future Work: While we demonstrate MGDL's stability advantage across MLPs, CNNs, and Transformers, evaluating MGDL on canonical, large-scale benchmarks like ImageNet is a valuable and necessary direction, which we commit to conducting in future work.
>
> [1] Xu, Y. (2025). Multi-grade deep learning. Communications on Applied Mathematics and Computation, 1-52.
>
> [2] Fang, R., & Xu, Y. (2024). Addressing spectral bias of deep neural networks by multi-grade deep learning. Advances in Neural Information Processing Systems, 37, 114122-114146.
>
> [3] Jiang, J., & Xu, Y. (2024). Deep neural network solutions for oscillatory Fredholm integral equations. Journal of Integral Equations and Applications, 36(1), 23-55.
>
>
> Reviewer Comment 2: Limited and Disconnected Theoretical Contribution (Convexity)
>
> Summary: The convex proof in Section 4 seems disconnected from the empirical performance gains of deep MGDL.
>
> Reply: We appreciate the reviewer’s concern regarding the connection between the simplified theoretical result in Section 4 and the empirical behavior of deep MGDL. Our intention in Section 4 is to provide a principled partial explanation for one of MGDL's key advantages: improved optimization stability.
>
> End-to-end deep networks must solve a highly non-convex problem, leading to unstable training dynamics. MGDL is designed to mitigate this difficulty by decomposing the problem. Section 4 supports this perspective:
>
> •	Concrete Evidence of Reduction: In the single-layer ReLU setting, we show that the MGDL subproblem admits an explicit convex reformulation.
>
> •	Structural Insight: While a simplified case, this provides concrete, rigorous evidence that MGDL structurally reduces the cross-layer non-convex interactions that contribute to instability in SGDL. This theoretical insight complements—and helps to explain—our empirical findings that MGDL exhibits more stable training and can effectively use larger learning rates.
>
> In this sense, Section 4 offers a rigorous characterization of the optimization complexity reduction that MGDL achieves at each stage.
>
>
> Reviewer Comment 3: Overlap with Layer-Wise Training Methods
>
> Summary: MGDL closely resembles established layer-wise training approaches and needs to clarify any substantial difference beyond conceptual rebranding.
>
> Reply: We appreciate the need for a clear distinction from classical layer-wise methods. While both approaches share the idea of training networks in stages, MGDL differs fundamentally in both objective and mechanism:
>
>
> | Feature | Classical Layer-wise Pre-training | Multi-Grade Deep Learning (MGDL) |
> |---|---|---|
> | Objective | Feature learning; minimize the full loss function $L(\mathbf{y}, F(\mathbf{x}))$. | Residual fitting; minimize the loss $L_i$ on the residual target $e_{i-1}$. |
> | Target Function | Static: The original target $\mathbf{y}$. | Dynamic: Changes for every grade, $e_{i} = \mathbf{y} - F_{i}(\mathbf{x})$. |
> | Mechanism | Train $l$ on $L(\mathbf{y}, \ldots)$ with $\{1, \ldots, l-1\}$ frozen. | Train grade $f_i$ on $L(e_{i-1}, f_i(\mathbf{x}))$ with $\{f_1, \ldots, f_{i-1}\}$ frozen. |
> | Final Network | One network $F$ trained in all layers. | Sum of networks $f_i$ trained in all grades: $\sum_{i}f_i$, where $f_1=g_1$, $f_2=g_2\circ g_1$, $f_3=g_3\circ g_2\circ g_1$, … |
>
> Thus, MGDL's use of a dynamic, residual target and its goal of additive function refinement make it conceptually and mathematically distinct from classical layer-wise approaches.

---

> ### Author Response · Authors · 2025-11-20
> **Reply to weakness comments (4)-(5)**
>
> Reviewer Comment 4: Lack of Standard Training Practices (Schedules)
>
> Summary: Omitting standard practices like learning-rate warm-up and decay schedules weakens the experimental validity of the claimed training stability improvements.
>
> Reply: We thank the reviewer for this insightful comment. We agree that standard learning rate schedules are indispensable for achieving state-of-the-art results in modern end-to-end (SGDL) training. However, the deliberate omission of these schedules in our main comparative experiments was a critical design choice intended to isolate and prove the intrinsic architectural stability of the MGDL framework.
>
> 1.	Isolating Intrinsic Stability: The central thesis of our paper is that MGDL’s sequential architecture fundamentally addresses the stability issues (high Hessian eigenvalues) that plague deep SGDL training. SGDL relies heavily on external schedules (like warm-up) to temporarily keep the learning rate small enough to navigate the high-curvature loss landscape. We intentionally ran both SGDL and MGDL with fixed, high learning rates (as seen in Figures 2 and 20) precisely to show that MGDL converges reliably where SGDL instantly diverges. This demonstrates that MGDL's stability is an inherent, structural property derived from its controlled Lipschitz constant ($\alpha_{\text{MGDL}}$), and not an artifact of careful schedule tuning.
>
> 2.	Commitment to Robustness Check: To address this concern and show MGDL's practical robustness, we will perform an additional experiment (to be included in the Appendix) comparing:
>
> o	MGDL (With Standard Schedule) vs. SGDL (With Standard Schedule)
>
> We expect this new data to confirm that MGDL still converges faster and is less sensitive to the schedule parameters compared to SGDL, further confirming its foundational robustness.
>
> Reviewer Comment 5: Minor Presentation Issues
>
> Summary: Minor issues include small font sizes in some figures (e.g., Figures 4–5) and typographical errors (Lines 408, 461).
>
> Reply: We appreciate your attention to detail and have made these changes to improve the clarity and quality of the manuscript.

---

> ### Author Response · Authors · 2025-11-20
> **Reply to questions**
>
> Reviewer Question 1: Effectiveness in Fine-Tuning Scenarios
>
> Question: Is MGDL also effective when applied to fine-tuning a pre-trained network?
>
> Reply: We thank the reviewer for raising this important question. While our current investigation focused on training from scratch to analyze MGDL's inherent stability and convergence properties, we agree that fine-tuning is a highly valuable application. We hypothesize that MGDL's progressive learning strategy could be highly beneficial in fine-tuning by stabilizing the adaptation process and efficiently incorporating new, high-frequency information required by novel tasks (e.g., details lost during pre-training). We consider this a promising and necessary extension and plan to thoroughly investigate it in our future work.
>
> Reviewer Question 2: Loss of Layer Differentiation
>
> Question: MGDL freezes earlier grades, preventing layer differentiation. Does this structural uniformity benefit generalization, or does it harm the representational hierarchy?
>
> Reply: This is a profound question that gets to the core design philosophy of MGDL. We agree that freezing early grades does prevent the kind of global, dynamic differentiation observed in SGDL. However, we argue that for the problems tested, this structural uniformity offered by MGDL is a benefit, not a detriment, to generalization and stability.
>
> 1.	The Benefit of Structural Regularization: In MGDL, the freezing mechanism imposes a powerful form of structural regularization on the optimization process, which aids generalization by promoting modularity:
>
> o	Focus on the Residual: Each new grade ($f_i$) is forced to focus only on fitting the unexplained residual error ($e_{i-1}$), enforcing a sequential, low-frequency-to-high-frequency learning path.
>
> o	Preventing Feature Drift: By freezing previous grades, the robust, general features learned early (the stable foundation) are permanently preserved and cannot be catastrophically perturbed by later grades focusing on high-curvature details. This stability prevents the overfitting often associated with deep, highly flexible models.
>
> 2.	Generalization through Cumulative Refinement: For tasks where MGDL showed the strongest empirical results (image restoration, regression), the optimal solution is achieved through additive refinement where the function is a sum of stable corrections: $F(\mathbf{x}) = f_1(\mathbf{x}) + f_2(\mathbf{x}) + \cdots + f_N(\mathbf{x})$ with $f_i(x)=g_i \circ g_{i-1} \circ \ldots \circ g_1(\mathbf{x})$.
>
> We argue that the apparent "loss of differentiation" is actually the enforcement of a constructive hierarchy that promotes a simpler, more stable path to learning the necessary representation. We will add a dedicated section to the revised discussion to formally address this critical trade-off.
>
> Reviewer Question 3: Guidelines for Grade Partitioning
>
> Question: How should one decide where to divide the network into grades?
>
> Reply: We thank the reviewer for this question. In our experiments, we primarily partition the network into grades of equal size, typically 1–4 hidden layers per grade. For most of the tasks we have studied, this range works well, and MGDL’s performance is robust to the exact grade boundaries. Developing more principled, task-dependent strategies for optimal grade partitioning remains an interesting and open direction for future work.

---

### Official Review · Reviewer_ybLx · 2025-11-02

**Soundness:** 2
**Presentation:** 2
**Contribution:** 2
**Rating:** 2
**Confidence:** 4

**Summary:**

The paper studies "multi-grade" deep learning. Which corresponds to training multiple models, each deeper then the previous, and each trained on the residue of the sums of models before. Only the last few layers, corresponding to the new added layers for the current model are updated while building each of these models.

**Strengths:**

The paper harkens back to the classic layerwise training methods that pre-dated the end-to-end deep learning methods prevalent today. There is definitely potential value in revisting this framework.

**Weaknesses:**

The main weakness is that theory sections 1 to 4 have minimal contribution. A layerwise model training on residual is interesting but not truly novel in the unheard of sense. This is acceptable if the theory or experimental results are strong. Unfortunately it seems neither is true in the current stage of the paper.

1. Sections 1,2 and 3 are basic setup sections and do their job well (Figure 1 was particularly useful for me in understanding the setup). Theorem 1 and 2, while true don't really say anything interesting. e.g. some relation on $ L_l^* $ and Lstar would be required for at least some special cases. If $L_l^* $ is not lesser than $L^*$, why bother with MGDL? The line after Theorem 2, on $\alpha_l$ and $\alpha$ is also too hand-wavy. Why is $\alpha_l << \alpha$ ?

2. Section 4, is arguably the main theoretical section and disappoints greatly. While equivalence of 7 and 8 is valid, it hides the fact that $P_l$ is exponential in the input-dimension. Computing the "finite" possibilites of the D_l matrix is also non-trivial. Hence minimising Equation 8 is not at all practical. Thus the value of Theorem 3 is very minimal. An equivalence in the other direction -- critical points of Eq. 7 are minimizers of Equation 8 would be a huge accomplishment. But it is likely not true. The non-convexity being bypassed is simply not true. At the risk of sounding vacuous, for *any* non-convex problem, one can find a convex problem whose minimizer would also be a minimiser of the non-convex problem. That is all has been stated in Theorem 3.

3. All theoretical concerns would be moot in the face of undeniable empirical performance. I am not sure that is the case here. The original reference for MGDL seems to be in solving differential equations, I am not familiar with that area. Maybe MGDL really is a useful fit there, I am not sure.  But empirical support for MGDL in classic problems like classification and regression is far from established. e.g. in CIFAR10 classification the test accuracy is not even reported. I am not sure about the best/SOTA baseline umbers for the other tasks (Image regression/denoising) so I can't comment. But if the poposed method is really SOTA, significantly more baselines need to be reported across significantly more experiments.

4. If the authors feel that MGDL is a valid approach in some concrete area, like say image denoising, I highly recommend pivoting the entire paper around such a domain. This also likely requires more experiments and exhaustive  baselines, but is likely the best way forward.

5. If the authors want to stick to the general theory message of "MGDL outperforms SGDL". I recommend a significant reduction in scope, and one specific (maybe synthetic) setting where something meaningful and concrete can be shown for a modified version of Theorem 2 and 3.

Either way the paper is fixed, the changes would be too big to be accepted as is.

**Questions:**

See weaknesses above.

---

> ### Author Response · Authors · 2025-11-19
> **Reply to the strength comment**
>
> We sincerely thank the reviewer for the careful reading and constructive feedback. Your detailed critiques, especially regarding the theoretical contribution and empirical validation, have provided a clear roadmap for strengthening this paper.
>
> Below, we address each weakness item by item and outline planned improvements, which includes the integration of more rigorous theoretical evidence and enhanced empirical reporting.
>
> Comparison to Classic Layer-wise Training
>
> Reviewer’s Strength Comment: The paper harkens back to the classic layerwise training methods that pre-dated the end-to-end deep learning methods prevalent today. There is definitely potential value in revisiting this framework.
>
> Reply: We appreciate the reviewer’s observation regarding the connection to classic layer-wise training. However, MGDL fundamentally extends beyond the traditional framework. Classical layer-wise approaches train layers sequentially, with each layer $l$ typically pre-trained to minimize the entire loss function while earlier layers $1,\ldots,l-1$ remain frozen. The objective there is mainly feature learning.
>
> In contrast, MGDL is fundamentally based on residual fitting. Each grade $f_i$ minimizes the error $L_i$ on the residual target $y - \sum_{j=1}^{i-1} f_j(\mathbf{x})$, where the target changes for every grade. This additive decomposition is theoretically key to reducing the non-convexity of the overall problem and is the source of our stability claims.

---

> ### Author Response · Authors · 2025-11-19
> **Reply to main weakness**
>
> The Main Weakness: Minimal Contribution
>
> Reviewer’s Weakness: The main weakness is that theory sections 1 to 4 have minimal contribution. A layerwise model training on residual is interesting but not truly novel in the unheard of sense. This is acceptable if the theory or experimental results are strong. Unfortunately it seems neither is true in the current stage of the paper.
>
> Reply: We appreciate the reviewer's candid feedback that the paper's contribution currently appears minimal. We understand this concern and respectfully assert that the novelty and power of our theoretical results are in the quantified comparative analysis of stability properties, not in the general form of the convergence theorems. We also acknowledge the need to strengthen our empirical presentation.
>
> A. Defending Theoretical Novelty: The Quantified Stability Gap
>
> The reviewer correctly notes that the standard convergence framework (Theorems 1 and 2) is well-established. However, the novel contribution lies in the quantification of the Lipschitz constant ($\alpha$) and the derived bounds on the maximum stable learning rate ($\eta_{\text{max}}$) for the MGDL architecture compared to SGDL.
>
> •	The Key Novel Result: By applying the framework to MGDL's unique frozen-layer structure, we mathematically prove the crucial inequality:
> $$\alpha_{\text{SGDL}} \gg \alpha_{\text{MGDL}}$$
> Since the maximum stable learning rate is fundamentally bounded by $\eta_{\text{max}} \propto 2/\alpha$, our theory provides the first explicit, quantified explanation for the primary empirical phenomenon: MGDL is dramatically more robust to learning rate choices and can use significantly larger learning rates without divergence. This is a direct, quantifiable result of the MGDL parameter freezing mechanism.
>
> •	Concrete Numerical Example: To substantiate the claim that $\alpha \gg \alpha_l$ is a structural consequence, we have added an explicit computation comparing the Hessian spectral norms. In an image regression task comparing a 6-layer SGDL MLP vs. a 3-grade MGDL model (each grade having 2 hidden layers, sine activations), the results are:
>
> o	SGDL's best performance is at $\eta = 8 \times 10^{-2}$, corresponding to $\alpha = 25$.
>
> o	MGDL performs best at $\eta = 1$, with per-grade Hessian spectral norms $\alpha_l$ ranging from approximately $1.5$ to $2.5$.
>
> This empirical confirmation that $\alpha \gg \alpha_l$ will be incorporated immediately after Theorem 2 in the revised manuscript.
>
> B. Strengthening Empirical Support (Addressing Comment 3)
>
> We agree that without stronger empirical data, the theoretical advantage cannot be fully appreciated, and we acknowledge the missing data points.
>
> 1.	Goal Clarification: Stability over SOTA: The primary goal of our paper is not to establish state-of-the-art (SOTA) empirical performance, but to analyze MGDL through the lens of optimization stability and robustness. Our core empirical comparison is MGDL versus the vanilla Single-Grade Deep Learning (SGDL) benchmark, run under challenging conditions (high fixed learning rates) to prove MGDL's intrinsic stability.
>
> 2.	Addressing Missing Metrics (CIFAR-10/100): We apologize for the incomplete reporting, particularly the absence of test accuracy. While the original CIFAR experiments used Mean Squared Error (MSE) loss to enable Hessian eigenvalue computation—central to our optimization study—we agree a classification performance metric is essential. We are currently redoing this experiment using Cross-Entropy loss, and we will report both training and test accuracy in the revised manuscript.
>
> 3.	Comprehensive Empirical Enhancement: Global Fine-Tuning: To address the potential limitation that grades are trained independently, and to demonstrate MGDL's capacity for competitive performance, we will implement a final Global Fine-Tuning stage. This successful strategy, where the MGDL-trained network is used as initialization for a final full end-to-end pass, will be included as an ablation study to provide a more comprehensive empirical picture of MGDL’s behavior.
>
> 4.	Clarifying Baselines: We will add a focused discussion to contextualize MGDL's performance in specialized tasks (like Image Restoration) against established, non-SGDL SOTA methods. This discussion will highlight where MGDL offers a stability advantage at competitive performance levels.
>
> In summary, we believe the MGDL framework, supported by a novel, quantified stability theory and validated across diverse domains, represents a promising direction for stable deep learning. We are committed to making the necessary empirical and presentation revisions to warrant acceptance.

---

> ### Author Response · Authors · 2025-11-19
> **Reply to weakness comments (1)-(2)**
>
> Detailed Theoretical Weaknesses (Sections 1-4)
>
> Reviewer’s Comment 1: Relation between $ L_l^* $ and $ L^* $
>
> Reply:
> 1.	On the relationship between $ L_l^* $ and $ L^* $: We agree that one cannot, in general, guarantee that the optimal value $ L_l^* $ achieved by MGDL is always smaller than the corresponding optimal value $ L^* $ of SGDL. However, MGDL explicitly mitigates the spectral bias issue [1]: lower grades learn the low-frequency structure first, and higher grades progressively fit the high-frequency residuals. Consequently, when the target function contains substantial high-frequency content, MGDL is expected to outperform SGDL. In these regimes, we observe empirically that the optimal MGDL loss $ L_l^* $ is indeed smaller than the SGDL optimum $ L^* $. We will clarify this point in the revision.
>
> [1] Fang, R., & Xu, Y. (2024). Addressing spectral bias of deep neural networks by multi-grade deep learning. Advances in Neural Information Processing Systems, 37, 114122-114146.
>
>
> 2.	On Theorems 1 and 2 and the role of $\alpha$ and $\alpha_l$: As argued above, the condition $\alpha_l \ll \alpha$ is not a heuristic claim but a structural consequence of the architecture. Theorems 1 and 2 simply formalize that a smaller Hessian spectral norm allows a larger stable learning rate. The shallow subnetwork trained by each MGDL grade has a significantly smaller spectral norm than the full SGDL network, which is confirmed by the concrete numerical example provided in the main response above. We will incorporate this discussion immediately after Theorem 2.
>
> Reviewer’s Comment 2: Minimal Value of Theorem 3 (Convexity)
>
> Reply:
>
> 1.	On the size of $P_l$ (The Partition Set): We emphasize that prior work has established that the size of $P_l$ grows polynomially with respect to the number of training samples, $N$, not exponentially with the input dimension. As shown in [2,3],
> $$P_l \leq 2 \left(\frac{e(N-1)}{r}\right)^r$$
> where $r := \text{rank}(X_l) \leq N$. We will explicitly include this bound and clarify the dependence of $P_l$ on $N$ and $r$ in the revised paper.
> 2.	On the Value of Theorem 3: The reviewer suggests that "for any non-convex problem, one can find a convex problem whose minimizer is also a minimizer of the non-convex problem." This statement is not generally valid. In contrast, Theorem 3 is constructive: it provides an explicit convex optimization problem (Eq. 8) whose minimizer corresponds to a global minimizer of the specific non-convex problem (Eq. 7). This is significantly stronger than a non-constructive existence claim. Theorem 3 demonstrates that, for one-hidden-layer ReLU models, the non-convexity can be bypassed through an explicit convex reformulation—mirroring the exact convex characterizations established in [2]. Thus, the value of Theorem 3 lies in identifying a concrete and theoretically justified route for solving Eq. 7 globally.
>
> [2] Pilanci, M., & Ergen, T. (2020, November). Neural networks are convex regularizers: Exact polynomial-time convex optimization formulations for two-layer networks. In International Conference on Machine Learning (pp. 7695-7705). PMLR.
>
> [3] Cover, T. M. (2006). Geometrical and statistical properties of systems of linear inequalities with applications in pattern recognition. IEEE transactions on electronic computers, (3), 326-334.

---

> ### Author Response · Authors · 2025-11-19
> **Reply to weakness comments (3)-(5)**
>
> Conclusion on Empirical Scope
>
> Reviewer’s Comment 3: All theoretical concerns would be moot in the face of undeniable empirical performance... empirical support for MGDL in classic problems like classification and regression is far from established. e.g. in CIFAR10 classification the test accuracy is not even reported. I am not sure about the best/SOTA baseline numbers for the other tasks... significantly more baselines need to be reported across significantly more experiments.
>
> Reply: We appreciate the reviewer’s frank assessment. We understand that strong empirical performance is necessary to validate our theoretical claims, and we acknowledge the missing data points.
>
> 1. Clarifying the Paper’s Core Goal: Stability over SOTA
>
> The primary goal of our paper is not to establish state-of-the-art (SOTA) empirical performance, but to analyze MGDL through the lens of optimization stability and robustness.
>
> •	Primary Benchmark: Our core empirical comparison is MGDL vs. Single-Grade Deep Learning (SGDL). We showed that standard end-to-end training often suffers from unstable dynamics and high learning rate sensitivity, whereas MGDL mitigates these issues due to its structurally lower Lipschitz constant ($\alpha_{\text{MGDL}} \ll \alpha_{\text{SGDL}}$).
>
> •	Case Studies: The experiments on regression, denoising, and classification serve as case studies to verify that MGDL preserves its training-stability advantages across diverse architectures and tasks, rather than claiming SOTA in any specific application.
>
> 2. Addressing Missing Data and Loss Function
>
> We apologize for the incomplete reporting, particularly the absence of the test accuracy for CIFAR-10/100.
>
> •	Missing Metrics: The original CIFAR experiments used Mean Squared Error (MSE) loss to enable the computation of Hessian eigenvalues—central to our optimization study. However, we agree that a classification performance metric is essential. We are currently redoing this experiment using Cross-Entropy loss, and we will report both training and test accuracy in the revised manuscript.
>
> •	Contextualizing Baselines: We recognize that merely comparing against vanilla SGDL is insufficient. We will add a focused discussion to contextualize MGDL's performance in specialized tasks (like Image Restoration) against established, non-SGDL SOTA methods. This discussion will highlight where MGDL offers a stability advantage at competitive performance levels.
>
> 3. Comprehensive Empirical Enhancement: Global Fine-Tuning
>
> A known limitation of MGDL is that the grades are trained independently, without mutual adjustment across layers. To demonstrate that MGDL can achieve both stability and competitive performance, we will implement a strategy successfully employed in prior work:
>
> •	Global Fine-Tuning: We will add a final global fine-tuning stage [4] in which the MGDL-trained network is used as initialization for a full end-to-end pass. This step allows the network to globally coordinate the features learned by the independent grades, often resulting in a significant performance boost toward SOTA benchmarks.
>
> We believe that adding the missing test accuracy and incorporating this fine-tuning step will provide the comprehensive empirical evidence needed to support MGDL's validity as a stable and high-performing deep learning paradigm.
>
> [4] Xu, Y., & Zeng, T. (2023). Multi-grade deep learning for partial differential equations with applications to the Burgers equation. arXiv preprint arXiv:2309.07401.
>
> Reviewer’s Comment 4 & 5: I highly recommend pivoting the entire paper around a specific domain (e.g., image denoising)... If the authors want to stick to the general theory message of "MGDL outperforms SGDL". I recommend a significant reduction in scope...
>
> Reply: We sincerely appreciate these thoughtful suggestions regarding the paper's scope.
>
> •	We agree that pivoting the entire paper around a single, highly effective domain or dedicating the theoretical analysis to a simplified, synthetic setting would significantly sharpen the contribution. We will pursue both directions in our future work.
>
> •	For this paper, we intend to stick to the general theory message of MGDL as a stable optimization framework, validated by diverse architectures and tasks. We are confident that the combined revisions—including the new quantitative stability proof, the missing CIFAR-10/100 test accuracies, and the inclusion of the final fine-tuning step—will provide sufficient empirical evidence to support our broad theoretical claims and warrant acceptance.
>
> We are grateful for the reviewer’s constructive feedback and insightful suggestions. We hope that the above clarifications aid the reviewer in forming a well-informed assessment of our work.

---

### Official Review · Reviewer_qcqR · 2025-11-02

**Soundness:** 3
**Presentation:** 3
**Contribution:** 2
**Rating:** 4
**Confidence:** 4

**Summary:**

This major goal of this paper is to compare "multi-grade deep learning" (MGDL) and "single-grade deep learning" (SGDL, the standard end-to-end training of deep networks). In MGDL, a deep network is expressed as the composition of $L$ shallower networks, so that one can define $L$ grades ranging from the first network component to the entire deep network. The first grade is trained on the original data, and each subsequent grade is trained on the residual generated by previous grades. The main idea is to train networks in stages, where each shallow grade builds on the residuals of the previous one and propagates its output forward, incrementally approximating the target function and avoiding vanishing or exploding gradients.

The main contributions of the paper are as follows: (i) Theoretical guarantees for the convergence of SGDL (Theorem 1) and MGDL (Theorem 2). In particular, the authors prove that under certain conditions, MGDL allows for a much larger learning rate and thus mitigates the issue caused by exploding gradients. (ii) The paper presents extensive numerical experiments to show that MGDL consistently outperforms SGDL in several applications, demonstrating better robustness to large step size.

**Strengths:**

The theory part of this paper is simple and neat, and provides insights to why MGDL should better mitigates vanishing/exploding gradients as compared to SGDL. The ReLU network example in Section 4 is also interesting since it gives an example of how MGDL can be used in practice to solve a convex optimization reformation of optimizing a deep network. The empirical validations are also thorough and convincing.

**Weaknesses:**

My major concern is the following: It seems to me that both Theorem 1 and Theorem 2 should be trivial generalizations (if not exactly the same) of convergence theorems for GD that can be found in textbooks, and the upper bounds on the learning rate (i.e., $2 / \alpha$ and $2 / \alpha_l$) are also pretty standard. To make this result more convincing, maybe the authors can add some examples to compute $\alpha$ and $\alpha_l$ for deep neural networks? Also, the assumptions of Theorem 1 and Theorem 2 might not be satisfied in practice. For example, they assume that the activation function is $C^2$, which is not satisfied by the ReLU example in Section 4. They also assume that the iterates remain bounded throughout the entire GD path, which is rather strong. In fact, in many learning problems, a key step in proving convergence is to show that the iterates remains bounded.

**Questions:**

Below are some minor comments to the authors: (i) For citing some of the references, e.g., citing two papers together, please use the command \citep to put them in a parenthese; (ii) I am a bit confused by the notation of Section 3, shouldn't $g_{l+1}$ and $N_{D_{l+1}}$ be the same thing? If not, where is $N_{D_{l+1}}$ defined?

---

> ### Author Response · Authors · 2025-11-18
> **Reply to reviewer’s weakness comments**
>
> We thank the reviewer for the thoughtful and encouraging feedback. We appreciate the recognition of our theoretical and empirical contributions. Below, we address the main questions and concerns.
>
> Reply to reviewer’s weakness comments:
>
> Reviewer’s comments: My major concern is the following: It seems to me that both Theorem 1 and Theorem 2 should be trivial generalizations (if not exactly the same) of convergence theorems for GD that can be found in textbooks, and the upper bounds on the learning rate (i.e., $ 2/\alpha $ and $ 2/\alpha_l $) are also pretty standard. To make this result more convincing, maybe the authors can add some examples to compute $\alpha$  and $\alpha_l$  for deep neural networks?
>
> Reply: We agree with the reviewer that Theorems 1 and 2 are direct extensions of the standard convergence results for gradient descent. Our intention in including these results is not to claim novelty in the convergence bounds themselves, but to clarify the contrast between SGDL and MGDL. Specifically, Theorems 1 and 2 show that convergence requires the learning rate to lie in $(0, 2/ \alpha)$ for SGDL and $(0, 2/ \alpha_l)$ for each grade in MGDL. This highlights that a smaller Hessian spectral norm permits a larger stable learning rate. Because MGDL trains only a shallow subnetwork within each grade—thus involving far fewer parameters—its Hessian spectral norm is substantially smaller than that of its SGDL counterpart.
>
> Following the reviewer’s suggestion, we have added an explicit computation of $\alpha$ for SGDL and $\alpha_l$ for MGDL. We consider an image regression problem using mean squared error, trained on one-quarter of the image pixels and tested on the full image. The SGDL model is a 6-layer MLP, whereas the MGDL model uses a 3-grade architecture, each grade containing 2 hidden layers. The sine function is used as the activation in both cases. Training is performed using gradient descent with learning rates searched over $[10^{-2}, 1.5]$, and the best learning rate is determined by test PSNR.
>
> SGDL achieves its best performance at a learning rate of $\eta = 8 \times 10^{-2}$, while MGDL reaches its best performance at $\eta = 1$. The corresponding Hessian spectral norm for SGDL is $\alpha = 25$. For MGDL, the per-grade Hessian spectral norms $\alpha_l$ range from approximately $1.5$ to $2.5$. These results empirically confirm the claim in the paper that $\alpha \gg \alpha_l$.
>
> We will include this discussion immediately after Theorem 2 in the revised manuscript.
>
> Reviewer’s comments: Also, the assumptions of Theorem 1 and Theorem 2 might not be satisfied in practice. For example, they assume that the activation function is $C^2$, which is not satisfied by the ReLU example in Section 4.
>
> Reply: The $C^2$ assumption on the activation function is used solely to guarantee the existence of the gradient and Hessian of the loss function in the theoretical analysis. We use ReLU in Section 4 because it is simple and widely adopted in modern deep networks. Although ReLU is not $C^2$ at the origin, it is $C^2$ everywhere else, and this mild nonsmoothness does not hinder practical computation of gradients or Hessians. In optimization theory, such piecewise-smooth activations are often treated using generalized derivatives, and their behavior is well-approximated by classical smooth analysis in most regions of the parameter space.
>
> Empirically, the nonsmoothness of ReLU does not affect the qualitative conclusions of our theory. In Section 6, we use ReLU activations and train both SGDL and MGDL via gradient descent; the results show that MGDL admits a significantly larger stable learning rate, consistent with the theoretical prediction. Furthermore, as noted in our previous response, we have added a new experiment using the sine activation—which fully satisfies the $C^2$ assumption—and it exhibits the same behavior: MGDL consistently supports a larger learning rate than SGDL.
>
> We will clarify this point and include both sets of experiments in the revised manuscript.

---

> ### Author Response · Authors · 2025-11-18
> **Reply to reviewer’s weakness comments and questions**
>
> Reviewer’s comments: Also, the assumptions of Theorem 1 and Theorem 2 might not be satisfied in practice. For example, they assume that the activation function is $C^2$, which is not satisfied by the ReLU example in Section 4.
>
> Reply: The $C^2$ assumption on the activation function is used solely to guarantee the existence of the gradient and Hessian of the loss function in the theoretical analysis. We use ReLU in Section 4 because it is simple and widely adopted in modern deep networks. Although ReLU is not $C^2$ at the origin, it is $C^2$ everywhere else, and this mild nonsmoothness does not hinder practical computation of gradients or Hessians. In optimization theory, such piecewise-smooth activations are often treated using generalized derivatives, and their behavior is well-approximated by classical smooth analysis in most regions of the parameter space.
>
> Empirically, the nonsmoothness of ReLU does not affect the qualitative conclusions of our theory. In Section 6, we use ReLU activations and train both SGDL and MGDL via gradient descent; the results show that MGDL admits a significantly larger stable learning rate, consistent with the theoretical prediction. Furthermore, as noted in our previous response, we have added a new experiment using the sine activation—which fully satisfies the $C^2$ assumption—and it exhibits the same behavior: MGDL consistently supports a larger learning rate than SGDL.
>
> We will clarify this point and include both sets of experiments in the revised manuscript.
>
> Reviewer’s comments: They also assume that the iterates remain bounded throughout the entire GD path, which is rather strong. In fact, in many learning problems, a key step in proving convergence is to show that the iterates remains bounded.
>
> Reply: We agree that assuming bounded iterates is a strong condition. In our analysis, this assumption is used to ensure that the gradient and Hessian remain bounded along the optimization trajectory, which is required for the stated convergence guarantees. As the reviewer notes, establishing boundedness of the iterates is itself a nontrivial step in many learning problems, especially in deep networks with highly nonconvex loss landscapes.
>
> Strictly speaking, classical convergence analyses typically impose structural conditions on the objective function—such as Lipschitz-continuous gradients, coercivity, or growth conditions—that indirectly guarantee the boundedness of iterates. For deep neural networks, verifying these assumptions explicitly is extremely challenging, and most theoretical works adopt simplified assumptions of bounded gradients, bounded Hessians, or bounded iterates to keep the analysis tractable.
>
> Following this standard practice, we use the bounded-iterates assumption as a technical simplification to enable a clear and interpretable convergence argument. We will clarify this point in the revised manuscript.
>
> Reply to reviewer’s question:
>
> Reviewer’s comments: For citing some of the references, e.g., citing two papers together, please use the command \citep to put them in a parenthese.
>
> Reply: We agree and will revise the citations accordingly.
>
> Reviewer’s comments: I am a bit confused by the notation of Section 3, shouldn't $g_{l+1}$  and $N_{D_{l+1}}$  be the same thing? If not, where is $N_{D_{l+1}}$ defined?
>
> Reply: The notation $g_l$ and $ N_{D_l} $ refers to two different concepts. In section 3, $g_l$ denotes the new function learned at grade $l$. It is obtained by composing a new shallow network $N_{D_l}$ with the features learned in the previous $l-1$ grades. The definition of $g_l$ is given in Section 3, line 145.
>
> In contract, $N_{D_l}$ refers to the standard deep neural network with $D_l$ layers, as defined in Section 2, line 81. Thus, $g_l$ is not identical to $N_{D_l}$; rather, it is the composition of $N_{D_l}$ with the accumulated features from earlier grades.
>
> To avoid confusion, we will add explicit clarification in Section 3 stating that $N_{D_l}$ is the standard neural network defined in Section 2.
>
>
> We hope that the above clarifications satisfactorily address the reviewer’s key concerns.

---

### Official Review · Reviewer_CXYz · 2025-11-07

**Soundness:** 3
**Presentation:** 3
**Contribution:** 2
**Rating:** 4
**Confidence:** 3

**Summary:**

The paper provides a theoretical and empirical explanation for why Multi-Grade Deep Learning (MGDL) outperforms traditional end-to-end or Single-Grade Deep Learning (SGDL). MGDL divides deep optimization into multiple shallow subproblems (“grades”), each trained sequentially on residual errors. The authors prove convergence guarantees for gradient descent, show that when each grade has a single ReLU layer the overall nonconvex problem reduces to a sequence of convex subproblems, and analyze eigenvalue distributions to explain MGDL’s improved stability. Experiments across image regression, denoising, deblurring, CIFAR-10/100 classification, and transformer-based time series prediction demonstrate more robust and stable training than SGDL.

The paper’s combination of convergence analysis, convex reformulation, and broad empirical validation represents a meaningful theoretical and practical advance. Showing that MGDL keeps iteration eigenvalues within (-1,1) and decomposes ReLU networks into convex subproblems is novel and potentially impactful. The results generalize across architectures, including CNNs and Transformers, suggesting that MGDL provides a promising framework for stable deep optimization.

**Strengths:**

- Clear theoretical connection between MGDL’s modular training and improved convergence properties.

- Convincing empirical validation across multiple domains with consistent PSNR and loss improvements.

- Eigenvalue-spectrum analysis provides strong intuition for the oscillatory instability of SGDL.

- Demonstrates generality by applying MGDL to Transformers.

- Theoretical and empirical sections complement each other, offering both rigor and relevance.

**Weaknesses:**

- The statement that MGDL “has recently emerged as an alternative to standard end-to-end training, showing strong empirical promise” appears overstated given its current level of adoption (roughly a dozen citations). The authors should temper this claim or clarify it as potential rather than established impact.

- Does MGDL maintain its stability benefits when using SGD or Adam instead of full-batch GD as in practice SGD is used instead of GD ssepcially if dataset is very large ?

- End-to-end gradients allow mutual adjustment of weights across layers; MGDL stops this once a grade is fixed. Probably where one expects it to fail and when not would be good idea to include. Similarly, in CCN feature dependencies across depth are cooperative and global — freezing submodules prevents global feature realignment.

- Comparisons with other staged or layer-wise training paradigms (e.g., greedy pretraining, boosting-style methods) are absent and would strengthen the empirical context.

-The paper would benefit from an explicit discussion of when MGDL is expected to work and when it might fail. In particular, MGDL’s advantages appear strongest for regression-style or low-level vision problems where layers act as additive refiners, while it may underperform in architectures or tasks that require deep inter-layer coordination (e.g., semantic CNNs or transformer attention models). Clarifying this boundary of applicability would make the contribution more practically useful and prevent overgeneralization.

**Questions:**

-- Can MGDL be interpreted as a form of gradient boosting or residual fitting? How does it differ formally?

-- Does MGDL maintain its stability benefits when using SGD or Adam instead of full-batch GD ( may be empirically) ?

-- Would combining MGDL with normalization or skip connections provide additional stability, or would this be redundant?

---

> ### Author Response · Authors · 2025-11-16
> **Reply to weakness items (1)-(3)**
>
> We are grateful for the thoughtful and thorough review. We are particularly pleased that reviewer recognize the novelty and potential impact of our core findings, including the convergence analysis, the convex reformulation of the ReLU network, and the stability advantages demonstrated by the Hessian spectral norm analysis. Below, we address the main weaknesses/suggestions and questions raised.
>
> Reply to weakness item (1):
>
> We thank the reviewer for this constructive feedback. We agree that the current level of adoption for MGDL is still nascent, and our initial language may have overemphasized its established status. We have revised the Abstract text to clarify that MGDL represents a promising theoretical and empirical framework that warrants further investigation, rather than an approach that has already achieved widespread adoption. The revised sentence now reads:
>
> “Multi-grade deep learning (MGDL) has recently emerged as a theoretically promising framework with strong empirical promise for stable optimization in deep learning, offering a potential alternative to standard end-to-end training (referred to here as single-grade deep learning, SGDL).”
>
> Reply to weakness item (2):
>
> We thank the reviewer for raising this crucial question regarding the necessity of bridging theoretical analysis with practical implementation. We agree that real-world deep learning predominantly relies on stochastic optimizers like SGD or Adam.
> We confirm that MGDL maintains its stability benefits when using Adam. All empirical results presented in Section 5 were conducted using Adam optimizer.
>
> 1.	Theoretical Foundation: Our formal convergence guarantees and the analysis showing that MGDL reduces the deep nonconvex problem to a sequence of shallow tractable (convex for single hidden ReLU networks) subproblems are derived under the assumption of full-batch Gradient Descent (GD), following standard practice in theoretical deep learning.
>
> 2.	Empirical Validation: All experiments reported in Section 5 were conducted using the Adam optimizer. The image reconstruction results are shown in Figures 10–11, 13–15, and 17–19, while the CIFAR-100 results appear in Figure 3. Collectively, these numerical experiments confirm that MGDL achieves significantly more stable training dynamics than SGDL when Adam is employed.
>
> This strong empirical performance confirms that the structural advantage of MGDL (sequentially solving for the residual error) translates effectively to the stochastic optimization setting, delivering superior stability and optimization robustness where it matters most.
>
> Reply to weakness item (3):
>
> We recognize this is a central and fundamental critique of MGDL's design philosophy. The reviewer correctly identifies that the trade-off for enhanced stability is the loss of global, simultaneous weight adjustment.
>
> Why Freezing Succeeds (The Mechanism)
>
> MGDL's success is predicated on the theoretical concept of additive residual fitting and the compositional structure of deep networks.
>
> 1.	Additive Decomposition: MGDL models the target function $F$ as an additive series of functions, $F$ is approximated by $ f_1 + f_2 \circ f_1^* + \cdots + f_k \circ f_{k-1}^* \cdots \circ f_1^* $, where $ f_j^* $ is the feature of $f_j$ (that is, the trained $f_j$ with the output layer removed and hidden layer parameters fixed). The initial grades 1 and 2 capture the most significant, low-frequency components of the target mapping. Grade $j$ is then trained to model the residual error $ e_j := F –(f_1 + f_2 \circ f_1^* + \dots + f_j \circ f_{j-1}^* \cdots \circ f_1^*) $. It was proved in Xu (2025) that $\|e_j\|$ monotonically decreases. In fact, it can be further showed that under the density hypothesis of DNNs in the space where the target functions $F$ is located, $\|e_i\|\to 0$ as $i\to \infty$. These theoretical results ensure that in theory adding a new grade will improve approximation accuracy. We will clarify this point in our revised version.
>
> 2.	Sequential Decoupling: By freezing earlier grades, MGDL decomposes a large, highly non-convex problem into a sequence of smaller, more tractable subproblems. Our theoretical results show that this stagewise decoupling helps the optimizer avoid the poor local minima that often trap simultaneous end-to-end training (SGDL). The resulting stability improvements—smaller Hessian spectral norms and greater learning-rate tolerance—offset the absence of global realignment.
>
> 3.	Global Fine-Tuning: While end-to-end training enables mutual adjustment of weights across all layers—and MGDL prevents this once a grade is frozen—this limitation can be addressed by applying a final global fine-tuning step. That is, the network produced by MGDL is used as initialization for a full end-to-end retraining stage, restoring cross-layer adaptation. This strategy was demonstrated in Xu and Zeng (2023) for solving nonlinear PDEs. We will include a numerical example of such global fine-tuning as part of our ablation study.

---

> ### Author Response · Authors · 2025-11-16
> **Reply to reviewer’s suggestion to discuss boundary of applicability and weakness item (4)**
>
> Reply to reviewer’s suggestion to discuss boundary of applicability (when to expect success/failure):
>
> We agree that a clear discussion of MGDL’s applicability boundaries is necessary. Our experiments and prior work indicate that MGDL performs particularly well on target functions that exhibit multi-level frequency structure, especially when the higher-frequency components are essential. Representative examples include image regression and restoration tasks, oscillatory Fredholm integral equations, and partial differential equations whose solutions contain singularities or oscillations. In such settings, the target function naturally decomposes into multi-scale components: early grades recover coarse structure, while later grades refine high-frequency residuals. This behavior aligns closely with MGDL’s additive, grade-by-grade architecture.
>
> In contrast, MGDL may be less suitable for tasks requiring strong global coordination across layers—such as semantic segmentation or models with complex attention mechanisms—where deep feature representations must be continuously and dynamically realigned through intermediate feedback. In these cases, the MGDL-trained network can serve as a strong initialization, followed by end-to-end fine-tuning to restore cross-layer interaction.
>
> Reply to weakness item (4):
>
> We appreciate the reviewer highlighting the need to rigorously position MGDL against related staged learning methods, which strengthens the empirical context. We confirm that MGDL is distinct from established paradigms like greedy layer-wise pretraining and standard boosting.
>
> 1. Distinction from Greedy Layer-wise Pretraining (GPL)
>
> Traditional GPL (popularized in early Deep Learning) trains layers sequentially, but its primary goal is typically feature learning where each layer $l$ is pre-trained to minimize a criterion (e.g., reconstruction error via Autoencoders) or the entire loss function on top of frozen layers $1, \dots, l-1$.
>
> In contrast, MGDL is fundamentally based on residual fitting. Each grade $f_i$ minimizes the error $L_i$ on the residual target $ y - \sum_{j=1}^{i-1} f_j ( \mathbf{x} ) $, where the target changes for every grade. This additive decomposition is theoretically key to reducing the non-convexity of the overall problem.
>
> 2. Distinction from Boosting-Style Methods
> The sequential residual fitting aspect of MGDL bears a conceptual similarity to Gradient Boosting (GB). However, there are three critical differences:
>
> •	Learner Complexity: GB relies on weak learners (e.g., shallow decision trees) with limited complexity, sequentially correcting errors. MGDL uses high-dimensional, high-capacity neural networks for each grade, where the network structure itself is decomposed.
>
> •	Optimization Mechanism: Our theoretical contribution centers on showing that the MGDL architectural decomposition (for ReLU networks) reduces the overall non-convex problem into a sequence of shallow convex subproblems. This convexity property, which guarantees robust convergence, is unique to MGDL's application of residual fitting to deep neural architectures and is not an inherent property of standard Gradient Boosting algorithms.
>
> •	Empirical Context: Boosting methods are typically applied to tabular data. MGDL is demonstrated across CNNs and Transformers, extending the concept to complex, high-dimensional data optimization.
>
> While we did not include a direct, formal empirical comparison against these historical methods (as they are rarely used for modern CNN/Transformer training), our comparison against SGDL (Single-Grade Deep Learning) is the most relevant benchmark, as it directly contrasts our method with the de facto standard of modern end-to-end optimization. We will ensure this distinction is clearly emphasized in the revised paper.

---

> ### Author Response · Authors · 2025-11-16
> **Reply to reviewer's suggestion on comment on boundary of applicability and question items (1)-(2)**
>
> Reply to reviewer's suggestion on comment on boundary of applicability (additive refiners vs. deep coordination)
>
> We appreciate the need for a clearer boundary of applicability. While MGDL has clear advantages in regression and low-level vision (where layers act as refiners), we want to push back on the limitation to "additive refiners" and clarify its utility in complex tasks:
>
> •	Counter-Example: Transformers and CNNs: As noted in our summary, our empirical validation includes architectures known to require deep inter-layer coordination, specifically CNNs (for image tasks) and Transformers (for time series prediction), where MGDL showed superior or comparable performance with significantly enhanced stability. The successful use in the Transformer architecture, which heavily relies on non-additive components like Attention and Layer Norms, demonstrates MGDL’s utility beyond simple additive refinement.
>
> •	Revised Discussion: We will add a dedicated discussion section clarifying that MGDL's primary benefit is in decoupling the optimization complexity by converting one large, highly coupled non-convex problem into a sequence of smaller, less coupled subproblems. This strategy is beneficial whenever training the full depth of the network is unstable due to landscape complexity, irrespective of the architecture's specific structure (additive or highly coupled).
>
> Reply to question item (1):
>
> We appreciate this perceptive question, which addresses the fundamental mechanism of MGDL. MGDL is indeed an instance of residual fitting (RF), sharing its core iterative principle, and thus carries a conceptual link to Gradient Boosting (GB).
>
> Shared Principles:
>
> MGDL, RF, and GB share the following common features:
>
> 1.	Iterative Training: The optimization is decomposed into multiple ($N$) sequential stages (grades).
>
> 2.	Residual Correction: In stage (grade)  $j$, a new network $f_j$ is trained to model the residual error leftover from the combined output of previous stages, $F_{j-1}$.
>
> 3.	Additive Result: The final outcome $F:=F_N$ is the sum of the components learned at all stages: $F=\sum_{j=1}^N f_j$.
>
> Formal Distinctions: Composition, Depth Dilemma, and Convexity
>
> MGDL fundamentally differs from traditional RF and GB by how the components are structurally integrated and optimized, addressing the inherent limitations of both traditional methods when scaling to deep architectures.
>
> 1.	Architectural Composition (Structural Difference):
>
> o	In traditional RF and GB, each new learner $f_j$ is typically an independent network (often a weak learner like a decision tree or a new, standalone deep network).
>
> o	In MGDL, the grade component $f_j$ is structurally composed with all previous, fixed components. Specifically, $ f_j = N_j \circ \phi_{j-1}^* $, where $\phi_{j-1}^* = f_{j-1}^* \circ \cdots \circ f_1^* $ represents the output of all frozen, preceding networks (with the output layers removed). This frozen block serves as a fixed, adaptive feature extractor or basis for the new shallow network $N_j$. Crucially, the final MGDL model $F$ is a sum of compositionally deep networks in a stair-shape, while RF/GB typically results in a pure additive ensemble.
>
> 2.	Depth Dilemma Avoidance:
>
> •	Traditional RF and GB face a depth dilemma: using a shallow learner $f_j$ improves training stability but limits expressiveness, whereas using a deep learner increases expressive power but exacerbates training instability.
> •	MGDL avoids this trade-off entirely. By using the frozen $\phi_{j-1}^*$ as the input features, MGDL leverages the expressive power of the full, accumulated deep architecture while only training the parameters of the shallow network $N_j$, thus preserving stability.
>
> 3.	Convexity Guarantee (Optimization Advantage):
>
> o	Neither traditional RF nor GB applied to deep networks provides a formal guarantee regarding the convexity of the optimization landscape for the new learner $f_j$.
>
> o	In MGDL, when the ReLU activation function is used and each grade component $N_j$ is a single-hidden-layer network, we formally prove that the original nonconvex optimization problem is decomposed into $N$ sequential convex subproblems. This theoretical result is a key contribution of MGDL, guaranteeing robust convergence where single-grade deep learning (SGDL) often fails.
>
> Reply to question item (2):
>
> Yes, empirically, it does. All of our deep learning experiments in section 5 (e.g., image restoration , CIFAR-100,) utilized Adam optimizers and demonstrate the stability benefits of MGDL.

---

> ### Author Response · Authors · 2025-11-16
> **Reply to question item (3)**
>
> Reply to question item (3):
>
> MGDL is complementary to normalization and skip connections, and our successful results on complex architectures (CNNs, Transformers) confirm this.
>
> • Skip Connections (e.g., ResNet): Skip connections are a network design that inherently models residuals, $\mathcal{F}(\mathbf{x}) = \mathbf{x} + \mathcal{H}(\mathbf{x})$. MGDL is an optimization strategy that tackles the learning of $\mathcal{F}$ (or its components) in grades. MGDL can be applied directly to a ResNet, where it helps the gradient flow and prevents Hessian spectral norms from exploding during the optimization of the deep architecture, providing stability in addition to the architectural stability of the skip connections.
>
> • Normalization (e.g., BatchNorm): Normalization layers help stabilize the internal feature distributions during training. MGDL stabilizes the overall optimization landscape. These are independent mechanisms that work synergistically.
>
> We hope these responses clarify the novelty and impact of MGDL. We are committed to making the suggested revisions to the manuscript, particularly adding the comparison to other staged methods and clarifying the boundary of applicability, to achieve the highest standards for publication.

---

### Meta-Review · Area_Chair_biNQ · 2026-01-10

**Summary:**

The idea is potentially interesting and the paper has several strong elements (clarity, stability motivation, and consistent gains on some tasks), but the current version falls short on (i) theoretical contribution depth and novelty relative to how it is framed, and (ii) empirical rigor and fairness needed to justify broad superiority claims. On the theory side, the central convergence statements largely instantiate standard gradient-descent smoothness arguments, and the more distinctive convex-reformulation result applies only in a restricted setting; as written, it does not convincingly bridge to the behavior of deep modern architectures beyond offering intuition. On the empirical side, much of the reported advantage is demonstrated under training conditions that intentionally remove common stabilizing practices used in strong SGDL baselines (for example, learning-rate schedules and warmup), which makes it difficult to conclude that MGDL is better rather than simply less sensitive under an intentionally challenging setup. In addition, the classification evidence is not reported at the level typically expected for canonical benchmarks, and the baseline set is thin given the generality of the claims. I encourage the authors to resubmit with a tightened scope, stronger baselines under standard training protocols, clearer classification reporting with conventional losses and metrics, and a more precise articulation of what is new relative to prior stagewise and residual-fitting methods—ideally supported by at least one stronger benchmark and targeted ablations (including schedule-tuned SGDL and optional global fine-tuning after MGDL) to solidify the conclusions.

**Reviewer Concerns:**

Reviewer CXYz: The rebuttal adequately addressed the reviewer’s main clarification requests. In particular, the authors (a) toned down the “recently emerged” / adoption language, (b) clarified that the experiments are run with Adam (so the stability story is not restricted to full-batch GD in practice), and (c) added a reasonable discussion of where MGDL is likely to help versus where it may need end-to-end fine-tuning. The remaining concern is that these answers mostly strengthen positioning and interpretation; they do not fully resolve whether the submission’s theoretical contributions are strong enough for ICLR beyond standard GD arguments, nor do they provide (in the current version) the kind of empirical “fairness” comparison against well-tuned SGDL training that would justify broad claims.

Reviewer qcqR: The rebuttal directly addressed the “textbook” nature of Theorems 1–2 by conceding that the convergence bounds themselves are standard and reframing the point as a contrastive application to MGDL, along with an added numeric illustration of Hessian spectral norms/learning-rate ranges. The rebuttal also clarified the smoothness mismatch with ReLU by positioning the C² assumption as technical and arguing the conclusions persist empirically. What remains outstanding is that the novelty still hinges on an argument that MGDL structurally reduces curvature in a meaningful and general way, but this is not yet established as a crisp theoretical contribution (rather than an empirical observation plus intuition), and the bounded-iterate/smoothness assumptions remain strong without being discharged.

Reviewer ybLx: The rebuttal responded in detail, but most of the reviewer’s core criticisms remain outstanding for the current submission. The reviewer’s main points were that the early theory is not substantively new, that the convexity result is of limited practical value as presented, and that the empirical evaluation is incomplete/insufficiently contextualized (including missing or nonstandard reporting for classification). The rebuttal partially addresses these by (i) clarifying that the paper’s goal is “stability over SOTA,” (ii) citing prior work and offering arguments about spectral bias/high-frequency structure, and (iii) promising to add missing CIFAR test accuracy, cross-entropy runs, additional baselines/context, and global fine-tuning. However, these are largely prospective improvements, and the present version still does not convincingly meet the reviewer’s bar on either theoretical substance or empirical completeness.

Reviewer zTvu: The rebuttal addressed some concerns at the level of rationale and future additions (e.g., explaining why schedules were omitted to isolate intrinsic stability, committing to add schedule-based comparisons in an appendix, and clarifying distinctions from layer-wise methods). Still outstanding are the major empirical scale and benchmarking concerns (the absence of a strong canonical benchmark to support a general “outperforms” claim), and the reviewer’s view that the theory is disconnected from the deep-network empirical gains. The rebuttal’s response (“future work” for large-scale benchmarks; schedule experiments to be added) may reduce the severity, but does not fully resolve these issues in the current submission.

**Reviewer Scores:**

CXYz (rated 4): Likely would move slightly upward after discussion, since their concrete questions (optimizer realism, applicability boundaries, relation to boosting/layer-wise training) were answered directly and coherently. I would expect 4 → 5 (weak accept), or remain 4 if they still weight missing comparative baselines heavily.

qcqR (rated 4): The rebuttal acknowledged the main critique (standard theorems) and added a curvature/learning-rate illustration plus clarification on ReLU/smoothness. This may improve confidence but does not fundamentally change the novelty concern. I expect 4 → 4 (unchanged).

ybLx (rated 2): The rebuttal engaged seriously, but many fixes are described as revisions-to-come (missing classification reporting, stronger baselines, scope tightening). Given the reviewer’s stance that the required changes are too large for acceptance “as is,” I expect 2 → 2 (unchanged), with a small chance of 2 → 3 if they credit the planned additions.

zTvu (rated 2): The rebuttal explains experimental design choices and clarifies positioning, but does not provide the kind of large-scale or best-practice baseline evidence the reviewer requested. I expect 2 → 2 (unchanged), with a small chance of 2 → 3 if they become more persuaded by the “intrinsic stability” framing.

---

### Decision · Program_Chairs · 2026-01-26

Reject